# On the Convergence of Adam-Type Algorithm for Bilevel Optimization under Unbounded Smoothness

**Xiaochuan Gong**                                                          *xgong2@gmu.edu*
*George Mason University*

**Jie Hao**                                                                *jhao6@gmu.edu*
*George Mason University*

**Mingrui Liu**                                                          *mingruil@gmu.edu*
*George Mason University*

**Reviewed on OpenReview:** *https://openreview.net/forum?id=cPnmtVnhk4*

## Abstract

Adam has become one of the most popular optimizers for training modern deep neural networks, such as transformers. However, its applicability is largely restricted to single-level optimization problems. In this paper, we aim to extend vanilla Adam to tackle bilevel optimization problems, which have important applications in machine learning, such as meta-learning. In particular, we study stochastic bilevel optimization problems where the lower-level function is strongly convex and the upper-level objective is nonconvex with potentially unbounded smoothness. This unbounded smooth objective function covers a broad class of neural networks, including transformers, which may exhibit non-Lipschitz gradients. In this work, we introduce AdamBO, a single-loop Adam-type method that achieves $\widetilde{O}(\epsilon^{-4})$ oracle complexity to find $\epsilon$-stationary points, where the oracle calls involve stochastic gradient or Hessian/Jacobian-vector product evaluations. The key to our analysis is a novel randomness decoupling lemma that provides refined control over the lower-level variable. We conduct extensive experiments on various machine learning tasks involving bilevel formulations with recurrent neural networks (RNNs) and transformers, demonstrating the effectiveness of our proposed Adam-type algorithm.

## 1 Introduction

The Adam algorithm (Kingma & Ba, 2014) is one of the most popular optimizers for training modern deep neural networks due to their computational efficiency and minimal need for hyperparameter tuning. For example, Adam has become the default choice for training transformers (Vaswani et al., 2017; Devlin et al., 2018) and vision transformers (ViT) (Dosovitskiy et al., 2021). Practitioners favor Adam and adaptive gradient methods in general because they significantly outperform stochastic gradient descent (SGD) for certain models, such as transformers (Zhang et al., 2019; Crawshaw et al., 2022; Kunstner et al., 2023; Ahn et al., 2023). Recently, there is a line of work analyzing the convergence of Adam under various assumptions (Guo et al., 2021b; Défossez et al., 2020; Wang et al., 2022; Zhang et al., 2022; Li et al., 2023a).

Despite the empirical and theoretical advances of Adam, it is only applicable for single-level optimization problems such as the empirical risk minimization. However, there is a huge class of machine learning problems which are inherently bilevel optimization problems (Bracken & McGill, 1973; Dempe, 2002), including meta-learning (Franceschi et al., 2018; Rajeswaran et al., 2019), reinforcement learning (Konda & Tsitsiklis, 2000), hyperparameter optimization (Franceschi et al., 2018; Feurer & Hutter, 2019) and continual learning (Borsos et al., 2020; Hao et al., 2023). Therefore, an important question arises: *How can we extend the applicability of vanilla Adam to solve bilevel optimization problems, while ensuring both provable theoretical convergence guarantees and strong empirical performance for machine learning applications?*

In this paper, we provide a positive answer to this question, under the setting of bilevel optimization under unbounded smoothness (Hao et al., 2024; Gong et al., 2024a). In particular, the bilevel optimization in this setting has the following form:

$$\min_{x \in \mathbb{R}^{d_x}} \Phi(x) := f(x, y^*(x)), \quad \text{s.t.} \ \ y^*(x) = \arg \min_{y \in \mathbb{R}^{d_y}} g(x, y), \tag{1}$$

where $f$ and $g$ are upper- and lower-level functions respectively, and $f$ satisfies a unbounded smoothness condition (see Definition 3.1) and $g$ is a strongly-convex function in $y$. One example satisfying this particular setting is meta-learning (Finn et al., 2017; Franceschi et al., 2018) with certain machine learning models such as RNNs (Elman, 1990) or transformers (Vaswani et al., 2017), where $x$ represents all layers except for the prediction head, $y$ represents the prediction head, and the goal is to learn the shared model parameter $x$ to find a common representation such that it can quickly adapt to various tasks by simply adjusting the task-specific prediction head $y$. The unbounded smoothness condition for the upper-level function $f$ is particularly relevant in this paper for two main reasons. First, recent studies have demonstrated that the gradient's Lipschitz constant (i.e., the smoothness constant) is unbounded in various modern neural networks, including RNNs and transformers (Zhang et al., 2020b; Crawshaw et al., 2022; Hao et al., 2024). Second, Adam is empirically successful on training these neural networks (Vaswani et al., 2017; Kunstner et al., 2023) and its convergence under unbounded smoothness was recently proved within the single-level optimization framework (Li et al., 2023a). Therefore it is natural and imperative to design new Adam-type algorithms, building on the vanilla Adam approach, to solve bilevel optimization problems in the unbounded smoothness setting.

We introduce an Adam-type algorithm for such bilevel optimization problems with provable convergence guarantees. Our algorithm is called Adam for Bilevel Optimization (AdamBO). AdamBO begins by running a few iterations of SGD to warm-start the lower-level variable, after which it simultaneously applies vanilla Adam updates to the upper-level variable and SGD updates to the lower-level variable. The primary challenge for the convergence analysis of AdamBO is tackling the complicated dependency between the upper-level hypergradient bias and the lower-level estimation error when the upper-level performs the vanilla Adam update. The convergence analysis of AdamBO for unbounded smooth upper-level functions builds upon the insight of regarding bilevel optimization as a stochastic optimization problem under distributional drift (Gong et al., 2024a), but with a few important differences. First, our analysis incorporates a novel randomness decoupling lemma for lower-level error control, which arises from using Adam updates for the upper-level variable. Second, unlike (Hao et al., 2024; Gong et al., 2024a), the lower-level error in our setting is not necessarily small across iterations, requiring a more refined analysis to handle the hypergradient bias and establish convergence guarantees. Our main contributions are summarized as follows.

- We design a variant of Adam, called AdamBO, for solving bilevel optimization problems under the unbounded smoothness setting. We prove that AdamBO converges to $\epsilon$-stationary points with $\widetilde{O}(\epsilon^{-4})$ oracle complexity.

- We develop a novel randomness decoupling lemma for lower-level error control and a refined analysis for the hypergradient bias, which are of independent interest and could be applied to analyzing the convergence of other adaptive optimizers in bilevel optimization.

- We conduct experiments on meta-learning and deep AUC maximization for text classification tasks with RNNs and transformers to verify the effectiveness of the proposed Adam-type algorithms. We show that AdamBO consistently outperforms other bilevel algorithms during the training process. Notably, for the transformer model, they improve the training (testing) AUC by at least 14% (7%) over other baselines. The running time results indicate that our algorithms converge much faster than baselines.

## 2 Related Work

**Convergence Analysis of Adam.** Adam was proposed by (Kingma & Ba, 2014) and the convergence guarantee was established under the framework of online convex optimization. Reddi et al. (2019) identified

a divergence example of Adam under fixed hyperparameters and designed new variants to fix the divergence issue of Adam. Recently, there is a line of work analyzing the convergence of Adam under various assumptions and problem-dependent hyperparameter choices (Zhou et al., 2018; Guo et al., 2021b; Défossez et al., 2020; Wang et al., 2022; Zhang et al., 2022; Li et al., 2023a). The most related work to our paper is (Li et al., 2023a), which studied the convergence of Adam under relaxed assumptions (i.e., generalized smoothness as defined by (Li et al., 2023a)). However, all of these works only consider Adam within the single-level optimization framework and are not applicable for bilevel optimization problems.

**Bilevel Optimization.** Bilevel optimization was extensively studied in the literature, most of which focus on asymptotic convergence guarantees (Bracken & McGill, 1973; Vicente et al., 1994; Anandalingam & White, 1990; White & Anandalingam, 1993). Ghadimi & Wang (2018) studied bilevel optimization algorithms with non-asymptotic convergence guarantees when the lower-level function is strongly convex. The complexity results were later improved by a series of work (Hong et al., 2023; Ji et al., 2021; Chen et al., 2021; Dagréou et al., 2022; Kwon et al., 2023; Chen et al., 2023a). When each realization of the functions has a Lipschitz stochastic gradient, several works incorporate momentum-based variance reduction techniques (Cutkosky & Orabona, 2019) to further improve the convergence rate (Khanduri et al., 2021; Guo et al., 2021a; Yang et al., 2021). Recently, (Hao et al., 2024; Gong et al., 2024a;b) considered bilevel optimization with unbounded smoothness for the upper-level function and designed stochastic algorithms with convergence guarantees. However, none of these works use the Adam update under the bilevel optimization setting.

**Relaxed Smoothness.** Zhang et al. (2020b) initiated the convergence analysis of the gradient clipping algorithms under the relaxed smoothness condition, which was motivated by the loss landscape of RNNs and LSTMs. The work of (Zhang et al., 2020b) inspired a line of work focusing on designing various algorithms under the relaxed smoothness condition (Zhang et al., 2020a; Jin et al., 2021; Liu et al., 2022; Crawshaw et al., 2023a;b; Faw et al., 2023; Wang et al., 2023; Li et al., 2023a;b), some of them achieved improved convergence rates (Liu et al., 2023; Reisizadeh et al., 2023; Li et al., 2023a). Several variants of relaxed smoothness were considered in (Crawshaw et al., 2022; Chen et al., 2023b; Hao et al., 2024; Gong et al., 2024a;b). This work considered the same problem setting as in (Hao et al., 2024; Gong et al., 2024a;b), focusing on designing Adam-type algorithms for bilevel optimization with unbounded smooth upper-level functions.

## 3 Preliminaries, Notations and Problem Setup

Denote $\langle \cdot, \cdot \rangle$ and $\| \cdot \|$ as the inner product and Euclidean norm of a vector or spectral norm of a matrix. For any vectors $x$ and $y$, denote $x^2, \sqrt{x}, |x|, x \odot y, x/y$ as the coordinate-wise square, square root, absolute value, product and quotient, respectively. We write $x \preceq y$ to denote the coordinate-wise inequality between $x$ and $y$. We use $\widetilde{O}(\cdot), \widetilde{\Theta}(\cdot), \widetilde{\Omega}(\cdot)$ to denote asymptotic notations that hide polylogarithmic factors of $1/\epsilon$. Define $f, g : \mathbb{R}^{d_x} \times \mathbb{R}^{d_y} \to \mathbb{R}$ as the upper- and lower-level functions, where $f(x, y) = \mathbb{E}_{\xi \sim \mathcal{D}_f}[F(x, y; \xi)]$ and $g(x, y) = \mathbb{E}_{\zeta \sim \mathcal{D}_g}[G(x, y; \zeta)]$, with $\mathcal{D}_f$ and $\mathcal{D}_g$ being the underlying data distributions, respectively. When the lower-level function is strongly convex, the hypergradient has the following form (Ghadimi & Wang, 2018):

$$\nabla \Phi(x) = \nabla_x f(x, y^*(x)) - \nabla^2_{xy} g(x, y^*(x))[\nabla^2_{yy} g(x, y^*(x))]^{-1} \nabla_y f(x, y^*(x)).$$

The goal of this paper is to design Adam-type algorithms that can find $\epsilon$-stationary points of function $\Phi$ (i.e., finding an $x$ such that $\|\nabla \Phi(x)\| \leq \epsilon$). For a given $(x, y)$, we estimate the hypergradient $\nabla \Phi(x)$ using Neumann series approach (Ghadimi & Wang, 2018) with the following formulation:

$$\hat{\nabla} \phi(x, y; \bar{\xi}) = \nabla_x F(x, y; \xi) - \nabla^2_{xy} G(x, y; \zeta^{(0)}) \left[ \frac{1}{l_{g,1}} \sum_{q=0}^{Q-1} \prod_{j=1}^{q} \left( I - \frac{\nabla^2_{yy} G(x, y; \zeta^{(q,j)})}{l_{g,1}} \right) \right] \nabla_y F(x, y; \xi),$$

where $\bar{\xi} := \{\xi, \zeta^{(0)}, \bar{\zeta}^{(0)}, \ldots, \bar{\zeta}^{(Q-1)}\}$ and $\bar{\zeta}^{(q)} := \{\zeta^{(q,1)}, \ldots, \zeta^{(q,q)}\}$ for $q \geq 0$.

**Definition 3.1** (($L_{x,0}, L_{x,1}, L_{y,0}, L_{y,1}$)-Smoothness (Hao et al., 2024, Assumption 1)). Let $z = (x, y)$ and $z' = (x', y')$, there exists $L_{x,0}, L_{x,1}, L_{y,0}, L_{y,1} > 0$ such that for all $z, z'$, if $\|z - z'\| \leq 1/\sqrt{L_{x,1}^2 + L_{y,1}^2}$, then $\|\nabla_x f(z) - \nabla_x f(z')\| \leq (L_{x,0} + L_{x,1} \|\nabla_x f(z)\|) \|z - z'\|$ and $\|\nabla_y f(z) - \nabla_y f(z')\| \leq (L_{y,0} + L_{y,1} \|\nabla_y f(z)\|) \|z - z'\|$.

**Remark**: This definition characterizes the unbounded smoothness of the upper-level function $f$ and has also been used in previous works (Hao et al., 2024; Gong et al., 2024a;b). It can be regarded as a generalization of the relaxed smooth assumption in (Zhang et al., 2020b) and the coordinate-wise relaxed smoothness assumption in (Crawshaw et al., 2022). Moreover, it has been empirically verified for bilevel formulations with RNNs (Hao et al., 2024).

**Assumption 3.2.** Suppose functions $f$ and $g$ satisfy: (i) $f$ is continuously differentiable and $(L_{x,0}, L_{x,1}, L_{y,0}, L_{y,1})$-smooth in $(x, y)$; (ii) For every $x$, $\|\nabla_y f(x, y^*(x))\| \leq l_{f,0}$; (iii) For every $x$, $g(x, y)$ is $\mu$-strongly convex in $y$ for $\mu > 0$; (iv) $g$ is continuously differentiable and $l_{g,1}$-smooth jointly in $(x, y)$; (v) $g$ is twice continuously differentiable, and $\nabla^2_{xy} g, \nabla^2_{yy} g$ are $l_{g,2}$-Lipschitz jointly in $(x, y)$; (vi) Objective function $\Phi$ is bounded from below by $\Phi^*$.

**Remark:** Assumption 3.2 is standard in the bilevel optimization literature (Kwon et al., 2023; Ghadimi & Wang, 2018; Hao et al., 2024). Under this assumption, the objective function $\Phi$ is $(L_0, L_1)$-smooth, see Lemma B.10 in Appendix B for definitions of $L_0, L_1$ and more details.

**Assumption 3.3.** The stochastic estimators are unbiased and satisfy: (i) $\|\nabla_x F(x, y; \xi) - \nabla_x f(x, y)\| \leq \sigma_f$; (ii) $\|\nabla_y F(x, y; \xi) - \nabla_y f(x, y)\| \leq \sigma_f$; (iii) $\|\nabla_y G(x, y; \zeta) - \nabla_y g(x, y)\| \leq \sigma_{g,1}$; (iv) $\|\nabla^2_{xy} G(x, y; \zeta) - \nabla^2_{xy} g(x, y)\| \leq \sigma_{g,2}$; (v) $\|\nabla^2_{yy} G(x, y; \zeta) - \nabla^2_{yy} g(x, y)\| \leq \sigma_{g,2}$.

**Remark:** Assumption 3.3 assumes the noise in the stochastic gradient and Hessian/Jacobian is almost-surely bounded or light-tailed. This is an standard assumption in the literature of optimization for single-level relaxed smooth functions (Zhang et al., 2020b;a), as well as for bilevel optimization under unbounded smooth upper-level functions (Hao et al., 2024; Gong et al., 2024a;b).

**Assumption 3.4.** (i) If $\|z - z'\| \leq 1/\sqrt{L^2_{x,1} + L^2_{y,1}}$, then for every $\xi$, $\|\nabla_x F(x, y; \xi) - \nabla_x F(x, y'; \xi)\| \leq (L_{x,0} + L_{x,1}\|\nabla_x f(x, y)\|)\|y - y'\|$ and $\|\nabla_y F(z; \xi) - \nabla_y F(z'; \xi)\| \leq (L_{y,0} + L_{y,1}\|\nabla_y f(z)\|)\|z - z'\|$; (ii) For every $\zeta$, $G(x, y; \zeta)$ satisfy Assumption 3.2 (iv) and (v).

**Remark**: Assumption 3.4 (i) requires that certain properties of the second argument (i.e., the lower-level variable $y$) in the upper-level function at the population level also hold almost surely for each random realization. Assumption 3.4 (ii) requires each random realization of the lower-level function satisfies the same property as in the population level. Similar assumptions were made implicitly in the bilevel optimization literature (Ghadimi & Wang, 2018). Note that this assumption does not assume any properties in terms of the upper-level variable $x$ under each random realization.

# 4 AdamBO and Convergence Analysis

## 4.1 Algorithm Design and Technique Overview

**Algorithm Design.** Our proposed Adam-type algorithm AdamBO is presented in Algorithm 1. It consists of the following components. First, the algorithm requires several warm-start steps for updating the lower-level variable $y$ for a given initialization of the upper-level variable $x_0$ (line 2), which is designed to obtain a good estimate of the optimal lower-level variable at the very beginning and shares the same spirit of the bilevel algorithms introduced in (Hao et al., 2024; Gong et al., 2024a;b). Second, the algorithm updates both the upper- and lower-level variables simultaneously: the lower-level variable $y$ is updated by SGD, and the upper-level variable $x$ is updated by the vanilla Adam algorithm (lines $3 \sim 9$). Therefore, the upper-level update benefits from the coordinate-wise adaptive learning rate. In contrast, the existing bilevel optimization algorithms under the unbounded smoothness setting use normalized SGD with momentum to update the upper-level variable (Hao et al., 2024; Gong et al., 2024a;b), which use a universal learning rate for every coordinate.

**Main Challenges.** The main challenges for the convergence analysis of AdamBO are listed as follows. First, the analysis of vanilla Adam in the single-level generalized smooth optimization setting (Li et al., 2023a) is not directly applicable for bilevel problems. This is because the hypergradient estimator in bilevel optimization may have a non-negligible bias due to inaccurate estimation of the lower-level variable, whereas the single-level analysis in (Li et al., 2023a) does not need to account for this issue. Second, the existing

---

**Algorithm 1** ADAMBO

---

1: **Input:** $\beta, \beta_{\text{sq}}, \eta, \gamma, \lambda, T_0, T, x_1, y_0$
2: **Initialize** $y_1 = \texttt{SGD}(x_1, y_0, \gamma, T_0)$, $\hat{m}_1 = \hat{\nabla}\phi(x_1, y_1; \bar{\xi}_1)$, $\hat{v}_1 = (\hat{\nabla}\phi(x_1, y_1; \bar{\xi}_1))^2$  (see $\texttt{SGD}$ in Algorithm 2)
3: **for** $t = 1, \ldots, T$ **do**
4:     $y_{t+1} = y_t - \gamma\nabla_y G(x_t, y_t; \zeta_t)$
5:     $m_t = (1 - \beta)m_{t-1} + \beta\hat{\nabla}\phi(x_t, y_t; \bar{\xi}_t)$
6:     $v_t = (1 - \beta_{\text{sq}})v_{t-1} + \beta_{\text{sq}}(\hat{\nabla}\phi(x_t, y_t; \bar{\xi}_t))^2$
7:     $\hat{m}_t = \frac{m_t}{1 - (1-\beta)^t}$, $\hat{v}_t = \frac{v_t}{1 - (1-\beta_{\text{sq}})^t}$
8:     $x_{t+1} = x_t - \frac{\eta}{\sqrt{\hat{v}_t} + \lambda} \odot \hat{m}_t$
9: **end for**

---

algorithms and analyses for bilevel optimization with unbounded smooth upper-level functions require the lower-level error to be small (Hao et al., 2024; Gong et al., 2024a;b), which may not hold for AdamBO. In particular, the existing analysis crucially relies on a fixed update length for the upper-level variable at every iteration (due to normalization): the analysis in (Hao et al., 2024; Gong et al., 2024a;b) views the update of the upper-level variable as a fixed distributional drift for the lower-level function, which is crucial to show that the lower-level error is small and the hypergradient bias is negligible. However, such an argument is not true for AdamBO: the Adam update for the lower-level variable does not have a fixed update size and it depends on randomness from both upper-level and lower-level random variables in the stochastic setting, which make the lower-level error control more challenging.

**Technique Overview.** To address these challenges, one of our main technical contributions is the introduction of a novel randomness decoupling lemma for controlling the lower-level error when the upper-level variable is updated by Adam, as illustrated in Section 4.3.2. This lemma provide a high probability guarantee for the lower-level error control when the upper-level update rule satisfies certain conditions (which are satisfied by the vanilla Adam update rule for the upper-level variable). The key novelty of this lemma lies in the randomness-decoupling fact: the high-probability bound depends solely on the randomness $\{\zeta_t\}_{t=1}^{T}$ from the lower-level random variables, and it holds for any fixed sequence of upper-level variables $\{x_t\}_{t=1}^{T}$ and any fixed upper-level random variables $\{\bar{\xi}_t\}_{t=1}^{T}$ that respect the Adam updates. To describe the condition that Adam satisfies and to prove this lemma, we introduce an auxiliary sequence (defined in (3)) that separates the randomness in the upper- and lower-level random variables, which is new and has not been leveraged in previous bilevel optimization literature.

## 4.2 Main Results

We first introduce some notations and technical definitions. Denote $\sigma(\cdot)$ as the $\sigma$-algebra generated by the random variables within the argument. Let $\mathcal{F}_{\text{init}}$ be the filtration for updating $y_1$ (see Algorithm 2): $\mathcal{F}_{\text{init}} = \sigma(\pi_0, \ldots, \pi_{T_0-1})$. For any $t \geq 2$, define $\mathcal{F}_t^x, \mathcal{F}_t^y$ and $\mathcal{F}_t$ as $\mathcal{F}_t^x = \sigma(\bar{\xi}_1, \ldots, \bar{\xi}_{t-1})$, $\mathcal{F}_t^y = \sigma(\zeta_1, \ldots, \zeta_{t-1})$ and $\mathcal{F}_t = \sigma(\mathcal{F}_{\text{init}} \cup \mathcal{F}_t^x \cup \mathcal{F}_t^y)$. We use $\mathbb{E}_t[\cdot]$ to denote the conditional expectation $\mathbb{E}[\cdot \mid \mathcal{F}_t]$. We also use $c_1, c_2, c_3$ to denote small enough constants and $C_1, C_2$ to denote large enough constants, all of which are independent of $\epsilon$ and $\delta$, where $\epsilon$ denotes the target gradient norm and $\delta$ denotes the failure probability. The definitions of problem-dependent constants $\sigma_\phi, C_{\phi,0}, C_{\phi,1}, \Delta_1, L_0, L_1, L, C_\beta$ are comprehensively listed in Appendix D.1.

**Theorem 4.1.** *Suppose Assumptions 3.2 to 3.4 hold. Let $G$ be a constant satisfying $G \geq \max\left\{4\lambda, 2\sigma_\phi, 4C_{\phi,0}, \frac{C_{\phi,1}}{L_1}, \sqrt{\frac{C_1\Delta_1 L_0}{C_L}}, \frac{C_1\Delta_1 L_1}{C_L}\right\}$. Given any $\epsilon > 0$ and $\delta \in (0,1)$, choose $0 \leq \beta_{\text{sq}} \leq 1$, $\beta = \widetilde{\Theta}(\epsilon^2)$, $\gamma = \widetilde{\Theta}(\epsilon^2)$, $\eta = \widetilde{\Theta}(\epsilon^2)$, $Q = \widetilde{\Theta}(1)$, $T_0 = \widetilde{\Theta}(\epsilon^{-2})$. Run Algorithm 1 for $T = \max\left\{\frac{1}{\beta^2}, \frac{C_2\Delta_1 G}{\eta\epsilon^2}\right\} = \widetilde{O}(\epsilon^{-4})$ iterations. Then with probability at least $1-\delta$ over the randomness in $\mathcal{F}_{T+1}$, we have $\frac{1}{T}\sum_{t=1}^{T}\|\nabla\Phi(x_t)\| \leq \epsilon^2$.*

**Remark 1**: The full statement of Theorem 4.1 with detailed parameter choices is deferred to Theorem D.12 in Appendix D.7. Theorem 4.1 provides the convergence guarantee for Algorithm 1: AdamBO converges to $\epsilon$-stationary points with $T_0 + QT = \widetilde{O}(\epsilon^{-4})$ oracle complexity. This complexity result matches that of non-adaptive bilevel optimization algorithms in (Hao et al., 2024; Gong et al., 2024a) when the upper-level

Table 1: Comparison of Adam-related papers under different settings and assumptions. ✓ represents dropping the bias correction term for the first-order momentum while keeping it for the second-order momentum. $d$ denotes the dimension. Only the key assumptions (see Appendix E for details) are listed here.

| Adam Paper | Problem | Stochastic Setting | Assumptions | Choice of $\beta$ | Bias Correction | Complexity |
|---|---|---|---|---|---|---|
| De et al. (2018) | Single-Level | Deterministic | E.1(A) + E.2 | $1 - O(\epsilon)$ | ✗ | $O(\epsilon^{-6})$ |
| Défossez et al. (2020) | Single-Level | Stochastic (Expectation) | E.1(A) + E.2 | $[\beta_{sq}, 1]$ | ✓ | $\widetilde{O}(d\epsilon^{-4})$ |
| Guo et al. (2021b) | Single-Level | Stochastic (Expectation) | E.1(A) + E.2 [1] | $O(\epsilon^2)$ | ✗ | $O(\epsilon^{-4})$ |
| Zhang et al. (2022) | Single-Level | Stochastic (Finite Sum) | E.1(A) | $(1 - \sqrt{1 - \beta_{sq}}, 1]$ | ✓ (Randomly Reshuffled) | Not Converge [2] |
| Wang et al. (2022) | Single-Level | Stochastic (Finite Sum) | E.1(B) | $(1 - \sqrt{1 - \beta_{sq}}, 1]$ | ✗ (Randomly Reshuffled) | Not Converge |
| Li et al. (2023a) | Single-Level | Stochastic (Expectation) | E.1(C) | $O(\epsilon^2)$ | ✓ | $O(\epsilon^{-4})$ |
| AdamBO (Theorem 4.1) | Bilevel | Stochastic (Expectation) | E.1(B) [3] | $\widetilde{\Theta}(\epsilon^2)$ | ✓ | $\widetilde{O}(\epsilon^{-4})$ |

Table 2: Comparison of bilevel optimization algorithms under the unbounded smoothness setting.

| Method | Problem | Stochastic Setting | Loop Style | Assumptions | Adam-Type | Learning Rate $\eta$ | Complexity |
|---|---|---|---|---|---|---|---|
| BO-REP (Hao et al., 2024) | Bilevel | Stochastic (Expectation) | Double | Assumptions 3.2 and E.3 | ✗ | $O(\epsilon^3)$ | $\widetilde{O}(\epsilon^{-4})$ |
| SLIP (Gong et al., 2024a) | Bilevel | Stochastic (Expectation) | Single | Assumptions 3.2 and E.3 | ✗ | $\widetilde{\Theta}(\epsilon^3)$ | $\widetilde{O}(\epsilon^{-4})$ |
| AdamBO (Theorem 4.1) | Bilevel | Stochastic (Expectation) | Single | Assumptions 3.2 to 3.4 | ✓ | $\widetilde{\Theta}(\epsilon^2)$ | $\widetilde{O}(\epsilon^{-4})$ |

function exhibits unbounded smoothness, as well as the complexity of Adam for single-level optimization with generalized smooth functions (Li et al., 2023a). It is also worth noting that we choose a larger learning rate $\eta = \widetilde{\Theta}(\epsilon^2)$ for the upper-level updates, compared to $\eta = \widetilde{\Theta}(\epsilon^3)$ used in the SLIP algorithm (Gong et al., 2024a). See Table 2 for a comparison of bilevel optimization algorithms under the unbounded smoothness setting.

**Remark 2**: In Theorem 4.1, we require the momentum parameter $\beta$ to be small. Note that the default choice of $\beta_1$ in Kingma & Ba (2014) is 0.9, which corresponds to $\beta = 0.1$ in our algorithm. This seemingly different choice of $\beta$ (i.e., $\beta = \widetilde{\Theta}(\epsilon^2)$ in Theorem 4.1 versus $\beta = 0.1$ in Kingma & Ba (2014)) is due to the problem setting. In practice, Adam is typically used to minimize functions with finite-sum structure Zhang et al. (2022), while our paper considers a more challenging stochastic optimization setting. In stochastic optimization setting, constant $\beta$ makes Adam diverge. For example, (Reddi et al., 2019, Theorem 3) has shown that there is a stochastic convex optimization problem for which Adam does not converge for any constant $\beta$. We believe a small $\beta = \widetilde{\Theta}(\epsilon^2)$ is a reasonable surrogate for $\beta = 0.1$ under stochastic optimization setting: such a choice of $\beta$ is also used in the analysis of Adam under the single-level stochastic optimization setting with a generalized smooth upper-level function Li et al. (2023a). Moreover, existing convergence analyses of Adam that do not need such choice of $\beta$ require other strong assumptions for the objective function, which is incompatible to our setting. Also, Figure 6 shows that AdamBO's performance remains largely robust to the choice of $(\beta, \beta_{sq})$.

**Remark 3.** Existing convergence analyses of (single-level) Adam that do not need such choice of $\beta$ require other strong assumptions for the objective function, which is incompatible to our setting. They either rely on the bounded gradient assumption (De et al., 2018; Défossez et al., 2020), or they only prove convergence to some neighborhood of stationary points with a constant radius unless assuming the strong growth condition under the finite sum setting (Zhang et al., 2022; Wang et al., 2022). Please see Table 1 for details.

**Remark 4**: One limitation of our complexity dependence on $\lambda$ is $O(\lambda^{-2})$, which can be large since $\lambda$ is typically small in practice. To address this concern, we conduct additional experiments in Figure 5 to evaluate the empirical sensitivity of our algorithm to $\lambda$. Although the default choice of $\lambda$ is $10^{-8}$ (Kingma & Ba, 2014), increasing it up to $10^{-4}$ only causes minor differences in AUC maximization, and increasing it up to $10^{-3}$ leads to minor changes in hyper-representation performance with BERT (Devlin et al., 2018).

---

[1] (Guo et al., 2021b, Assumption 2) can be implied by Assumption E.2, although it is weaker.
[2] Adam can converge with an additional strong growth condition (Zhang et al., 2022; Wang et al., 2022).
[3] Under Assumption 3.2, the objective function $\Phi$ is $(L_0, L_1)$-smooth, see Lemma B.10 for details.

### 4.3 Proof Sketch

In this section, we provide a proof sketch for Theorem 4.1. The detailed proof can be found in Appendix D. Let $y_t^* = y^*(x_t)$. The key idea is to provide a high probability bound of lower-level estimation error $\|y_t - y_t^*\|$ when the upper-level variable $x$ is updated by the vanilla Adam. Lemma 4.4 provides such a guarantee: the lower-level error $\|y_t - y_t^*\|$ is bounded by a function of the initial estimation error $\|y_1 - y_1^*\|$, the variance term $\sigma_{g,1}^2$, and an auxiliary momentum estimator of the hypergradient $\|\hat{u}_t\|$ (see definition of $\hat{u}_t$ in (6)). Based on Lemma 4.4, we introduce Lemma 4.5 and 4.6, which incorporate the lower-level error into the upper-level problems and adopt the stopping time technique of Adam (Li et al., 2023a) to prove the convergence. The proof of Lemma 4.4 is a direct application of the randomness decoupling lemma (i.e., Lemma 4.2 in Section 4.3.2). All of the proofs in this section are based on Assumptions 3.2 to 3.4. The full statements and proofs of Lemmas 4.2 to 4.6 are provided in Appendices C.2 and D.3 to D.6.

#### 4.3.1 Equivalent Update Rule of AdamBO

Let $\alpha_t = \frac{\beta}{1-(1-\beta)^t}$ and $\alpha_t^{\mathrm{sq}} = \frac{\beta_{\mathrm{sq}}}{1-(1-\beta_{\mathrm{sq}})^t}$. Inspired by (Li et al., 2023a), we provide an equivalent yet simpler update rule of lines 5-8 of Algorithm 1 (see Proposition A.1 for more details):

$$\hat{m}_t = (1 - \alpha_t)\hat{m}_{t-1} + \alpha_t \hat{\nabla}\phi(x_t, y_t; \bar{\xi}_t),$$
$$\hat{v}_t = (1 - \alpha_t^{\mathrm{sq}})\hat{v}_{t-1} + \alpha_t^{\mathrm{sq}}(\hat{\nabla}\phi(x_t, y_t; \bar{\xi}_t))^2.$$

#### 4.3.2 Randomness Decoupling Lemma

In this section, we introduce the random decoupling lemma (Lemma 4.2) for the lower-level error control. The rationale is as follows: for any given upper-level variable sequence and any given randomness from the upper-level updates that satisfy certain conditions and are consistent with the AdamBO updates, we can bound the lower-level error with high probability, where the randomness is taken solely from lower-level random variables. Specifically, for any given sequence $\{\tilde{x}_t\}$, define $\tilde{\zeta}_t$ and $\hat{\xi}_t$ as the random variables from the lower-level and upper-level, respectively, at the $t$-th iteration (see (25) for definition). We consider the following update rule for $\{\tilde{y}_t\}$, which is exactly SGD and corresponds to line 4 of Algorithm 1:

$$\tilde{y}_{t+1} = \tilde{y}_t - \gamma \nabla_y G(\tilde{x}_t, \tilde{y}_t; \tilde{\zeta}_t). \tag{2}$$

Let $\tilde{y}_t^* = y^*(\tilde{x}_t)$ and $\tilde{\mathcal{F}}_t^y = \sigma(\tilde{\zeta}_1, \dots, \tilde{\zeta}_{t-1})$. Denote $\tilde{G}_t := \max_{k \leq t} \|\nabla\Phi(\tilde{x}_k)\|$, $\tilde{L}_t := L_0 + L_1 \tilde{G}_t$. We also introduce the following auxiliary sequences $\{\tilde{m}_t\}$ and $\{\tilde{u}_t\}$ for our analysis:

$$\tilde{m}_t = (1 - \alpha_t)\tilde{m}_{t-1} + \alpha_t \hat{\nabla}\phi(\tilde{x}_t, \tilde{y}_t; \hat{\xi}_t),$$
$$\tilde{u}_t = (1 - \alpha_t)\tilde{u}_{t-1} + \alpha_t \hat{\nabla}\phi(\tilde{x}_t, \tilde{y}_t^*; \hat{\xi}_t). \tag{3}$$

**Lemma 4.2** (Randomness Decoupling). *Given any sequence $\{\tilde{x}_t\}$ and randomness $\{\hat{\xi}_t\}$ such that*

$$\|\tilde{x}_{t+1} - \tilde{x}_t\|^2 \leq \frac{2\eta^2}{\lambda^2} \left( \|\tilde{u}_t\|^2 + \tilde{L}_t^2 \sum_{j=1}^t d_{t,j} \|\tilde{y}_j - \tilde{y}_j^*\|^2 \right), \tag{4}$$

*where $\{d_{t,j}\}_{j=1}^t$ is defined in (9). Let $\{\tilde{y}_t\}$ be the iterates generated by the update rule (2) with $\gamma \leq 1/2l_{g,1}$ and choose $\gamma = 2\beta/\mu$. For any given $\delta \in (0,1)$ and all $t \geq 1$, the following holds with probability at least $1 - \delta$ over the randomness in $\tilde{\mathcal{F}}_{T+1}^y$:*

$$\|\tilde{y}_t - \tilde{y}_t^*\|^2 \leq \left(1 - \frac{\mu\gamma}{2}\right)^{t-1} \|\tilde{y}_1 - \tilde{y}_1^*\|^2 + \frac{8\gamma\sigma_{g,1}^2}{\mu} \ln\frac{eT}{\delta} \quad \text{(Variance)}$$

$$+ \left(\frac{4\eta^2 l_{g,1}^2}{\lambda^2 \mu^3 \gamma} \|\tilde{y}_1 - \tilde{y}_1^*\|^2 + \frac{16\eta^2 l_{g,1}^2 \sigma_{g,1}^2}{\lambda^2 \mu^4}\right) \sum_{i=1}^{t-1} \left(1 - \frac{\mu\gamma}{2}\right)^{t-1-i} \tilde{L}_i^2 \quad \text{(Drift)} \tag{5}$$

$$+ \frac{4\eta^2 l_{g,1}^2}{\lambda^2 \mu^3 \gamma} \sum_{i=1}^{t-1} \left(1 - \frac{\mu\gamma}{2}\right)^{t-1-i} \|\tilde{u}_i\|^2 + \frac{64\eta^4 l_{g,1}^4}{\lambda^4 \mu^8 \gamma^4} \sum_{i=1}^{t-1} \left(1 - \frac{\mu\gamma}{2}\right)^{t-1-i} \alpha_i \tilde{L}_i^2 \|\tilde{u}_i\|^2. \quad \text{(Drift)}$$

**Remark**: Lemma 4.2 shows that, when (4) holds for any sequence $\{\tilde{x}_t\}$ and any $\{\hat{\xi}_t\}$ (as satisfied by the vanilla Adam update for the upper-level variable), the lower-level error can be controlled with high probability as in (5). In addition, the high probability is taken over the randomness solely from the lower-level filtration $\tilde{\mathcal{F}}^y_{T+1}$. This lemma provides a technical tool to control the lower-level error without concerns about the dependency issues from the upper-level randomness. In particular, the right-hand side of (5) consists of two parts: the standard variance term, which does not involve the update of $\{\tilde{x}_t\}$ over $t$; and the drift terms, which account for the update of $\{\tilde{x}_t\}$ over time.

### 4.3.3 Applications of the Randomness Decoupling Lemma and Remaining Proof

Given a large enough constant $G$, denote $L = L_0 + L_1 G$ and $\psi = C_L G^2 / 2L$, where $G$ is defined in Theorem 4.1 and $C_L$ is defined in (42). Let $\tau$ be the first time the sub-optimality gap becomes strictly larger than $\psi$, truncated at $T + 1$; formally,

$$\tau := \min\{t \mid \Phi(x_t) - \Phi^* > \psi\} \wedge (T + 1).$$

Then, for any $t < \tau$, we have $\Phi(x_t) - \Phi^* \leq \psi$ and therefore $\|\nabla\Phi(x_t)\| \leq G$. Based on our discussion above, we can analyze the updates before time $\tau$ and construct a Lyapunov function to derive an upper bound on $\Phi(x_\tau) - \Phi^*$. On the other hand, if $\tau \leq T$, the definition of $\tau$ immediately gives a lower bound, namely $\Phi(x_\tau) - \Phi^* > \psi$. If this lower bound exceeds the upper bound, we obtain a contradiction, which implies $\tau = T + 1$. That is, before the algorithm terminates, the sub-optimality gap and gradient norm are always bounded by $F$ and $G$, respectively. Based on Lemma D.1, we know that if $t < \tau$, we have both $\Phi(x_t) - \Phi^* \leq \psi$ and $\|\nabla\Phi(x_t)\| \leq G$. Similar to Section 4.3.2, we introduce the following auxiliary sequence $\{\hat{u}_t\}$ for our analysis:

$$\hat{u}_t = (1 - \alpha_t)\hat{u}_{t-1} + \alpha_t\hat{\nabla}\phi(x_t, y_t^*; \bar{\xi}_t). \tag{6}$$

Now we provide a roadmap explaining how the lemmas are connected in proving the main theorem before formally stating them. Specifically, we first define the stopping time $\tau$ and explain the contradiction framework as above. Next, the warm-start and the lower-level event lemmas (obtained via the randomness-decoupling lemma) provide high-probability control of lower-level tracking error. Under this good event, the upper-level lemma bounds the momentum estimators and update size, and the bias-virtual lemma bounds $\sum_{t<\tau} \|\hat{m}_t - \hat{u}_t\|^2$, i.e., the gap between actual and virtual momentum. Finally, plugging these bounds into the descent inequality and summing over $t < \tau$ yields the required upper bound; this forces $\tau = T + 1$, which gives the final stationarity guarantee.

**Lemma 4.3** (Warm-Start). *Choose $\gamma \leq 1/2l_{g,1}$. With probability at least $1 - \delta/4$ over the randomness in $\mathcal{F}_{init}$ (denote this event as $\mathcal{E}_0$) that:* $\|y_1 - y_1^*\|^2 \leq \left(1 - \frac{\mu\gamma}{2}\right)^{T_0} \|y_0 - y_0^*\|^2 + \frac{8\gamma\sigma_{g,1}^2}{\mu} \ln\frac{4e}{\delta}$.

**Lemma 4.4.** *Under the parameter choices in Lemma D.4, apply Lemma 4.2 with $\{\tilde{x}_t\} = \{x_t\}$, $\{\tilde{y}_t\} = \{y_t\}, \{\tilde{u}_t\} = \{\hat{u}_t\}$ and $\{\tilde{L}_t\} = \{\hat{L}_t\}$, then (5) holds with probability at least $1 - \delta/4$ over the randomness in $\mathcal{F}^y_{T+1}$ (denote this event as $\mathcal{E}_y$).*

**Remark**: Lemma 4.3 and Lemma 4.4 together provide a high probability bound for the lower-level error, where the randomness is taken only from the lower-level filtrations $\mathcal{F}_{\text{init}}$ and $\mathcal{F}^y_{T+1}$. Lemma 4.4 is a direct application of Lemma 4.2 to the actual sequence $\{x_t\}$ and $\{y_t\}$ in Algorithm 1.

**Lemma 4.5.** *If $t < \tau$, we have $\|\nabla\Phi(x_t)\| \leq G$, $\|\hat{u}_t\| \leq C_{u,0}$; under event $\mathcal{E}_0 \cap \mathcal{E}_y$, if $t < \tau$, we have $\|\hat{m}_t\| \leq C_{u,0} + C_{u,1}\varrho$, $\hat{v}_t \preceq (C_{u,0} + C_{u,1}\varrho)^2$, where constants $C_{u,0}, C_{u,1}, \varrho$ are defined in (42) and (52), respectively.*

**Remark**: Lemma 4.5 generalizes the stopping time analysis from the single-level setting (Li et al., 2023a) to the bilevel setting and is useful for upper-level analysis. It shows that the momentum estimators of the hypergradient remains bounded when $t < \tau$ and $\mathcal{E}_0 \cap \mathcal{E}_y$ holds. This implies that $x_{t+1}$ and $x_t$ remains close for small enough $\eta$, allowing us to apply Lemmas B.10 and B.11.

**Lemma 4.6.** *Under event $\mathcal{E}_0 \cap \mathcal{E}_y$ and the parameter choices in Lemma D.4, we have $\sum_{t=1}^{\tau-1} \|\hat{m}_t - \hat{u}_t\|^2 \leq O(\sqrt{T}) + O(1)\sum_{t=1}^{\tau-1}(\|\epsilon_t\|^2 + \|\nabla\Phi(x_t)\|^2)$.*

**Remark**: Lemma 4.6 provides a bound for the difference between the actual momentum $\hat{m}_t$ versus the virtual momentum $\hat{u}_t$ under the good event $\mathcal{E}_0 \cap \mathcal{E}_y$, which is essential for establishing the convergence guarantees for AdamBO.

## 5 Experiments

### 5.1 Hyper-representation learning

Hyper-representation learning, i.e., meta-learning (Finn et al., 2017), aims to find a good meta learner parameterized by $x$, such that it can quickly adapt to a new task $i$ by fine-tuning the corresponding adapter $y_i$. Consider a meta-learning task consisting of $K$ tasks with the training set $\{\mathcal{D}_i^{tr} \mid i = 1, \ldots, K\}$ and validation set $\{\mathcal{D}_i^{val} \mid i = 1, \ldots, K\}$. Each task has a loss function $\mathcal{L}(x, y_i; \xi_i)$ over each sample $\xi_i$. This meta-learning problem can be reformulated as a bilevel optimization, where the lower-level objective function tries to find an optimal task-specific adapter $y_i^*(x)$ on training data $\mathcal{D}_i^{tr}$, and the upper-level minimizes the objective function on validation data $\mathcal{D}_i^{val}$ by finding the optimal meta-learner $x$ with a set of adapters $y = \{y_1^*(x), y_2^*(x), \ldots, y_K^*(x)\}$. We have the following formulation:

$$\min_x \frac{1}{K} \sum_{i=1}^{K} \frac{1}{|\mathcal{D}_i^{val}|} \sum_{\xi \in \mathcal{D}_i^{val}} \mathcal{L}(x, y^*(x); \xi), \quad \text{s.t.,} \ \ y^*(x) = \arg\min_y \frac{1}{K} \sum_{i=1}^{K} \mathcal{L}_{\mathcal{D}_i^{tr}}(x, y_i; \zeta) + \frac{\mu}{2} \|y_i\|^2,$$

where $\mathcal{L}_{\mathcal{D}_i^{tr}}(x, y_i; \zeta) = \frac{1}{|\mathcal{D}_i^{tr}|} \sum_{\zeta \in \mathcal{D}_i^{tr}} \mathcal{L}(x, y_i; \zeta)$. The adapter (parameterized by $y_i$) is typically instantiated as the last linear layer, and the meta learner (parameterized by $x$) is the remaining layers of model, which guarantees that the lower-level function to be strongly-convex when $\mu > 0$.

We conduct meta-learning experiments on a larger language model, specifically an 8-layer BERT (Devlin et al., 2018) model. The experiments are performed on a widely-used question classification dataset TREC (Li & Roth, 2002) (under Creative Commons Attribution 4.0 License), which contains 6 coarse-grained categories. To evaluate our approach on meta-learning, we construct $K = 500$ meta tasks, where the training data $\mathcal{D}_i^{tr}$ and validation data $\mathcal{D}_i^{val}$ for the $i$-th task are randomly sampled from two disjoint categories, with 5 examples per category. Therefore, there are at most 15 unique class-pair types, and the 500 tasks are different episodes sampled from these class pairs. A BERT model, with 8 self-attention layers and a fully-connected layer, is used in our experiment. The self-attention layers serve as representation layers (with their parameters treated as upper-level variables) and the fully-connected layer (with its parameters treated as lower-level variables) serves as an adapter, where each self-attention layer consists of 8 self-attention heads with the hidden size being 768. The fully-connected layer acts as a classifier, with the input dimension of 768 and the output dimension of 6 (corresponding to the 6 categories). Our bilevel optimization algorithm trains the representation layers and the adapter on the meta tasks ($\mathcal{D}^{tr}$ and $\mathcal{D}^{val}$) from scratch, and then evaluate it on the test set $\mathcal{D}^{te}$. During the evaluation phase, we fix the parameters of representation layers and just fine-tune the adapters. We train the models for 20 epochs and compare it with other bilevel optimization baseline algorithms.

We compare with typical meta-learning algorithms, MAML (Rajeswaran et al., 2019) and ANIL (Raghu et al., 2019), and recent bilevel optimization algorithms, StocBio (Ji et al., 2021), TTSA (Hong et al., 2023), SABA (Dagréou et al., 2022), MA-SOBA (Chen et al., 2023a), BO-REP (Hao et al., 2024), SLIP (Gong et al., 2024a). The comparison results of training and testing accuracy are shown in Figure 1. AdamBO achieves fast convergence to the best training and test results among all baselines. One can refer to Appendix F for detailed hyper-parameter choices and experimental settings. All the experiments are run on an single NVIDIA A6000 (48GB memory) GPU and a AMD EPYC 7513 32-Core CPU.

We also conduct the meta-learning experiments for the text classification on dataset Stanford Natural Language Inference (SNLI) (Bowman et al., 2015), which consists of 570k pairs of sentences with 3 classes. We construct $K = 500$ tasks, where each task $\mathcal{D}_i^{tr}$ and $\mathcal{D}_i^{val}$ randomly sample two disjoint categories from the original data, respectively. Empirically, we use mini-batches of meta-tasks for training, with a task batch size of 25. A 3-layer recurrent network is used as representation layers and a fully-connected layer as an adapter. The input dimension, hidden dimension and output dimension are set to be 300, 4096, and 3, respectively.

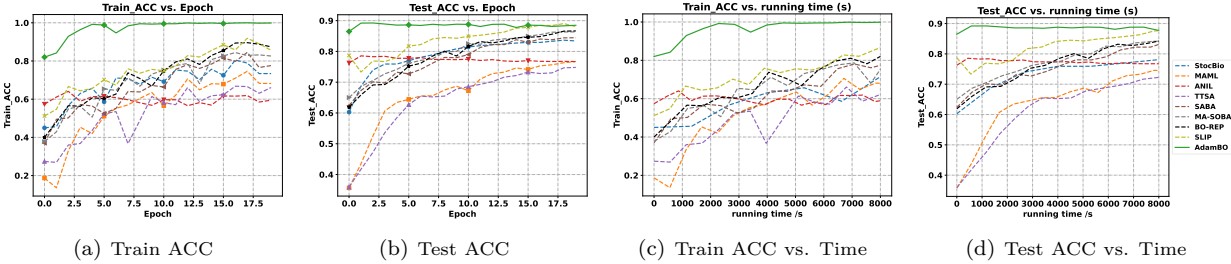

Figure 1: Comparison with bilevel optimization baselines on BERT for hyper-representation.

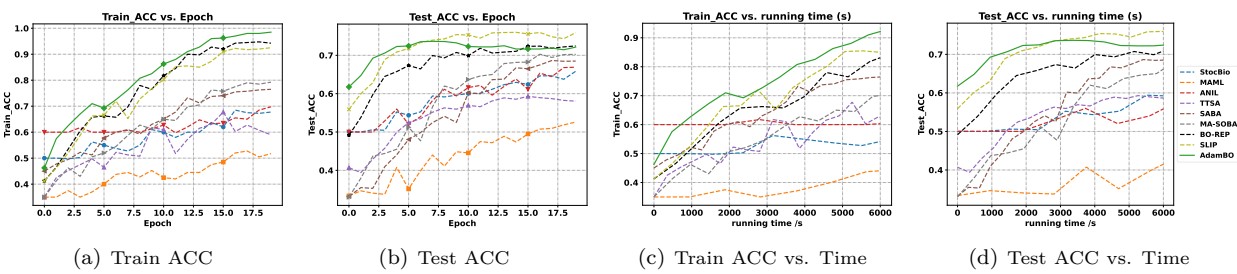

Figure 2: Comparison with bilevel optimization baselines on RNN for hyper-representation.

The comparison results of training and testing accuracy are shown in Figure 2. AdamBO outperforms other baselines on training set, and exhibits faster convergence rate.

## 5.2 Deep AUC Maximization with RNNs/Transformers

The Area Under the ROC Curve (AUC) (Hanley & McNeil, 1983) is a widely used metric for evaluating the effectiveness of binary classification models, especially in the imbalanced data scenarios. It is defined as the probability that the prediction score of a positive example is higher than that of a negative example (Hanley & McNeil, 1982). Deep AUC maximization (Liu et al., 2020; Ying et al., 2016) can be formulated as a min-max optimization problem (Liu et al., 2020): $\min_{\boldsymbol{w} \in \mathbb{R}^d, (a,b) \in \mathbb{R}^2} \max_{\alpha \in \mathbb{R}} f(\boldsymbol{w}, a, b, \alpha) \coloneqq \mathbb{E}_{\boldsymbol{z}}[F(\boldsymbol{w}, a, b, \alpha; \boldsymbol{z})]$, where $F(\boldsymbol{w}, a, b, \alpha; \boldsymbol{z}) = (1-p)(h(\boldsymbol{w}; \boldsymbol{x}) - a)^2 \mathbb{I}_{[c=1]} + p(h(\boldsymbol{w}; \boldsymbol{x}) - b)^2 \mathbb{I}_{[c=-1]} + 2(1 + \alpha)(ph(\boldsymbol{w}; \boldsymbol{x}) \mathbb{I}_{[c=-1]} - (1-p)h(\boldsymbol{w}; \boldsymbol{x}) \mathbb{I}_{[c=1]}) - p(1-p)\alpha^2$, $\boldsymbol{w}$ denotes the model parameter of a deep neural network, and $\boldsymbol{z} = (\boldsymbol{x}, c)$ represents a random training data sample ($\boldsymbol{x}$ represents the feature vector and $c \in \{+1, -1\}$ represents the class label), the function $h(\boldsymbol{w}, \boldsymbol{x})$ is a scoring function for the sample with feature $\boldsymbol{x}$, and $p = \Pr(c = 1)$ indicates the proportion of positive samples in the population. This min-max problem can be reformulated as the form of a bilevel optimization problem with lower-level objective function $g = -f$:

$$\min_{\boldsymbol{w} \in \mathbb{R}^d, (a,b) \in \mathbb{R}^2} \mathbb{E}_{\boldsymbol{z}}[F(\boldsymbol{w}, a, b, \alpha^*(\boldsymbol{w}, a, b); \boldsymbol{z})] \quad \text{s.t.,} \ \alpha^*(\boldsymbol{w}, a, b) \in \arg\min_{\alpha \in \mathbb{R}} -\mathbb{E}_{\boldsymbol{z}}[F(\boldsymbol{w}, a, b, \alpha; \boldsymbol{z})].$$

In above, $(\boldsymbol{w}, a, b)$ is the upper-level variable, and $\alpha$ is the lower-level variable. The lower-level problem is a strongly convex one-dimensional quadratic function with respect to $\alpha$, while the upper-level objective is non-convex and can exhibit unbounded smoothness when using a recurrent neural network or a transformer as the predictive model (Crawshaw et al., 2022; Zhang et al., 2020b).

In our experiment, we focus on tackling an imbalanced text classification task by maximizing the AUC metric. Specifically, we conduct experiments using deep AUC maximization on the imbalanced Sentiment140 dataset (Go et al., 2009), a binary text classification benchmark. Following the approach in (Yuan et al., 2021), we introduce imbalance in the training set using a pre-specified imbalance ratio ($p$) while keeping the test set distribution unchanged. For a given $p$, we randomly remove positive samples (labeled as 1) from the training set until the desired proportion of positive examples is achieved. In our experiment, we set $p$ to

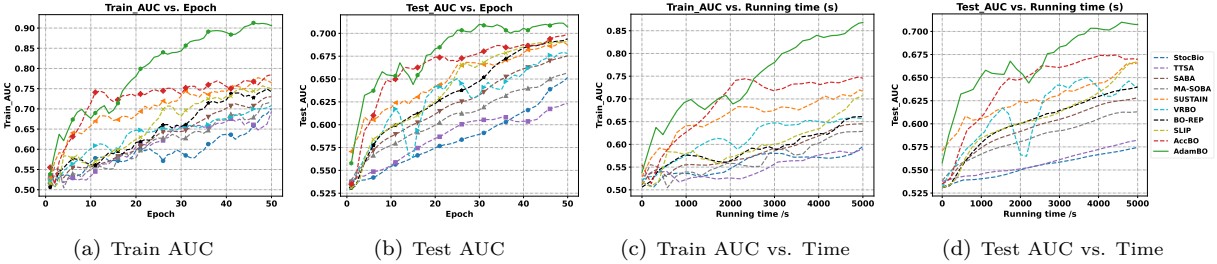

(a) Train AUC  (b) Test AUC  (c) Train AUC vs. Time  (d) Test AUC vs. Time

Figure 3: Transformer for AUC maximization on Sentiment140 dataset with imbalance ratio of 0.9.

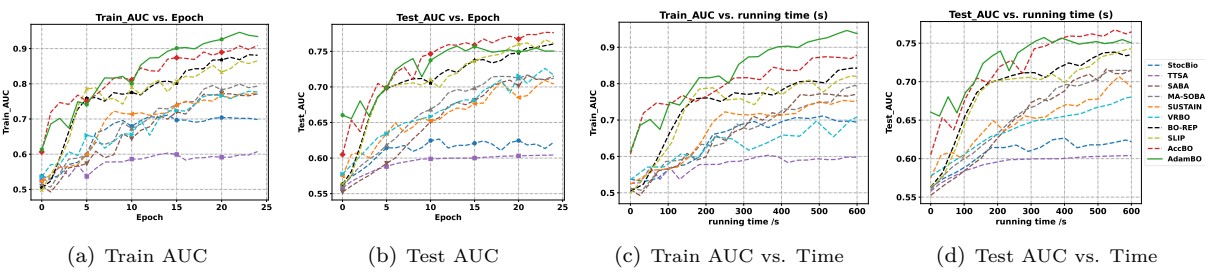

(a) Train AUC  (b) Test AUC  (c) Train AUC vs. Time  (d) Test AUC vs. Time

Figure 4: RNN for AUC maximization on Sentiment140 dataset with imbalance ratio of 0.8. Figures (a), (b) are the results over epochs, Figures (c), (d) are the results over running time.

0.8 (0.9), meaning that 80% (90%) of the training samples are positive examples. We run the experiment using two different models, a two-layer transformer, and a two-layer recurrent neural network (RNN) with the same input dimension of 300, hidden dimension of 4096, and an output dimension of 2.

To evaluate the effectiveness of our proposed bilevel optimization algorithm, we compare with recent bilevel optimization baselines, including StocBio (Ji et al., 2021), TTSA (Hong et al., 2023), SABA (Dagréou et al., 2022), MA-SOBA (Chen et al., 2023a), SUSTAIN (Khanduri et al., 2021), VRBO (Yang et al., 2021), BO-REP (Hao et al., 2024), SLIP (Gong et al., 2024a), and AccBO (Gong et al., 2024b). The training and testing results of the transformer model over 50 epochs are presented in Figure 3(a) and (b), while the corresponding running times are shown in Figure 3(c) and (d). Our proposed Adam-type algorithms, AdamBO, shows the faster convergence rate and significantly outperform other baselines. In particular, the performance on the training AUC (testing AUC) is better by at least 14% (7%) over other baselines. The running time results indicate that AdamBO converges much faster to a high AUC value compared to the other baselines. We also perform the AUC maximization on a RNN model with imbalance rario of 0.8, and the results for both training and testing over 25 epochs are presented in Figure 4(a) and (b), while the corresponding running times are shown in Figure 4(c) and (d). Our proposed Adam-type algorithm, AdamBO, shows the faster convergence rate and significantly outperform other baselines during training process. More detailed parameter tuning and selection can be found in Appendix F.

## 5.3 Sensitivity Analysis of $\lambda$, $\beta$, and $\beta_{\mathrm{sq}}$

In this section, we present a empirical sensitivity analysis of $\lambda$ and ablation studies on the parameters $\beta$ and $\beta_{\mathrm{sq}}$, respectively. In particular, Figure 5 shows that the empirical performance of our algorithm is robust to the choice of $\lambda$. Although the default value of $\lambda$ is $10^{-8}$ (Kingma & Ba, 2014), increasing it to $10^{-4}$ results in only minor differences in AUC maximization, and increasing it to $10^{-3}$ causes only slight variations in hyper-representation performance when using BERT (Devlin et al., 2018). Additionally, Figure 6 demonstrates that AdamBO's performance remains largely robust to the choice of $(\beta, \beta_{\mathrm{sq}})$ within a reasonable range.

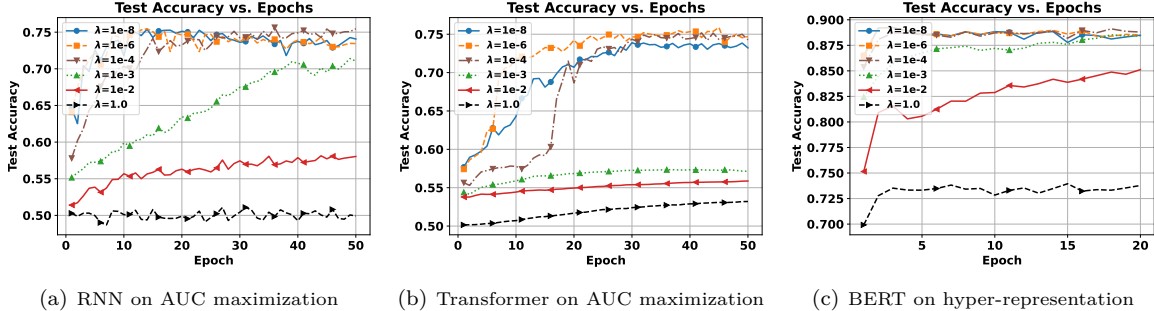

(a) RNN on AUC maximization     (b) Transformer on AUC maximization     (c) BERT on hyper-representation

Figure 5: Test accuracy of different models on AUC maximization and hyper-representaion using AdamBO with $\beta = 0.1, \beta_{\text{sq}} = 0.001$ and different $\lambda$s. (a) a 2-layer RNN model on AUC maximization (data imbalanced ratio = 0.8); (b) a 2-layer Transformer model on AUC maximization (data imbalanced ratio = 0.9); (c) an 8-layer BERT model on hyper-representation.

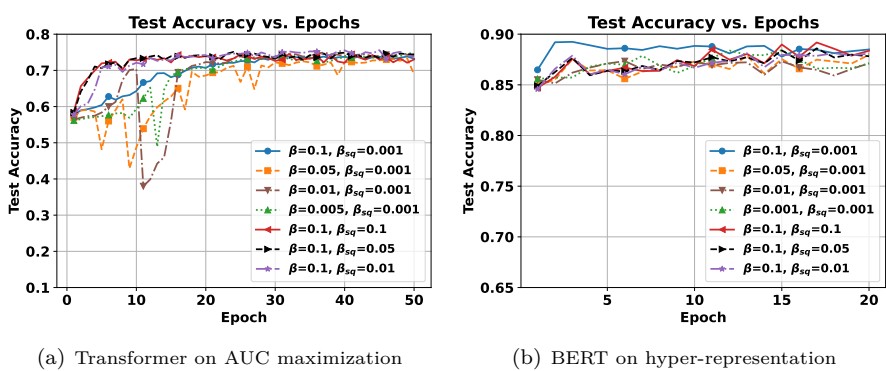

(a) Transformer on AUC maximization     (b) BERT on hyper-representation

Figure 6: Test accuracy of different models on AUC maximization and hyper-representaion using AdamBO with different $(\beta, \beta_{\text{sq}})$. (a) 2-layer Transformer model on AUC maximization (data imbalanced ratio = 0.9); (b) 8-layer BERT model on hyper-representation.

More ablation studies about the lower level update strategies and the approximation error of the lower level variable are deferred to Appendix G and Appendix H, respectively.

## 6 Conclusion

In this paper, we propose an Adam-type algorithm termed AdamBO for solving bilevel optimization problems under the unbounded smoothness setting. AdamBO is a single-loop algorithm with $\widetilde{O}(\epsilon^{-4})$ oracle complexity to find $\epsilon$-stationary points. We conduct experiments on meta-learning and deep AUC maximization for text classification using transformers. The experimental results demonstrate the superior performance of our proposed method. One limitation of our analysis is that the complexity bound of AdamBO depends on $O(\lambda^{-2})$, which can be large when $\lambda$ is small. However, our empirical sensitivity analysis indicates that AdamBO's performance remains largely unaffected by the choice of $\lambda$ within a reasonable range. In the future, we plan to improve the dependency on $\lambda$ in the complexity bound.

### Broader Impact Statement

This paper presents work whose goal is to advance the field of machine learning. There are many potential societal consequences of our work, none of which we feel must be specifically highlighted here.

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

## Contents

## A    Equivalent Update Rule of AdamBO (Algorithm 1)

In this section, we aim to provide a simplified version of the bias correction steps (lines 7-8) of Algorithm 1. Inspired by (Li et al., 2023a, Appendix C.1), we present an equivalent yet simpler update rule of Algorithm 1 in the following Proposition A.1. The detailed equivalent framework is also outlined in Algorithm 3.

**Proposition A.1.** *Let* $\alpha_t = \frac{\beta}{1-(1-\beta)^t}$ *and* $\alpha_t^{\mathrm{sq}} = \frac{\beta_{\mathrm{sq}}}{1-(1-\beta_{\mathrm{sq}})^t}$. *Then the update rule in Bi-Adam (Algorithm 1) is equivalent to that in Algorithm 3:*

$$
\begin{aligned}
y_{t+1} &= y_t - \gamma \nabla_y G(x_t, y_t; \zeta_t), \\
\hat{m}_t &= (1-\alpha_t)\hat{m}_{t-1} + \alpha_t \hat{\nabla}\phi(x_t, y_t; \bar{\xi}_t), \\
\hat{v}_t &= (1-\alpha_t^{\mathrm{sq}})\hat{v}_{t-1} + \alpha_t^{\mathrm{sq}}(\hat{\nabla}\phi(x_t, y_t; \bar{\xi}_t))^2, \\
x_{t+1} &= x_t - \frac{\eta}{\sqrt{\hat{v}_t} + \lambda} \odot \hat{m}_t,
\end{aligned}
\tag{7}
$$

*where initially we set* $\hat{m}_1 = \hat{\nabla}\phi(x_1, y_1; \bar{\xi}_1)$ *and* $\hat{v}_1 = (\hat{\nabla}\phi(x_1, y_1; \bar{\xi}_1))^2$. *There is no need to define* $\hat{m}_0$ *and* $\hat{v}_0$ *since* $1 - \alpha_1 = 1 - \alpha_1^{\mathrm{sq}} = 0$.

*Proof of Proposition A.1.* We follow the same proof as in (Li et al., 2023a, Proposition E.1), but replace the stochastic gradient $\nabla f(x_t, \xi_t)$ in (Li et al., 2023a) with the stochastic hypergradient estimator $\hat{\nabla}\phi(x_t, y_t; \bar{\xi}_t)$ in our setting. We still provide the proof here for completeness.

Let $Z_t = 1 - (1-\beta)^t$. Then we know that $\alpha_t = \beta/Z_t$ and $m_t = Z_t \hat{m}_t$. By line 6 of Algorithm 1 (the momentum update rule for $m_t$), we have

$$
Z_t \hat{m}_t = (1-\beta)Z_{t-1}\hat{m}_{t-1} + \beta\hat{\nabla}\phi(x_t, y_t; \bar{\xi}_t).
$$

Note that $Z_t$ satisfies the following property

$$
(1-\beta)Z_{t-1} = 1 - \beta - (1-\beta)^t = Z_t - \beta.
$$

Then we have

$$
\begin{aligned}
\hat{m}_t &= \frac{Z_t - \beta}{Z_t}\hat{m}_{t-1} + \frac{\beta}{Z_t}\hat{\nabla}\phi(x_t, y_t; \bar{\xi}_t) \\
&= (1-\alpha_t)\hat{m}_{t-1} + \alpha_t\hat{\nabla}\phi(x_t, y_t; \bar{\xi}_t).
\end{aligned}
$$

Next, we verify the initial condition. By Algorithm 1, since we set $m_0 = 0$, then we have $m_1 = \beta\hat{\nabla}\phi(x_1, y_1; \bar{\xi}_1)$. Therefore, we have $\hat{m}_1 = m_1/Z_1 = \hat{\nabla}\phi(x_1, y_1; \bar{\xi}_1)$ since $Z_1 = \beta$. Then the proof is completed by applying the same analysis on $v_t$ and $\hat{v}_t$. □

## B    Technical Lemmas

In this section, we present several useful algebraic facts (Appendix B.1), probabilistic lemmas (Appendix B.2), and auxiliary lemmas for bilevel optimization under the unbounded smoothness setting (Appendix B.3).

### B.1    Useful Algebraic Facts

In this section, we will frequently use $\alpha_t$ and $\alpha_t^{\mathrm{sq}}$, so we restate their definitions here for the reader's convenience:

$$
\alpha_t = \frac{\beta}{1-(1-\beta)^t} \qquad \text{and} \qquad \alpha_t^{\mathrm{sq}} = \frac{\beta_{\mathrm{sq}}}{1-(1-\beta_{\mathrm{sq}})^t}.
\tag{8}
$$

The following two lemmas, i.e., Lemmas B.1 and B.2, are useful for bounding the norm of the difference between Neumann series approximation matrices in Appendix B.3.

---

**Algorithm 2** SGD

---

1: **Input:** $x, y_0, \gamma, T_0$                                                      # $\texttt{SGD}(x, y_0, \gamma, T_0)$
2: **Initialize** $y_0^{\text{init}} = y_0$
3: **for** $t = 0, 1, \ldots, T_0 - 1$ **do**
4:      Sample $\pi_t$ from distribution $\mathcal{D}_g$
5:      $y_{t+1}^{\text{init}} = y_t^{\text{init}} - \gamma \nabla_y G(x, y_t^{\text{init}}; \pi_t)$
6: **end for**

---

**Algorithm 3** ADAMBO (Equivalent update rule of Algorithm 1)

---

1: **Input:** $\beta, \beta_{\text{sq}}, \eta, \gamma, \lambda, T_0, T, x_1, y_0$
2: **Initialize** $y_1 = \texttt{SGD}(x_1, y_0, \gamma, T_0)$, $\hat{m}_1 = \hat{\nabla}\phi(x_1, y_1; \bar{\xi}_1)$ and $\hat{v}_1 = (\hat{\nabla}\phi(x_1, y_1; \bar{\xi}_1))^2$
3: **for** $t = 1, \ldots, T$ **do**
4:      $\alpha_t = \frac{\beta}{1-(1-\beta)^t}, \alpha_t^{\text{sq}} = \frac{\beta_{\text{sq}}}{1-(1-\beta_{\text{sq}})^t}$
5:      Draw new samples and perform the following updates
6:      $y_{t+1} = y_t - \gamma \nabla_y G(x_t, y_t; \zeta_t)$
7:      $\hat{m}_t = (1 - \alpha_t)\hat{m}_{t-1} + \alpha_t \hat{\nabla}\phi(x_t, y_t; \bar{\xi}_t)$
8:      $\hat{v}_t = (1 - \alpha_t^{\text{sq}})\hat{v}_{t-1} + \alpha_t^{\text{sq}}(\hat{\nabla}\phi(x_t, y_t; \bar{\xi}_t))^2$
9:      $x_{t+1} = x_t - \frac{\eta}{\sqrt{\hat{v}_t} + \lambda} \odot \hat{m}_t$
10: **end for**

---

**Lemma B.1.** *For any matrix sequences $\{A_i\}_{i=1}^k$ and $\{B_i\}_{i=1}^k$ (where $k \geq 1$), it holds that*

$$\left\| \prod_{i=1}^k A_i - \prod_{i=1}^k B_i \right\| = \sum_{i=1}^k \|B_1\| \cdots \|B_{i-1}\| \|A_i - B_i\| \|A_{i+1}\| \cdots \|A_k\|,$$

*where we use the convention $A_{k+1} = B_0 = I$.*

*Proof of Lemma B.1.* It is easy to check that

$$\prod_{i=1}^k A_i - \prod_{i=1}^k B_i = A_1 \cdots A_k - B_1 \cdots B_k$$

$$= (A_1 - B_1)A_2 \cdots A_k + B_1(A_2 - B_2)A_3 \cdots A_k + \cdots + B_1 \cdots B_{k-1}(A_k - B_k)$$

$$= \sum_{i=1}^k B_1 \cdots B_{i-1}(A_i - B_i)A_{i+1} \cdots A_k,$$

where we set $A_{k+1} = B_0 = I$ in the last equality. The result follows by noting that the operator norm is submultiplicative. $\square$

**Lemma B.2.** *For any $Q \geq 1$ and $a \in (0, 1)$, we have*

$$\sum_{q=0}^{Q-1} q \cdot a^{q-1} \leq \frac{1}{(1-a)^2}.$$

*Proof of Lemma B.2.* We obtain the result by simple calculation:

$$\sum_{q=0}^{Q-1} q \cdot a^{q-1} = \frac{1 - Qa^{Q-1} + (Q-1)a^Q}{(1-a)^2} \leq \frac{1 - Qa^{Q-1} + (Q-1)a^{Q-1}}{(1-a)^2}$$

$$= \frac{1 - a^{Q-1}}{(1-a)^2} \leq \frac{1}{(1-a)^2}.$$

$\square$

The next four lemmas, Lemmas B.3 to B.6, are useful for controlling the lower-level estimation error and for proving the randomness decoupling lemma (i.e., Lemma 4.2) in Appendix C.

**Lemma B.3.** *For any $t \geq 1$, define $\{d_{t,j}\}_{j=0}^t$ as the following:*

$$d_{t,j} = \begin{cases} \prod_{i=1}^t (1 - \alpha_i), & j = 0 \\ \alpha_j \prod_{i=j+1}^t (1 - \alpha_i), & 1 \leq j \leq t - 1 \\ \alpha_t, & j = t. \end{cases} \tag{9}$$

*Then $\{d_{t,j}\}_{j=0}^t$ has the following properties:*

- *For $j = 0$, $d_{t,j} = 0$.*

- *For $1 \leq j \leq t$, $d_{t,j} = \alpha_t(1 - \beta)^{t-j}$.*

- *$\sum_{j=0}^t d_{t,j} = \sum_{j=1}^t d_{t,j} = 1$.*

*Proof of Lemma B.3.* Recall the definition of $\alpha_t$ in Algorithm 3, we have

$$\alpha_t = \frac{\beta}{1 - (1 - \beta)^t} \qquad \text{and} \qquad 1 - \alpha_t = \frac{1 - (1 - \beta)^{t-1}}{1 - (1 - \beta)^t}(1 - \beta).$$

It is obvious to see $\alpha_1 = 1$, then for $j = 0$ we have

$$d_{t,0} = \prod_{i=1}^t (1 - \alpha_i) = (1 - \alpha_t) \cdots (1 - \alpha_1) = 0.$$

For $1 \leq j \leq t - 1$ we have

$$d_{t,j} = \alpha_j \prod_{i=j+1}^t (1 - \alpha_i) = \frac{\beta}{1 - (1 - \beta)^j} \prod_{i=j+1}^t \frac{1 - (1 - \beta)^{i-1}}{1 - (1 - \beta)^i}(1 - \beta) = \alpha_t(1 - \beta)^{t-j}.$$

For $j = t$ we have

$$d_{t,t} = \alpha_t = \frac{\beta}{1 - (1 - \beta)^t} = \alpha_t(1 - \beta)^{t-t}.$$

For the last result of the lemma, we have

$$\sum_{j=0}^t d_{t,j} = \sum_{j=1}^t d_{t,j} = \sum_{j=1}^t \alpha_t(1 - \beta)^{t-j} = \frac{1 - (1 - \beta)^t}{\beta}\alpha_t = 1,$$

where we use $d_{t,0} = 0$ in the first equality. □

**Lemma B.4.** *For any $x \in (0, 1]$, we have*

$$1 - \frac{1}{x} \leq \ln x \leq x - 1.$$

*Consequently, for any $\beta \in [0, 1)$ we have*

$$-\frac{\beta}{1 - \beta} \leq \ln(1 - \beta) \leq -\beta \qquad \text{and} \qquad \beta \leq -\ln(1 - \beta) \leq \frac{\beta}{1 - \beta}.$$

*Proof of Lemma B.4.* This is a well-known logarithm inequality, so we omit the proof here. □

**Lemma B.5.** *For any $t \geq 1$, we have*

$$t\alpha_t(1 - \beta)^{t-1} \leq 1.$$

*Proof of Lemma B.5.* By definition of $\alpha_t$, we have

$$t\alpha_t(1-\beta)^{t-1} = \frac{\beta t(1-\beta)^{t-1}}{1-(1-\beta)^t}.$$

Let $f : \mathbb{R} \to \mathbb{R}$ be

$$f(t) = \frac{\beta t(1-\beta)^{t-1}}{1-(1-\beta)^t}.$$

Then we have

$$f'(t) = \frac{\beta(1-\beta)^{t-1}}{(1-(1-\beta)^t)^2}(1-(1-\beta)^t + t\ln(1-\beta)).$$

Let $g : \mathbb{R} \to \mathbb{R}$ be

$$g(t) = 1-(1-\beta)^t + t\ln(1-\beta).$$

Then we have

$$g'(t) = (1-(1-\beta)^t)\ln(1-\beta) \le 0.$$

Note that Lemma B.4 gives $g(1) = \beta + \ln(1-\beta) \le 0$, then for any $t \ge 1$ we have $g(t) \le g(1) \le 0$, and

$$f'(t) = \frac{\beta(1-\beta)^{t-1}}{(1-(1-\beta)^t)^2}g(t) \le 0.$$

Therefore, for any $t \ge 1$ we conclude that

$$t\alpha_t(1-\beta)^{t-1} = f(t) \le f(1) = 1.$$

$\square$

**Lemma B.6.** *For any $t \ge 1$ and $0 < \beta \le 1/2$, we have*

$$\sum_{i=1}^{t}(1-\beta)^{t-i}\alpha_i \le 32 + 16\ln\frac{1}{\beta}.$$

*Proof of Lemma B.6.* We split the summation as the following:

$$\sum_{i=1}^{t}(1-\beta)^{t-i}\alpha_i = \beta\sum_{i=1}^{t}\frac{(1-\beta)^{t-i}}{1-(1-\beta)^i} = \beta(1-\beta)^t\sum_{i=1}^{t}\frac{(1-\beta)^{-i}}{1-(1-\beta)^i}$$

$$= \beta(1-\beta)^t\left(\sum_{1\le i<1/\beta}\frac{(1-\beta)^{-i}}{1-(1-\beta)^i} + \sum_{1/\beta\le i\le t}\frac{(1-\beta)^{-i}}{1-(1-\beta)^i}\right).$$

Note that when $i < 1/\beta$, we have

$$(1-\beta)^i \le 1 - \frac{1}{2}\beta i \implies 1-(1-\beta)^i \ge \frac{1}{2}\beta i \implies \frac{1}{1-(1-\beta)^i} \le \frac{2}{\beta i},$$

and by Lemma B.4 and $\beta \le 1/2$ we know that

$$(1-\beta)^{-i} = \exp(-i\ln(1-\beta)) \le \exp\left(\frac{i\beta}{1-\beta}\right) \le \exp\left(\frac{1}{1-\beta}\right) \le e^2.$$

Then for the first part of the summation we have

$$\sum_{1\le i<1/\beta}\frac{(1-\beta)^{-i}}{1-(1-\beta)^i} \le \frac{2e^2}{\beta}\sum_{1\le i<1/\beta}\frac{1}{i} \le \frac{2e^2}{\beta}\left(1 + \ln\frac{1}{\beta}\right). \tag{10}$$

Also note that when $i \geq 1/\beta$, we have

$$(1-\beta)^i \leq \frac{1}{e} \quad \implies \quad 1 - (1-\beta)^i \geq 1 - \frac{1}{e} \quad \implies \quad \frac{1}{1-(1-\beta)^i} \leq \frac{e}{e-1}.$$

Then for the second part of the summation we have

$$\sum_{1/\beta \leq i \leq t} \frac{(1-\beta)^{-i}}{1-(1-\beta)^i} \leq \frac{e}{e-1} \sum_{1/\beta \leq i \leq t} (1-\beta)^{-i} \leq \frac{e}{e-1} \sum_{1 \leq i \leq t} (1-\beta)^{-i} \leq \frac{e(1-\beta)^{-t}}{(e-1)\beta}. \tag{11}$$

Combining (10) and (11) we obtain that

$$
\begin{aligned}
\sum_{i=1}^{t}(1-\beta)^{t-i}\alpha_i &\leq \beta(1-\beta)^t \left( \sum_{1 \leq i < 1/\beta} \frac{(1-\beta)^{-i}}{1-(1-\beta)^i} + \sum_{1/\beta \leq i \leq t} \frac{(1-\beta)^{-i}}{1-(1-\beta)^i} \right) \\
&\leq \beta(1-\beta)^t \left( \frac{2e^2}{\beta}\left(1+\ln\frac{1}{\beta}\right) + \frac{e(1-\beta)^{-t}}{(e-1)\beta} \right) \\
&= 2e^2(1-\beta)^t\left(1+\ln\frac{1}{\beta}\right) + \frac{e}{e-1} \\
&\leq 2e^2\left(1+\ln\frac{1}{\beta}\right) + \frac{e}{e-1} \\
&\leq 32 + 16\ln\frac{1}{\beta}.
\end{aligned}
$$

$\square$

Finally, we provide a useful lemma regarding the time-dependent re-scaled momentum parameters in (7) and Algorithm 3 for upper-level analysis.

**Lemma B.7** ((Li et al., 2023a, Lemma C.3)). *Let $\alpha_t = \frac{\beta}{1-(1-\beta)^t}$, then for all $T \geq 2$, we have*

$$\sum_{t=2}^{T} \alpha_t^2 \leq 3(1+\beta^2 T).$$

## B.2 Probabilistic Lemmas

In this section, we provide a well-known probabilistic lemma without proof.

**Lemma B.8** (Optional Stopping Theorem). *Let $\{Z_t\}_{t \geq 1}$ be a martingale with respect to a filtration $\{\mathcal{F}_t\}_{t \geq 0}$. Let $\tau$ be a bounded stopping time with respect to the same filtration. Then we have $\mathbb{E}[Z_\tau] = \mathbb{E}[Z_0]$.*

## B.3 Auxiliary Lemmas for Bilevel Optimization

In this section, we provide several useful lemmas for bilevel optimization under the unbounded smoothness setting, including the properties of the objective function $\Phi$ (Appendix B.3.1), the Neumann series approximation error (Appendix B.3.2), and the hypergradient estimation error (Appendix B.3.3).

### B.3.1 Properties of the Objective Function

**Lemma B.9** ((Hao et al., 2024, Lemma 8)). *Under Assumption 3.2, we have*

*(I) $y^*(x)$ is $(l_{g,1}/\mu)$-Lipschitz continuous.*

*(II) $\|\nabla_x f(x, y^*(x))\| \leq \|\nabla\Phi(x)\| + l_{g,1}l_{f,0}/\mu$.*

**Lemma B.10** $((L_0, L_1)$-smoothness (Hao et al., 2024, Lemma 9)). *Under Assumption 3.2, for any $x, x' \in \mathbb{R}^{d_x}$ we have*

$$\|\nabla\Phi(x) - \nabla\Phi(x')\| \leq (L_0 + L_1\|\nabla\Phi(x')\|)\|x - x'\|$$

$$\text{if} \quad \|x - x'\| \leq r := \frac{1}{\sqrt{(1 + l_{g,1}^2/\mu^2)(L_{x,1}^2 + L_{y,1}^2)}}, \tag{12}$$

*where the $(L_0, L_1)$-smoothness constants $L_0$ and $L_1$ are defined as*

$$L_0 = \sqrt{1 + \frac{l_{g,1}^2}{\mu^2}}\left(L_{x,0} + L_{x,1}\frac{l_{g,1}l_{f,0}}{\mu} + \frac{l_{g,1}}{\mu}(L_{y,0} + L_{y,1}l_{f,0}) + l_{f,0}\frac{l_{g,1}l_{g,2} + \mu l_{g,2}}{\mu^2}\right),$$

$$L_1 = \sqrt{1 + \frac{l_{g,1}^2}{\mu^2}}L_{x,1}. \tag{13}$$

**Lemma B.11** (Descent Inequality (Hao et al., 2024, Lemma 10)). *Under Assumption 3.2, for any $x, x' \in \mathbb{R}^{d_x}$ we have*

$$\Phi(x) \leq \Phi(x') + \langle\nabla\Phi(x'), x - x'\rangle + \frac{L_0 + L_1\|\nabla\Phi(x')\|}{2}\|x - x'\|^2$$

$$\text{if} \quad \|x - x'\| \leq r = \frac{1}{\sqrt{(1 + l_{g,1}^2/\mu^2)(L_{x,1}^2 + L_{y,1}^2)}}.$$

### B.3.2 Neumann Series Approximation

Throughout the paper, for given $(x, y) \in \mathbb{R}^{d_x} \times \mathbb{R}^{d_y}$, we estimate the hypergradient $\nabla\Phi(x)$ using Neumann series approach and the following formulation:

$$\hat{\nabla}\phi(x, y; \bar{\xi}) = \nabla_x F(x, y; \xi) - \nabla_{xy}^2 G(x, y; \zeta^{(0)})\left[\frac{1}{l_{g,1}}\sum_{q=0}^{Q-1}\prod_{j=1}^{q}\left(I - \frac{\nabla_{yy}^2 G(x, y; \zeta^{(q,j)})}{l_{g,1}}\right)\right]\nabla_y F(x, y; \xi),$$

where the randomness $\bar{\xi}$ is defined as

$$\bar{\xi} := \{\xi, \zeta^{(0)}, \bar{\zeta}^{(0)}, \ldots, \bar{\zeta}^{(Q-1)}\}, \quad \text{with} \quad \bar{\zeta}^{(q)} := \{\zeta^{(q,1)}, \ldots, \zeta^{(q,q)}\}.$$

For simplicity, denote $P$ as the Neumann series approximation matrix for the Hessian inverse, then $P$ and $\mathbb{E}_{\bar{\xi}}[P]$ can be written as:

$$P = \frac{1}{l_{g,1}}\sum_{q=0}^{Q-1}\prod_{j=1}^{q}\left(I - \frac{\nabla_{yy}^2 G(x, y; \zeta^{(q,j)})}{l_{g,1}}\right) \quad \text{and} \quad \mathbb{E}_{\bar{\xi}}[P] = \frac{1}{l_{g,1}}\sum_{q=0}^{Q-1}\left(I - \frac{\nabla_{yy}^2 g(x, y)}{l_{g,1}}\right)^q. \tag{14}$$

Hence the simplified version of the hypergradient estimator and its expectation are

$$\hat{\nabla}\phi(x, y; \bar{\xi}) = \nabla_x F(x, y; \xi) - \nabla_{xy}^2 G(x, y; \zeta^{(0)})P\nabla_y F(x, y; \xi),$$

$$\mathbb{E}_{\bar{\xi}}[\hat{\nabla}\phi(x, y; \bar{\xi})] = \nabla_x f(x, y) - \nabla_{xy}^2 g(x, y)\mathbb{E}_{\bar{\xi}}[P]\nabla_y f(x, y). \tag{15}$$

Also, we define $\bar{\nabla}f(x, y)$ as

$$\bar{\nabla}f(x, y) = \nabla_x f(x, y) - \nabla_{xy}^2 g(x, y)[\nabla_{yy}^2 g(x, y)]^{-1}\nabla_y f(x, y),$$

which is useful for the following analysis.

The following lemma bounds the norm of the Neumann series approximation matrix $P$ and characterizes the approximation error for the Hessian inverse in expectation.

**Lemma B.12.** *Under Assumptions 3.2 to 3.4, we have*

$$\|\mathbb{E}_{\bar{\xi}}[P]\| \le \|P\| \le \frac{1}{\mu} \qquad and \qquad \|\mathbb{E}_{\bar{\xi}}[P] - [\nabla_{yy}^2 g(x,y)]^{-1}\| \le \frac{1}{\mu}\left(1 - \frac{\mu}{l_{g,1}}\right)^Q.$$

*Proof of Lemma B.12.* We follow the similar proof as in (Ghadimi & Wang, 2018, Lemma 3.2). By Assumption 3.4 and definition of $P$ in (14), for any $Q \ge 1$ we have

$$\|\mathbb{E}_{\bar{\xi}}[P]\| \le \|P\| = \left\|\frac{1}{l_{g,1}}\sum_{q=0}^{Q-1}\prod_{j=1}^{q}\left(I - \frac{\nabla_{yy}^2 G(x,y;\zeta^{(q,j)})}{l_{g,1}}\right)\right\| \le \frac{1}{l_{g,1}}\sum_{q=0}^{Q-1}\left(1 - \frac{\mu}{l_{g,1}}\right)^q \le \frac{1}{\mu}.$$

As for the second result, we have

$$\|\mathbb{E}_{\bar{\xi}}[P] - [\nabla_{yy}^2 g(x,y)]^{-1}\| \le \frac{1}{l_{g,1}}\left\|\sum_{q=Q}^{\infty}\left(I - \frac{\nabla_{yy}^2 G(x,y)}{l_{g,1}}\right)^q\right\|$$

$$\le \frac{1}{l_{g,1}}\sum_{q=Q}^{\infty}\left\|\left(I - \frac{\nabla_{yy}^2 G(x,y)}{l_{g,1}}\right)\right\|^q \le \frac{1}{\mu}\left(1 - \frac{\mu}{l_{g,1}}\right)^Q.$$

$\square$

### B.3.3 Hypergradient Estimation Error

**Lemma B.13.** *Under Assumptions 3.2 to 3.4, if $\|y - y^*(x)\| \le r$, we have*

$$\|\hat{\nabla}\phi(x,y;\bar{\xi}) - \mathbb{E}_{\bar{\xi}}[\hat{\nabla}\phi(x,y;\bar{\xi})]\|$$
$$\le \frac{\mu + 3l_{g,1} + \sigma_{g,2}}{\mu}\sigma_f + \frac{2l_{g,1} + \sigma_{g,2}}{\mu}l_{f,0} + \frac{2l_{g,1} + \sigma_{g,2}}{\mu}(L_{y,0} + L_{y,1}l_{f,0})\|y - y^*(x)\|.$$

*Proof of Lemma B.13.* We will use a short hand $y^* = y^*(x)$. By triangle inequality, we have

$$\|\hat{\nabla}\phi(x,y;\bar{\xi}) - \mathbb{E}_{\bar{\xi}}[\hat{\nabla}\phi(x,y;\bar{\xi})]\|$$
$$= \|(\nabla_x F(x,y;\xi) - \nabla_{xy}^2 G(x,y;\zeta^{(0)})P\nabla_y F(x,y;\xi)) - (\nabla_x f(x,y) - \nabla_{xy}^2 g(x,y)\mathbb{E}_{\bar{\xi}}[P]\nabla_y f(x,y))\|$$
$$\le \underbrace{\|\nabla_x F(x,y;\xi) - \nabla_x f(x,y)\|}_{(A_1)} + \underbrace{\|(\nabla_{xy}^2 G(x,y;\zeta^{(0)}) - \nabla_{xy}^2 g(x,y))P\nabla_y F(x,y;\xi)\|}_{(A_2)}$$
$$+ \underbrace{\|\nabla_{xy}^2 g(x,y)(P - \mathbb{E}_{\bar{\xi}}[P])\nabla_y F(x,y;\xi)\|}_{(A_3)} + \underbrace{\|\nabla_{xy}^2 g(x,y)\mathbb{E}_{\bar{\xi}}[P](\nabla_y F(x,y;\xi) - \nabla_y f(x,y))\|}_{(A_4)}$$

**Bounding** $(A_1)$**.** By Assumption 3.3, we have

$$(A_1) = \|\nabla_x F(x,y;\xi) - \nabla_x f(x,y)\| \le \sigma_f.$$

**Bounding** $(A_2)$**.** By Assumptions 3.2 and 3.3 and Lemma B.12, we have

$$(A_2) = \|(\nabla_{xy}^2 G(x,y;\zeta^{(0)}) - \nabla_{xy}^2 g(x,y))P\nabla_y F(x,y;\xi)\|$$
$$\le \|\nabla_{xy}^2 G(x,y;\zeta^{(0)}) - \nabla_{xy}^2 g(x,y)\|\|P\|\|\nabla_y F(x,y;\xi)\|$$
$$\le \frac{\sigma_{g,2}}{\mu}(\|\nabla_y F(x,y;\xi) - \nabla_y f(x,y)\| + \|\nabla_y f(x,y) - \nabla_y f(x,y^*)\| + \|\nabla_y f(x,y^*)\|)$$
$$\le \frac{\sigma_{g,2}}{\mu}(\sigma_f + (L_{y,0} + L_{y,1}l_{f,0})\|y - y^*\| + l_{f,0})$$
$$= \frac{\sigma_{g,2}}{\mu}(\sigma_f + l_{f,0}) + \frac{\sigma_{g,2}}{\mu}(L_{y,0} + L_{y,1}l_{f,0})\|y - y^*\|.$$

**Bounding** $(A_3)$. By Assumptions 3.2 and 3.3 and Lemma B.12, we have

$$\begin{aligned}
(A_3) &= \|\nabla_{xy}^2 g(x,y)(P - \mathbb{E}_{\bar{\xi}}[P])\nabla_y F(x,y;\xi)\| \\
&\leq \|\nabla_{xy}^2 g(x,y)\|\|(P - \mathbb{E}_{\bar{\xi}}[P])\|\|\nabla_y F(x,y;\xi)\| \\
&\leq \frac{2l_{g,1}}{\mu}(\sigma_f + (L_{y,0} + L_{y,1}l_{f,0})\|y - y^*\| + l_{f,0}) \\
&= \frac{2l_{g,1}}{\mu}(\sigma_f + l_{f,0}) + \frac{2l_{g,1}}{\mu}(L_{y,0} + L_{y,1}l_{f,0})\|y - y^*\|,
\end{aligned}$$

where the second inequality uses the same step (the third inequality above) as in bounding $(A_2)$.

**Bounding** $(A_4)$. By Assumptions 3.2 and 3.3 and Lemma B.12, we have

$$(A_4) = \|\nabla_{xy}^2 g(x,y)\mathbb{E}_{\bar{\xi}}[P](\nabla_y F(x,y;\xi) - \nabla_y f(x,y))\| \leq \frac{l_{g,1}}{\mu}\sigma_f.$$

Then we obtain the final bound

$$\begin{aligned}
\|\hat{\nabla}\phi(x,y;\bar{\xi}) - \mathbb{E}_{\bar{\xi}}[\hat{\nabla}\phi(x,y;\bar{\xi})]\| &\leq (A_1) + (A_2) + (A_3) + (A_4) \\
&\leq \frac{\mu + 3l_{g,1} + \sigma_{g,2}}{\mu}\sigma_f + \frac{2l_{g,1} + \sigma_{g,2}}{\mu}l_{f,0} + \frac{2l_{g,1} + \sigma_{g,2}}{\mu}(L_{y,0} + L_{y,1}l_{f,0})\|y - y^*\|.
\end{aligned}$$

$\square$

**Lemma B.14.** *Under Assumptions 3.2 to 3.4, if $\|y - y^*(x)\| \leq r$, we have*

$$\|\hat{\nabla}\phi(x,y;\bar{\xi}) - \nabla\Phi(x)\| \leq C_{\phi,0} + (C_{\phi,1} + L_1\|\nabla\Phi(x)\|)\|y - y^*(x)\|,$$

*where $L_1$ is defined in (13) and constants $C_{\phi,0}$ and $C_{\phi,1}$ are defined as*

$$\begin{aligned}
C_{\phi,0} &= \frac{\mu + 3l_{g,1} + \sigma_{g,2}}{\mu}\sigma_f + \frac{2l_{g,1} + \sigma_{g,2}}{\mu}l_{f,0} + \frac{l_{g,1}l_{f,0}}{\mu}, \\
C_{\phi,1} &= \frac{2l_{g,1} + \sigma_{g,2}}{\mu}(L_{y,0} + L_{y,1}l_{f,0}) + \frac{l_{g,1}}{\mu}(L_{y,0} + L_{y,1}l_{f,0}) + L_0.
\end{aligned} \tag{16}$$

*Proof of Lemma B.14.* We have the following decomposition:

$$\begin{aligned}
\|\hat{\nabla}\phi(x,y;\bar{\xi}) - \nabla\Phi(x)\| &\leq \|\hat{\nabla}\phi(x,y;\bar{\xi}) - \mathbb{E}_{\bar{\xi}}[\hat{\nabla}\phi(x,y;\bar{\xi})]\| \\
&\quad + \|\mathbb{E}_{\bar{\xi}}[\hat{\nabla}\phi(x,y;\bar{\xi})] - \bar{\nabla}f(x,y)\| + \|\bar{\nabla}f(x,y) - \nabla\Phi(x)\|,
\end{aligned}$$

For the first term, by Lemma B.13 we have

$$\begin{aligned}
&\|\hat{\nabla}\phi(x,y;\bar{\xi}) - \mathbb{E}_{\bar{\xi}}[\hat{\nabla}\phi(x,y;\bar{\xi})]\| \\
&\quad \leq \frac{\mu + 3l_{g,1} + \sigma_{g,2}}{\mu}\sigma_f + \frac{2l_{g,1} + \sigma_{g,2}}{\mu}l_{f,0} + \frac{2l_{g,1} + \sigma_{g,2}}{\mu}(L_{y,0} + L_{y,1}l_{f,0})\|y - y^*\|.
\end{aligned} \tag{17}$$

For the second term, by Assumption 3.2 and Lemma B.12 we have

$$\begin{aligned}
&\|\mathbb{E}_{\bar{\xi}}[\hat{\nabla}\phi(x,y;\bar{\xi})] - \bar{\nabla}f(x,y)\| \\
&\quad = \|(\nabla_x f(x,y) - \nabla_{xy}^2 g(x,y)\mathbb{E}_{\bar{\xi}}[P]\nabla_y f(x,y)) \\
&\qquad - (\nabla_x f(x,y) - \nabla_{xy}^2 g(x,y)[\nabla_{yy}^2 g(x,y)]^{-1}\nabla_y f(x,y))\| \\
&\quad = \|\nabla_{xy}^2 g(x,y)(\mathbb{E}_{\bar{\xi}}[P] - [\nabla_{yy}^2 g(x,y)]^{-1})\nabla_y f(x,y)\| \\
&\quad \leq \frac{l_{g,1}}{\mu}\left(1 - \frac{\mu}{l_{g,1}}\right)^Q (\|\nabla_y f(x,y) - \nabla_y f(x,y^*)\| + \|\nabla_y f(x,y)\|) \\
&\quad \leq \frac{l_{g,1}}{\mu}\left(1 - \frac{\mu}{l_{g,1}}\right)^Q ((L_{y,0} + L_{y,1}l_{f,0})\|y - y^*\| + l_{f,0}) \\
&\quad = \frac{l_{g,1}l_{f,0}}{\mu}\left(1 - \frac{\mu}{l_{g,1}}\right)^Q + \frac{l_{g,1}}{\mu}\left(1 - \frac{\mu}{l_{g,1}}\right)^Q (L_{y,0} + L_{y,1}l_{f,0})\|y - y^*\|.
\end{aligned} \tag{18}$$

For the third term, by Assumption 3.2 and Lemma B.9 we have

$$
\begin{aligned}
&\|\bar{\nabla} f(x, y) - \nabla \Phi(x)\| \\
&\leq \|\nabla_x f(x, y) - \nabla_x f(x, y^*)\| \\
&\quad + \|\nabla_{xy}^2 g(x, y)[\nabla_{yy}^2 g(x, y)]^{-1} \nabla_y f(x, y) - \nabla_{xy}^2 g(x, y^*)[\nabla_{yy}^2 g(x, y^*)]^{-1} \nabla_y f(x, y^*)\| \\
&\leq (L_{x,0} + L_{x,1} \|\nabla_x f(x, y^*)\|) \|y - y^*\| \\
&\quad + \|\nabla_{xy}^2 g(x, y)[\nabla_{yy}^2 g(x, y)]^{-1} \nabla_y f(x, y) - \nabla_{xy}^2 g(x, y^*)[\nabla_{yy}^2 g(x, y)]^{-1} \nabla_y f(x, y)\| \\
&\quad + \|\nabla_{xy}^2 g(x, y^*)[\nabla_{yy}^2 g(x, y)]^{-1} \nabla_y f(x, y) - \nabla_{xy}^2 g(x, y^*)[\nabla_{yy}^2 g(x, y^*)]^{-1} \nabla_y f(x, y)\| \\
&\quad + \|\nabla_{xy}^2 g(x, y^*)[\nabla_{yy}^2 g(x, y^*)]^{-1} \nabla_y f(x, y) - \nabla_{xy}^2 g(x, y^*)[\nabla_{yy}^2 g(x, y^*)]^{-1} \nabla_y f(x, y^*)\| \quad (19) \\
&\leq \left( L_{x,0} + L_{x,1} \left( \frac{l_{g,1} l_{f,0}}{\mu} + \|\nabla \Phi(x)\| \right) \right) \|y - y^*\| \\
&\quad + \frac{l_{f,0}}{\mu} l_{g,2} \|y - y^*\| + \frac{l_{f,0} l_{g,1}}{\mu^2} l_{g,2} \|y - y^*\| + \frac{l_{g,1}}{\mu} (L_{y,0} + L_{y,1} \|\nabla_y f(x, y^*)\|) \|y - y^*\| \\
&= \left( L_{x,0} + L_{x,1} \frac{l_{g,1} l_{f,0}}{\mu} + \frac{l_{g,1}}{\mu} (L_{y,0} + L_{y,1} l_{f,0}) + l_{f,0} \frac{\mu l_{g,2} + l_{g,1} l_{g,2}}{\mu^2} + L_{x,1} \|\nabla \Phi(x)\| \right) \|y - y^*\| \\
&\leq (L_0 + L_1 \|\nabla \Phi(x)\|) \|y - y^*\|,
\end{aligned}
$$

where the last inequality uses the definition of $L_0$ and $L_1$ as in (13). Summing up $(17) + (18) + (19)$ gives the final bound

$$
\begin{aligned}
\|\hat{\nabla} \phi(x, y; \bar{\xi}) - \nabla \Phi(x)\| &\leq \|\hat{\nabla} \phi(x, y; \bar{\xi}) - \mathbb{E}_{\bar{\xi}}[\hat{\nabla} \phi(x, y; \bar{\xi})]\| \\
&\quad + \|\mathbb{E}_{\bar{\xi}}[\hat{\nabla} \phi(x, y; \bar{\xi})] - \bar{\nabla} f(x, y)\| + \|\bar{\nabla} f(x, y) - \nabla \Phi(x)\| \\
&\leq \frac{\mu + 3 l_{g,1} + \sigma_{g,2}}{\mu} \sigma_f + \frac{2 l_{g,1} + \sigma_{g,2}}{\mu} l_{f,0} + \frac{l_{g,1} l_{f,0}}{\mu} \left( 1 - \frac{\mu}{l_{g,1}} \right)^Q \\
&\quad + \left( \frac{2 l_{g,1} + \sigma_{g,2}}{\mu} (L_{y,0} + L_{y,1} l_{f,0}) + \frac{l_{g,1}}{\mu} \left( 1 - \frac{\mu}{l_{g,1}} \right)^Q (L_{y,0} + L_{y,1} l_{f,0}) + L_0 + L_1 \|\nabla \Phi(x)\| \right) \|y - y^*\| \\
&\leq \frac{\mu + 3 l_{g,1} + \sigma_{g,2}}{\mu} \sigma_f + \frac{2 l_{g,1} + \sigma_{g,2}}{\mu} l_{f,0} + \frac{l_{g,1} l_{f,0}}{\mu} \\
&\quad + \left( \frac{2 l_{g,1} + \sigma_{g,2}}{\mu} (L_{y,0} + L_{y,1} l_{f,0}) + \frac{l_{g,1}}{\mu} (L_{y,0} + L_{y,1} l_{f,0}) + L_0 + L_1 \|\nabla \Phi(x)\| \right) \|y - y^*\| \\
&= C_{\phi,0} + (C_{\phi,1} + L_1 \|\nabla \Phi(x)\|) \|y - y^*\|,
\end{aligned}
$$

where the second and the third inequalities use $Q \geq 1$, and the last inequality is due to the definitions of $C_{\phi,0}$ and $C_{\phi,1}$ in (16). $\qquad \square$

### B.3.4 Other Useful Lemmas

**Lemma B.15.** *Under Assumptions 3.2 to 3.4, if $\|y - y^*(x)\| \leq r$, we have* [4]

$$
\|\hat{\nabla} \phi(x, y; \bar{\xi}) - \hat{\nabla} \phi(x, y^*(x); \bar{\xi})\| \leq (L_0 + L_1 \|\nabla \Phi(x)\|) \|y - y^*(x)\|;
$$

*if $\|x_1 - x_2\| \leq \mu r / (\mu + l_{g,1})$, we have*

$$
\|\mathbb{E}_{\bar{\xi}_1}[\hat{\nabla} \phi(x_1, y_1^*; \bar{\xi}_1)] - \mathbb{E}_{\bar{\xi}_2}[\hat{\nabla} \phi(x_2, y_2^*; \bar{\xi}_2)]\| \leq (L_0 + L_1 \|\nabla \Phi(x_1)\|) \|x_1 - x_2\|,
$$

*where $y_i^* = y^*(x_i)$ for $i = 1, 2$, and constants $L_0$ and $L_1$ are defined in (13).*

---

[4] Please note that $x_1$ and $x_2$ here are unrelated to Algorithm 1 and are deterministic.

*Proof of Lemma B.15.* We will use a short hand $y^* = y^*(x)$. Recall the definition of $\hat{\nabla}\phi(x,y;\bar{\xi})$ and $\hat{\nabla}\phi(x,y^*;\bar{\xi})$ in (15), we have

$$\hat{\nabla}\phi(x,y;\bar{\xi}) = \nabla_x F(x,y;\xi) - \nabla_{xy}^2 G(x,y;\zeta^{(0)})P\nabla_y F(x,y;\xi),$$
$$\hat{\nabla}\phi(x,y^*;\bar{\xi}) = \nabla_x F(x,y^*;\xi) - \nabla_{xy}^2 G(x,y^*;\zeta^{(0)})P^*\nabla_y F(x,y^*;\xi).$$

where similar to (14), we define the Neumann series approximation matrix $P^*$ as

$$P^* = \frac{1}{l_{g,1}} \sum_{q=0}^{Q-1} \prod_{j=1}^{q} \left( I - \frac{\nabla_{yy}^2 G(x,y^*;\zeta^{(q,j)})}{l_{g,1}} \right). \tag{20}$$

Then by triangle inequality we have

$$\|\hat{\nabla}\phi(x,y;\bar{\xi}) - \hat{\nabla}\phi(x,y^*;\bar{\xi})\|$$
$$\leq \|\nabla_x F(x,y;\xi) - \nabla_x F(x,y^*;\xi)\|$$
$$\quad + \|\nabla_{xy}^2 G(x,y;\zeta^{(0)})P\nabla_y F(x,y;\xi) - \nabla_{xy}^2 G(x,y^*;\zeta^{(0)})P^*\nabla_y F(x,y^*;\xi)\|$$
$$\leq \underbrace{\|\nabla_x F(x,y;\xi) - \nabla_x F(x,y^*;\xi)\|}_{(A_1)} + \underbrace{\|\nabla_{xy}^2 G(x,y;\zeta^{(0)})P(\nabla_y F(x,y;\xi) - \nabla_y F(x,y^*;\xi))\|}_{(A_2)}$$
$$\quad + \underbrace{\|\nabla_{xy}^2 G(x,y;\zeta^{(0)})(P-P^*)\nabla_y F(x,y^*;\xi)\|}_{(A_3)}$$
$$\quad + \underbrace{\|(\nabla_{xy}^2 G(x,y;\zeta^{(0)}) - \nabla_{xy}^2 G(x,y^*;\zeta^{(0)}))P^*\nabla_y F(x,y^*;\xi)\|}_{(A_4)}.$$

**Bounding $(A_1)$.** By Assumption 3.4 and Lemma B.9, we have

$$(A_1) = \|\nabla_x F(x,y;\xi) - \nabla_x F(x,y^*;\xi)\| \leq (L_{x,0} + L_{x,1}\|\nabla_x f(x,y^*)\|)\|y-y^*\|$$
$$\leq \left( L_{x,0} + L_{x,1}\left( \frac{l_{g,1}l_{f,0}}{\mu} + \|\nabla\Phi(x)\| \right) \right)\|y-y^*\|$$
$$= \left( L_{x,0} + \frac{L_{x,1}l_{g,1}l_{f,0}}{\mu} + L_{x,1}\|\nabla\Phi(x)\| \right)\|y-y^*\|.$$

**Bounding $(A_2)$.** By Assumption 3.4 and Lemma B.12, we have

$$(A_2) = \|\nabla_{xy}^2 G(x,y;\zeta^{(0)})P(\nabla_y F(x,y;\xi) - \nabla_y F(x,y^*;\xi))\|$$
$$= \|\nabla_{xy}^2 G(x,y;\zeta^{(0)})\|\|P\|\|\nabla_y F(x,y;\xi) - \nabla_y F(x,y^*;\xi)\|$$
$$\leq \frac{l_{g,1}}{\mu}(L_{y,0} + L_{y,1}\|\nabla_y f(x,y^*)\|)\|y-y^*\| \leq \frac{l_{g,1}}{\mu}(L_{y,0} + L_{y,1}l_{f,0})\|y-y^*\|.$$

**Bounding $(A_3)$.** We first apply Lemma B.1 to obtain

$$\left\| \prod_{j=1}^{q} \left( I - \frac{\nabla_{yy}^2 G(x,y;\zeta^{(q,j)})}{l_{g,1}} \right) - \prod_{j=1}^{q} \left( I - \frac{\nabla_{yy}^2 G(x,y^*;\zeta^{(q,j)})}{l_{g,1}} \right) \right\|$$
$$\leq \sum_{j=1}^{q} \left( 1 - \frac{\mu}{l_{g,1}} \right)^{q-1} \frac{l_{g,2}}{l_{g,1}}\|y-y^*\| = q\left( 1 - \frac{\mu}{l_{g,1}} \right)^{q-1} \frac{l_{g,2}}{l_{g,1}}\|y-y^*\|.$$

Hence we can write

$$\|P - P^*\| \leq \frac{1}{l_{g,1}} \sum_{q=0}^{Q-1} q\left( 1 - \frac{\mu}{l_{g,1}} \right)^{q-1} \frac{l_{g,2}}{l_{g,1}}\|y-y^*\| \leq \frac{\mu^2 l_{g,2}}{l_{g,1}^4}\|y-y^*\| \leq \frac{l_{g,2}}{\mu^2}\|y-y^*\|,$$

where the second inequality uses Lemma B.2 with $a = \mu/l_{g,1}$, and the last inequality is due to $\mu < l_{g,1}$. Then by Assumption 3.4 we have

$$(A_3) = \|\nabla^2_{xy}G(x, y; \zeta^{(0)})(P - P^*)\nabla_y F(x, y^*; \xi)\|$$

$$\leq \|\nabla^2_{xy}G(x, y; \zeta^{(0)})\|\|(P - P^*)\|\|\nabla_y F(x, y^*; \xi)\| \leq \frac{l_{g,1}l_{g,2}l_{f,0}}{\mu^2}\|y - y^*\|.$$

**Bounding** $(A_4)$. By Assumption 3.4 and Lemma B.12, we have

$$(A_4) = \|(\nabla^2_{xy}G(x, y; \zeta^{(0)}) - \nabla^2_{xy}G(x, y^*; \zeta^{(0)}))P^*\nabla_y F(x, y^*; \xi)\|$$

$$\leq \|\nabla^2_{xy}G(x, y; \zeta^{(0)}) - \nabla^2_{xy}G(x, y^*; \zeta^{(0)})\|\|P^*\|\|\nabla_y F(x, y^*; \xi)\| \leq \frac{l_{g,2}l_{f,0}}{\mu}\|y - y^*\|.$$

**Final Bound.** Summing up $(A_1) + (A_2) + (A_3) + (A_4)$ yields the final bound

$$\|\hat{\nabla}\phi(x, y; \bar{\xi}) - \hat{\nabla}\phi(x, y^*; \bar{\xi})\| \leq (A_1) + (A_2) + (A_3) + (A_4)$$

$$\leq \left(L_{x,0} + L_{x,1}\frac{l_{g,1}l_{f,0}}{\mu} + \frac{l_{g,1}}{\mu}(L_{y,0} + L_{y,1}l_{f,0}) + l_{f,0}\frac{l_{g,1}l_{g,2} + \mu l_{g,2}}{\mu^2} + L_{x,1}\|\nabla\Phi(x)\|\right)\|y - y^*\|$$

$$\leq (L_0 + L_1\|\nabla\Phi(x)\|)\|y - y^*\|,$$

where the last inequality uses the definitions of $L_0$ and $L_1$ as in (13).

For the second result, we follow a similar procedure as above and obtain:

$$\|\mathbb{E}_{\bar{\xi}_1}[\hat{\nabla}\phi(x_1, y_1^*; \bar{\xi}_1)] - \mathbb{E}_{\bar{\xi}_2}[\hat{\nabla}\phi(x_2, y_2^*; \bar{\xi}_2)]\| \leq (A_1) + (A_2) + (A_3) + (A_4)$$

$$\leq \sqrt{1 + \frac{l_{g,1}^2}{\mu^2}}\left(L_{x,0} + L_{x,1}\frac{l_{g,1}l_{f,0}}{\mu} + \frac{l_{g,1}}{\mu}(L_{y,0} + L_{y,1}l_{f,0}) + l_{f,0}\frac{l_{g,1}l_{g,2} + \mu l_{g,2}}{\mu^2} + L_{x,1}\|\nabla\Phi(x_1)\|\right)\|x_1 - x_2\|$$

$$= (L_0 + L_1\|\nabla\Phi(x_1)\|)\|x_1 - x_2\|,$$

where the last inequality uses the definitions of $L_0$ and $L_1$ as in (13). □

**Lemma B.16.** *Under Assumptions 3.2 to 3.4, we have*

$$\|\mathbb{E}_{\bar{\xi}}[\hat{\nabla}\phi(x, y^*(x); \bar{\xi})] - \nabla\Phi(x)\| \leq \frac{l_{g,1}l_{f,0}}{\mu}\left(1 - \frac{\mu}{l_{g,1}}\right)^Q.$$

*Proof of Lemma B.16.* We will use a short hand $y^* = y^*(x)$. By definition of $\hat{\nabla}\phi(x, y; \bar{\xi})$ in (15) and the hypergradient formulation, we have

$$\mathbb{E}_{\bar{\xi}}[\hat{\nabla}\phi(x, y^*; \bar{\xi})] = \nabla_x f(x, y^*) - \nabla^2_{xy}g(x, y^*)\mathbb{E}_{\bar{\xi}}[P]\nabla_y f(x, y^*),$$

$$\nabla\Phi(x) = \nabla_x f(x, y^*) - \nabla^2_{xy}g(x, y^*)[\nabla^2_{yy}g(x, y^*)]^{-1}\nabla_y f(x, y^*).$$

Then we obtain the conclusion by applying Assumption 3.2 and Lemma B.12:

$$\|\mathbb{E}_{\bar{\xi}}[\hat{\nabla}\phi(x, y; \bar{\xi})] - \nabla\Phi(x)\| = \|\nabla^2_{xy}g(x, y^*)(\mathbb{E}_{\bar{\xi}}[P] - [\nabla^2_{yy}g(x, y^*)]^{-1})\nabla_y f(x, y^*)\|$$

$$\leq \|\nabla^2_{xy}g(x, y^*)\|\|\mathbb{E}_{\bar{\xi}}[P] - [\nabla^2_{yy}g(x, y^*)]^{-1}\|\|\nabla_y f(x, y^*)\| \leq \frac{l_{g,1}l_{f,0}}{\mu}\left(1 - \frac{\mu}{l_{g,1}}\right)^Q.$$

□

## C  Proof of the Random Decoupling Lemma (Lemma 4.2)

### C.1  Recursive Control on Moment Generating Function

The following technical lemma on recursive control is crucial for establishing high probability guarantee for controlling the lower-level estimation error at anytime. We follow a similar argument as in (Cutler et al., 2023, Proposition 29) with a slight generalization.

**Proposition C.1** (Recursive control on MGF). *Consider scalar stochastic processes* $(V_t)$, $(D_t)$, $(X_t)$ *and* $(Y_t)$ *on a probability space with filtration* $(\mathcal{H}_t)$, *which are linked by the inequality*

$$V_{t+1} \leq \rho_t V_t + D_t \sqrt{V_t} + X_t + Y_t + \kappa_t \tag{21}$$

*for some deterministic constants* $\rho_t \in (-\infty, 1]$ *and* $\kappa_t \in \mathbb{R}$. *Suppose the following properties hold.*

- $V_t$ *and* $Y_t$ *are non-negative and* $\mathcal{H}_t$-*measurable.*

- $D_t$ *is mean-zero sub-Gaussian conditioned on* $\mathcal{H}_t$ *with deterministic parameter* $\sigma_t$:

$$\mathbb{E}[\exp(\theta D_t) \mid \mathcal{H}_t] \leq \exp(\theta^2 \sigma_t^2 / 2) \quad \textit{for all} \quad \theta \in \mathbb{R}.$$

- $X_t$ *is non-negative and sub-exponential conditioned on* $\mathcal{H}_t$ *with deterministic parameter* $\nu_t$:

$$\mathbb{E}[\exp(\theta X_t) \mid \mathcal{H}_t] \leq \exp(\theta \nu_t) \quad \textit{for all} \quad 0 \leq \theta \leq 1/\nu_t.$$

*Then the estimate*

$$\mathbb{E}[\exp(\theta V_{t+1})] \leq \exp(\theta(\nu_t + \kappa_t)) \mathbb{E}[\exp(\theta((1 + \rho_t) V_t / 2 + Y_t))]$$

*holds for any* $\theta$ *satisfying* $0 \leq \theta \leq \min\left\{\frac{1 - \rho_t}{2\sigma_t^2}, \frac{1}{2\nu_t}\right\}$.

*Proof of Proposition C.1.* For any index $t \geq 0$ and any scalar $\theta \geq 0$, the law of total expectation implies

$$\mathbb{E}[\exp(\theta V_{t+1})] \leq \mathbb{E}\left[\exp\left(\theta\left(\rho_t V_t + D_t \sqrt{V_t} + X_t + Y_t + \kappa_t\right)\right)\right]$$
$$= \exp(\theta \kappa_t) \mathbb{E}\left[\exp(\theta(\rho_t V_t + Y_t)) \mathbb{E}\left[\exp(\theta D_t \sqrt{V_t}) \exp(\theta X_t) \mid \mathcal{H}_t\right]\right].$$

Hölder's inequality in turn yields

$$\mathbb{E}\left[\exp(\theta D_t \sqrt{V_t}) \exp(\theta X_t) \mid \mathcal{H}_t\right] \leq \sqrt{\mathbb{E}\left[\exp(2\theta D_t \sqrt{V_t}) \mid \mathcal{H}_t\right] \cdot \mathbb{E}\left[\exp(2\theta X_t) \mid \mathcal{H}_t\right]}$$
$$\leq \sqrt{\exp(2\theta^2 \sigma_t^2 V_t) \exp(2\theta \nu_t)}$$
$$= \exp(\theta^2 \sigma_t^2 V_t) \exp(\theta \nu_t)$$

provided $0 \leq \theta \leq \frac{1}{2\nu_t}$. Therefore, if $\theta$ satisfies

$$0 \leq \theta \leq \min\left\{\frac{1 - \rho_t}{2\sigma_t^2}, \frac{1}{2\nu_t}\right\},$$

then the following estimate holds for all $t \geq 0$:

$$\mathbb{E}[\exp(\theta V_{t+1})] \leq \exp(\theta \kappa_t) \mathbb{E}\left[\exp(\theta(\rho_t V_t + Y_t)) \exp(\theta^2 \sigma_t^2 V_t) \exp(\theta \nu_t)\right]$$
$$= \exp(\theta(\nu_t + \kappa_t)) \mathbb{E}\left[\exp(\theta((\rho_t + \theta \sigma_t^2) V_t + Y_t))\right]$$
$$\leq \exp(\theta(\nu_t + \kappa_t)) \mathbb{E}\left[\exp(\theta((1 + \rho_t) V_t / 2 + Y_t))\right],$$

where the last inequality uses the given range of $\theta$. Thus the proof is completed. $\square$

## C.2 Proof of Lemma 4.2

In this section, we aim to provide a high-probability guarantee for the approximation error of the lower-level variable, namely $\|y_t - y_t^*\|$. Our main technical contribution is the any-sequence argument, which separates the randomness in the updates of the upper-level variable $x_t$ and the lower-level variable $y_t$. Specifically, for any given sequence $\{\tilde{x}_t\}$, we consider the following update rule for $\{\tilde{y}_t\}$ (which is the same as line 5 of Algorithm 1):

$$\tilde{y}_{t+1} = \tilde{y}_t - \gamma \nabla_y G(\tilde{x}_t, \tilde{y}_t; \tilde{\zeta}_t). \tag{22}$$

Before proceeding, we will first define (or restate) a few key concepts and useful notations.

**Filtration.** For any $t \geq 2$, define $\tilde{\mathcal{F}}_t^y$ as the filtration of the randomness used in updating $\tilde{y}_t$ before the $t$-th iteration:

$$\tilde{\mathcal{F}}_t^y = \sigma(\tilde{\zeta}_1, \ldots, \tilde{\zeta}_{t-1}), \tag{23}$$

where $\sigma(\cdot)$ denotes the $\sigma$-algebra generated by the random variables within the argument.

**Auxiliary Sequence.** We also introduce the following auxiliary sequence $\{\tilde{u}_t\}$ for our analysis:

$$\tilde{u}_t = (1 - \alpha_t)\tilde{u}_{t-1} + \alpha_t \hat{\nabla}\phi(\tilde{x}_t, \tilde{y}_t^*; \hat{\xi}_t) = \sum_{j=1}^t d_{t,j} \hat{\nabla}\phi(\tilde{x}_t, \tilde{y}_t^*; \hat{\xi}_t), \tag{24}$$

where the sequence $\{d_{t,j}\}_{j=1}^t$ is defined in (9) of Lemma B.3. Similar to (14), (15) and (20) in Appendix B, the hypergradient estimators $\hat{\nabla}\phi(\tilde{x}_t, \tilde{y}_t; \hat{\xi}_t)$ and $\hat{\nabla}\phi(\tilde{x}_t, \tilde{y}_t^*; \hat{\xi}_t)$ can be written as

$$\hat{\nabla}\phi(\tilde{x}_t, \tilde{y}_t; \hat{\xi}_t) = \nabla_x F(\tilde{x}_t, \tilde{y}_t; \tilde{\xi}_t) - \nabla_{xy}^2 G(\tilde{x}_t, \tilde{y}_t; \tilde{\zeta}_t^{(0)}) \tilde{P}_t \nabla_y F(\tilde{x}_t, \tilde{y}_t; \tilde{\xi}_t),$$

$$\hat{\nabla}\phi(\tilde{x}_t, \tilde{y}_t^*; \hat{\xi}_t) = \nabla_x F(\tilde{x}_t, \tilde{y}_t^*; \tilde{\xi}_t) - \nabla_{xy}^2 G(\tilde{x}_t, \tilde{y}_t^*; \tilde{\zeta}_t^{(0)}) \tilde{P}_t^* \nabla_y F(\tilde{x}_t, \tilde{y}_t^*; \tilde{\xi}_t),$$

where the randomness $\hat{\xi}_t$ is defined as

$$\hat{\xi}_t := \{\tilde{\xi}_t, \tilde{\zeta}_t^{(0)}, \tilde{\tilde{\zeta}}^{(0)}, \ldots, \tilde{\tilde{\zeta}}^{(Q-1)}\}, \qquad \text{where} \quad \tilde{\tilde{\zeta}}^{(q)} := \{\tilde{\zeta}^{(q,1)}, \ldots, \tilde{\zeta}^{(q,q)}\}; \tag{25}$$

and the Neumann series approximation matrices $\tilde{P}_t$ and $\tilde{P}_t^*$ are defined as

$$\tilde{P}_t = \frac{1}{l_{g,1}} \sum_{q=0}^{Q-1} \prod_{j=1}^q \left( I - \frac{\nabla_{yy}^2 G(\tilde{x}_t, \tilde{y}_t; \tilde{\zeta}_t^{(q,j)})}{l_{g,1}} \right) \quad \text{and} \quad \tilde{P}_t^* = \frac{1}{l_{g,1}} \sum_{q=0}^{Q-1} \prod_{j=1}^q \left( I - \frac{\nabla_{yy}^2 G(\tilde{x}_t, \tilde{y}_t^*; \tilde{\zeta}_t^{(q,j)})}{l_{g,1}} \right).$$

**Constants.** We define the following constants, which will be useful for analysis. Given any sequence $\{\tilde{x}_t\}$, denote $\tilde{G}_t$ and $\tilde{L}_t$ as

$$\tilde{G}_t := \max_{1 \leq k \leq t} \|\nabla\Phi(\tilde{x}_k)\|, \quad \tilde{L}_t := L_0 + L_1 \tilde{G}_t, \tag{26}$$

where constants $L_0$ and $L_1$ are defined in (13).

**Lemma C.2** (Distance recursion, (Cutler et al., 2023, Lemma 25)). *Suppose that Assumptions 3.2 and 3.3 hold. For any given sequence $\{\tilde{x}_t\}$, let $\{\tilde{y}_t\}$ be the iterates generated by the update rule (22) with constant learning rate $\gamma \leq 1/2l_{g,1}$. Then for any $t \geq 1$, we have the following recursion:*

$$\|\tilde{y}_{t+1} - \tilde{y}_{t+1}^*\|^2 \leq (1 - \mu\gamma)\|\tilde{y}_t - \tilde{y}_t^*\|^2 + 2\gamma\langle\tilde{\varepsilon}_t, \tilde{v}_t\rangle\|\tilde{y}_t - \tilde{y}_t^*\| + 2\gamma^2\|\tilde{\varepsilon}_t\|^2 + \frac{2}{\mu\gamma}D_t^2, \tag{27}$$

*where $\tilde{v}_t := \frac{\tilde{y}_t - \tilde{y}_t^*}{\|\tilde{y}_t - \tilde{y}_t^*\|}$ if $\tilde{y}_t$ is distinct from $\tilde{y}_t^*$ and zero otherwise, $\tilde{\varepsilon}_t = \nabla_y g(\tilde{x}_t, \tilde{y}_t) - \nabla_y G(\tilde{x}_t, \tilde{y}_t; \tilde{\zeta}_t)$ denotes the noise, and $D_t := \|\tilde{y}_t^* - \tilde{y}_{t+1}^*\|$ is the minimizer drift at time $t$.*

**Lemma C.3** (Restatement of Lemma 4.2). *Suppose that Assumptions 3.2 and 3.3 hold. Given any sequence $\{\tilde{x}_t\}$ and any randomness $\{\hat{\xi}_t\}$ (see (25) for definition) such that*

$$\|\tilde{x}_{t+1} - \tilde{x}_t\|^2 \leq \frac{2\eta^2}{\lambda^2}\left(\|\tilde{u}_t\|^2 + \tilde{L}_t^2 \sum_{j=1}^{t} d_{t,j}\|\tilde{y}_j - \tilde{y}_j^*\|^2\right), \tag{28}$$

*where $\tilde{u}_t$, $\{d_{t,j}\}_{j=1}^{t}$ and $\tilde{L}_t$ are defined in (24), (9) and (26), respectively. Let $\{\tilde{y}_t\}$ be the iterates generated by the update rule (22) with constant learning rate $\gamma \leq 1/2l_{g,1}$, and choose $\gamma = 2\beta/\mu$. Then for any given $\delta \in (0,1)$ and all $t \geq 1$, the following estimate holds with probability at least $1 - \delta$ over the randomness in $\tilde{\mathcal{F}}_{T+1}^y$:*

$$\|\tilde{y}_t - \tilde{y}_t^*\|^2 \leq \left(\left(1 - \frac{\mu\gamma}{2}\right)^{t-1} + \frac{4\eta^2 l_{g,1}^2}{\lambda^2\mu^3\gamma}\sum_{i=1}^{t-1}\left(1 - \frac{\mu\gamma}{2}\right)^{t-1-i}\tilde{L}_i^2\right)\|\tilde{y}_1 - \tilde{y}_1^*\|^2$$

$$+ \left(\frac{8\gamma}{\mu}\ln\frac{eT}{\delta} + \frac{16\eta^2 l_{g,1}^2}{\lambda^2\mu^4}\sum_{i=1}^{t-1}\left(1 - \frac{\mu\gamma}{2}\right)^{t-1-i}\tilde{L}_i^2\right)\sigma_{g,1}^2 \tag{29}$$

$$+ \frac{4\eta^2 l_{g,1}^2}{\lambda^2\mu^3\gamma}\sum_{i=1}^{t-1}\left(1 - \frac{\mu\gamma}{2}\right)^{t-1-i}\|\tilde{u}_i\|^2 + \frac{64\eta^4 l_{g,1}^4}{\lambda^4\mu^8\gamma^4}\sum_{i=1}^{t-1}\left(1 - \frac{\mu\gamma}{2}\right)^{t-1-i}\alpha_i\tilde{L}_i^2\|\tilde{u}_i\|^2.$$

*Proof of Lemma C.3.* By Lemma C.2 and Lemma B.9, we have

$$\|\tilde{y}_{t+1} - \tilde{y}_{t+1}^*\|^2 \leq (1 - \mu\gamma)\|\tilde{y}_t - \tilde{y}_t^*\|^2 + 2\gamma\langle\tilde{\varepsilon}_t, \tilde{v}_t\rangle\|\tilde{y}_t - \tilde{y}_t^*\| + 2\gamma^2\|\tilde{\varepsilon}_t\|^2 + \frac{2}{\mu\gamma}D_t^2$$

$$\leq (1 - \mu\gamma)\|\tilde{y}_t - \tilde{y}_t^*\|^2 + 2\gamma\langle\tilde{\varepsilon}_t, \tilde{v}_t\rangle\|\tilde{y}_t - \tilde{y}_t^*\| + 2\gamma^2\|\tilde{\varepsilon}_t\|^2 + \frac{2l_{g,1}^2}{\mu^3\gamma}\|\tilde{x}_{t+1} - \tilde{x}_t\|^2$$

$$\leq (1 - \mu\gamma)\|\tilde{y}_t - \tilde{y}_t^*\|^2 + 2\gamma\langle\tilde{\varepsilon}_t, \tilde{v}_t\rangle\|\tilde{y}_t - \tilde{y}_t^*\| + 2\gamma^2\|\tilde{\varepsilon}_t\|^2 \tag{30}$$

$$+ \frac{4\eta^2 l_{g,1}^2}{\lambda^2\mu^3\gamma}\left(\|\tilde{u}_t\|^2 + \tilde{L}_t^2 \sum_{j=1}^{t} d_{t,j}\|\tilde{y}_j - \tilde{y}_j^*\|^2\right),$$

where the last inequality uses (28). Note that under Assumption 3.3, there exists an absolute constant $c \geq 1$ such that for all $t \geq 1$, $\|\tilde{\varepsilon}_t\|^2$ is sub-exponential conditioned on $\tilde{\mathcal{F}}_t^y$ with parameter $c\sigma_{g,1}^2$, and $\tilde{\varepsilon}_t$ is mean-zero sub-Gaussian conditioned on $\tilde{\mathcal{F}}_t^y$ with parameter $c\sigma_{g,1}$ (Cutler et al., 2023, Theorem 30). For simplicity we set $c = 1$ here. Thus $\langle\tilde{\varepsilon}_t, u_t\rangle$ is mean-zero sub-Gaussian conditioned on $\tilde{\mathcal{F}}_t^y$ with parameter $\sigma_{g,1}$. Hence, in light of (30), we apply Proposition C.1 with

$$\mathcal{H}_t = \tilde{\mathcal{F}}_t^y, \quad V_t = \|\tilde{y}_t - \tilde{y}_t^*\|^2, \quad D_t = 2\eta\langle\tilde{\varepsilon}_t, \tilde{v}_t\rangle, \quad X_t = 2\gamma^2\|\tilde{\varepsilon}_t\|^2,$$

$$Y_t = \frac{4\eta^2 l_{g,1}^2}{\lambda^2\mu^3\gamma}\tilde{L}_t^2 \sum_{j=1}^{t} d_{t,j}\|\tilde{y}_j - \tilde{y}_j^*\|^2,$$

$$\rho_t = 1 - \mu\gamma, \quad \kappa_t = \frac{4\eta^2 l_{g,1}^2}{\lambda^2\mu^3\gamma}\|\tilde{u}_t\|^2, \quad \sigma_t = 2\gamma\sigma_{g,1}, \quad \nu_t = 2\gamma^2\sigma_{g,1}^2,$$

yielding the following recursion

$$\mathbb{E}\left[\exp(\theta\tilde{V}_{t+1})\right] \leq \mathbb{E}\left[\exp\left\{\theta\left[\left(1 - \frac{\mu\gamma}{2}\right)\tilde{V}_t + 2\gamma^2\sigma_{g,1}^2 + \frac{4\eta^2 l_{g,1}^2}{\lambda^2\mu^3\gamma}\|\tilde{u}_t\|^2 + \frac{4\eta^2 l_{g,1}^2}{\lambda^2\mu^3\gamma}\tilde{L}_t^2 \sum_{j=1}^{t} d_{t,j}\tilde{V}_j\right]\right\}\right] \tag{31}$$

for all $\theta$ satisfying

$$0 \leq \theta \leq \min\left\{\frac{\mu}{8\gamma\sigma_{g,1}^2}, \frac{1}{4\gamma^2\sigma_{g,1}^2}\right\} \leq \frac{\mu}{8\gamma\sigma_{g,1}^2}, \tag{32}$$

where in (31) we denote $\tilde{V}_t := \|\tilde{y}_t - \tilde{y}_t^*\|^2$, and the last inequality of (32) uses $\gamma \leq 1/2l_{g,1} \leq 1/2\mu$. By Lemma C.4 we use induction to show that for any $t \geq 1$ and $\lambda$ satisfying (32), it holds that

$$
\mathbb{E}\left[\exp(\theta\tilde{V}_t)\right] \leq \mathbb{E}\left[\exp\left\{\theta\left[\left(1-\frac{\mu\gamma}{2}\right)^{t-1}\tilde{V}_1 + \frac{4\gamma\sigma_{g,1}^2}{\mu} + \frac{4\eta^2l_{g,1}^2}{\lambda^2\mu^3\gamma}\sum_{i=1}^{t-1}\left(1-\frac{\mu\gamma}{2}\right)^{t-1-i}\|\tilde{u}_i\|^2\right.\right.\right.
$$
$$
\left. + \frac{4\eta^2l_{g,1}^2}{\lambda^2\mu^3\gamma}\tilde{V}_1\sum_{i=1}^{t-1}\left(1-\frac{\mu\gamma}{2}\right)^{t-1-i}\tilde{L}_i^2 + \frac{16\eta^2l_{g,1}^2}{\lambda^2\mu^4}\sigma_{g,1}^2\sum_{i=1}^{t-1}\left(1-\frac{\mu\gamma}{2}\right)^{t-1-i}\tilde{L}_i^2\right.
$$
$$
\left.\left.\left. + \frac{64\eta^4l_{g,1}^4}{\lambda^4\mu^8\gamma^4}\sum_{i=1}^{t-1}\left(1-\frac{\mu\gamma}{2}\right)^{t-1-i}\alpha_i\tilde{L}_i^2\|\tilde{u}_i\|^2\right]\right\}\right],
$$

where the first and the last lines use the sum of geometric series, and the second line is due to Lemma B.5:

$$
\sum_{i=1}^{t-1}\left(1-\frac{\mu\gamma}{2}\right)^{i-1} \leq \frac{2}{\mu\gamma}, \qquad i\alpha_i(1-\beta)^{i-1} \leq 1,
$$

$$
\sum_{i=1}^{t-1}\left(1-\frac{\mu\gamma}{2}\right)^{t-1-i}\alpha_i\tilde{L}_i^2\sum_{j=1}^{i}\left(1-\frac{\mu\gamma}{2}\right)^{i-j}\|\tilde{u}_j\|^2 \leq \frac{2}{\mu\gamma}\sum_{i=1}^{t-1}\left(1-\frac{\mu\gamma}{2}\right)^{t-1-i}\alpha_i\tilde{L}_i^2\|\tilde{u}_i\|^2.
$$

Moreover, by setting $\vartheta$ as follows, we have

$$
\vartheta := \frac{8\gamma\sigma_{g,1}^2}{\mu} \quad \Longrightarrow \quad \frac{4\gamma\sigma_{g,1}^2}{\mu} \leq \vartheta \quad \text{and} \quad \frac{1}{\vartheta} = \frac{\mu}{8\gamma\sigma_{g,1}^2}.
$$

Hence for any $t \geq 1$ we obtain

$$
\mathbb{E}\left[\exp\left\{\theta\left[\tilde{V}_t - \left(1-\frac{\mu\gamma}{2}\right)^{t-1}\tilde{V}_1 - \frac{4\eta^2l_{g,1}^2}{\lambda^2\mu^3\gamma}\sum_{i=1}^{t-1}\left(1-\frac{\mu\gamma}{2}\right)^{t-1-i}\|\tilde{u}_i\|^2\right.\right.\right.
$$
$$
\left. - \frac{4\eta^2l_{g,1}^2}{\lambda^2\mu^3\gamma}\tilde{V}_1\sum_{i=1}^{t-1}\left(1-\frac{\mu\gamma}{2}\right)^{t-1-i}\tilde{L}_i^2 - \frac{16\eta^2l_{g,1}^2}{\lambda^2\mu^4}\sigma_{g,1}^2\sum_{i=1}^{t-1}\left(1-\frac{\mu\gamma}{2}\right)^{t-1-i}\tilde{L}_i^2\right.
$$
$$
\left.\left.\left. - \frac{64\eta^4l_{g,1}^4}{\lambda^4\mu^8\gamma^4}\sum_{i=1}^{t-1}\left(1-\frac{\mu\gamma}{2}\right)^{t-1-i}\alpha_i\tilde{L}_i^2\|\tilde{u}_i\|^2\right]\right\}\right] \leq \exp(\theta\vartheta) \quad \text{for all} \quad 0 \leq \theta \leq 1/\vartheta.
$$

Taking $\theta = 1/\vartheta$ and applying Markov's inequality and union bound completes the proof. $\qquad\square$

**Lemma C.4.** *Suppose* (31) *holds, where* $\tilde{u}_t$, $\{d_{t,j}\}_{j=1}^t$ *and* $\tilde{L}_t$ *are defined in* (24), (9) *and* (26), *respectively. Choosing* $\gamma = 2\beta/\mu$, *then for any* $t \geq 1$ *we have*

$$
\mathbb{E}\left[\exp(\theta\tilde{V}_t)\right] \leq \mathbb{E}\left[\exp\left\{\theta\left[\left(1-\frac{\mu\gamma}{2}\right)^{t-1}\tilde{V}_1 + 2\gamma^2\sigma_{g,1}^2\sum_{i=1}^{t-1}\left(1-\frac{\mu\gamma}{2}\right)^{i-1} + \frac{4\eta^2l_{g,1}^2}{\lambda^2\mu^3\gamma}\sum_{i=1}^{t-1}\left(1-\frac{\mu\gamma}{2}\right)^{t-1-i}\|\tilde{u}_i\|^2\right.\right.\right.
$$
$$
\left. + \frac{4\eta^2l_{g,1}^2}{\lambda^2\mu^3\gamma}\tilde{V}_1\sum_{i=1}^{t-1}\left(1-\frac{\mu\gamma}{2}\right)^{t-1-i}i\alpha_i(1-\beta)^{i-1}\tilde{L}_i^2 + \frac{16\eta^2l_{g,1}^2}{\lambda^2\mu^4}\sigma_{g,1}^2\sum_{i=1}^{t-1}\left(1-\frac{\mu\gamma}{2}\right)^{t-1-i}\tilde{L}_i^2\right.
$$
$$
\left.\left.\left. + \frac{32\eta^4l_{g,1}^4}{\lambda^4\mu^7\gamma^3}\sum_{i=1}^{t-1}\left(1-\frac{\mu\gamma}{2}\right)^{t-1-i}\alpha_i\tilde{L}_i^2\sum_{j=1}^{i}\left(1-\frac{\mu\gamma}{2}\right)^{i-j}\|\tilde{u}_j\|^2\right]\right\}\right].
$$

$$\tag{33}$$

*Proof of Lemma C.4.* We use induction to show that (33) holds for any $t \geq 1$ and $\lambda$ satisfying (32).

**Base Case.** For the base case $t = 1$, it is easy to check that

$$
\mathbb{E}[\exp(\theta\tilde{V}_1)] \leq \mathbb{E}[\exp(\theta\tilde{V}_1)].
$$

**Induction Step.** Now we assume that the induction hypothesis (33) holds for $1 \leq k \leq t$, then for $k = t+1$ we have

$$\mathbb{E}[\exp(\theta \tilde{V}_{t+1})] \leq \mathbb{E}[\exp(\theta[(A_1) + (A_2) + (A_3) + (A_4) + (A_5) + (A_6)])],$$

where $(A_1), (A_2), (A_3), (A_4), (A_5)$ and $(A_6)$ are defined as

$$(A_1) = \left(1 - \frac{\mu\gamma}{2}\right)\left(1 - \frac{\mu\gamma}{2}\right)^{t-1} \tilde{V}_1,$$

$$(A_2) = 2\gamma^2\sigma_{g,1}^2 + 2\gamma^2\sigma_{g,1}^2 \left(1 - \frac{\mu\gamma}{2}\right) \sum_{i=1}^{t-1} \left(1 - \frac{\mu\gamma}{2}\right)^{i-1},$$

$$(A_3) = \frac{4\eta^2 l_{g,1}^2}{\lambda^2\mu^3\gamma}\|\tilde{u}_t\|^2 + \frac{4\eta^2 l_{g,1}^2}{\lambda^2\mu^3\gamma}\left(1 - \frac{\mu\gamma}{2}\right) \sum_{i=1}^{t-1} \left(1 - \frac{\mu\gamma}{2}\right)^{t-1-i} \|\tilde{u}_i\|^2,$$

$$(A_4) = \frac{4\eta^2 l_{g,1}^2}{\lambda^2\mu^3\gamma}\tilde{L}_t^2 \sum_{j=1}^{t} d_{t,j}\left(1 - \frac{\mu\gamma}{2}\right)^{j-1} \tilde{V}_1 + \frac{4\eta^2 l_{g,1}^2}{\lambda^2\mu^3\gamma}\tilde{V}_1\left(1 - \frac{\mu\gamma}{2}\right) \sum_{i=1}^{t-1} \left(1 - \frac{\mu\gamma}{2}\right)^{t-1-i} i\alpha_i(1-\beta)^{i-1}\tilde{L}_i^2,$$

$$(A_5) = \frac{4\eta^2 l_{g,1}^2}{\lambda^2\mu^3\gamma}\tilde{L}_t^2 \sum_{j=1}^{t} d_{t,j} \cdot 2\gamma^2\sigma_{g,1}^2 \sum_{i=1}^{j-1} \left(1 - \frac{\mu\gamma}{2}\right)^{i-1} + \frac{16\eta^2 l_{g,1}^2}{\lambda^2\mu^4}\sigma_{g,1}^2\left(1 - \frac{\mu\gamma}{2}\right) \sum_{i=1}^{t-1} \left(1 - \frac{\mu\gamma}{2}\right)^{t-1-i} \tilde{L}_i^2,$$

$$(A_6) = \frac{4\eta^2 l_{g,1}^2}{\lambda^2\mu^3\gamma}\tilde{L}_t^2 \sum_{j=1}^{t} d_{t,j}\frac{4\eta^2 l_{g,1}^2}{\lambda^2\mu^3\gamma} \sum_{i=1}^{j-1} \left(1 - \frac{\mu\gamma}{2}\right)^{j-1-i} \|\tilde{u}_i\|^2$$

$$+ \frac{32\eta^4 l_{g,1}^4}{\lambda^4\mu^7\gamma^3}\left(1 - \frac{\mu\gamma}{2}\right) \sum_{i=1}^{t-1} \left(1 - \frac{\mu\gamma}{2}\right)^{t-1-i} \alpha_i\tilde{L}_i^2 \sum_{j=1}^{i} \left(1 - \frac{\mu\gamma}{2}\right)^{i-j} \|\tilde{u}_j\|^2.$$

We continue to bound each term individually.

**Bounding $(A_1)$.**

$$(A_1) = \left(1 - \frac{\mu\gamma}{2}\right)\left(1 - \frac{\mu\gamma}{2}\right)^{t-1} \tilde{V}_1 = \left(1 - \frac{\mu\gamma}{2}\right)^{t} \tilde{V}_1.$$

**Bounding $(A_2)$.**

$$(A_2) = 2\gamma^2\sigma_{g,1}^2 + 2\gamma^2\sigma_{g,1}^2\left(1 - \frac{\mu\gamma}{2}\right) \sum_{i=1}^{t-1} \left(1 - \frac{\mu\gamma}{2}\right)^{i-1} = 2\gamma^2\sigma_{g,1}^2\left(1 - \frac{\mu\gamma}{2}\right) \sum_{i=1}^{t} \left(1 - \frac{\mu\gamma}{2}\right)^{i-1}.$$

**Bounding $(A_3)$.**

$$(A_3) = \frac{4\eta^2 l_{g,1}^2}{\lambda^2\mu^3\gamma}\|\tilde{u}_t\|^2 + \frac{4\eta^2 l_{g,1}^2}{\lambda^2\mu^3\gamma}\left(1 - \frac{\mu\gamma}{2}\right) \sum_{i=1}^{t-1} \left(1 - \frac{\mu\gamma}{2}\right)^{t-1-i} \|\tilde{u}_i\|^2 = \frac{4\eta^2 l_{g,1}^2}{\lambda^2\mu^3\gamma} \sum_{i=1}^{t} \left(1 - \frac{\mu\gamma}{2}\right)^{t-i} \|\tilde{u}_i\|^2.$$

**Bounding $(A_4)$.** By Lemma B.3 and the choice of $\gamma = 2\beta/\mu$, we have

$$\frac{4\eta^2 l_{g,1}^2}{\lambda^2\mu^3\gamma}\tilde{L}_t^2 \sum_{j=1}^{t} d_{t,j}\left(1 - \frac{\mu\gamma}{2}\right)^{j-1} \tilde{V}_1 = \frac{4\eta^2 l_{g,1}^2}{\lambda^2\mu^3\gamma}\tilde{L}_t^2 \sum_{j=1}^{t} \alpha_t(1-\beta)^{t-j}(1-\beta)^{j-1}\tilde{V}_1$$

$$= \frac{4\eta^2 l_{g,1}^2}{\lambda^2\mu^3\gamma}\tilde{L}_t^2 \sum_{j=1}^{t} \alpha_t(1-\beta)^{t-1}\tilde{V}_1$$

$$= \frac{4\eta^2 l_{g,1}^2}{\lambda^2\mu^3\gamma}t\alpha_t(1-\beta)^{t-1}\tilde{L}_t^2\tilde{V}_1.$$

Then we obtain

$$
\begin{aligned}
(A_4) &= \frac{4\eta^2 l_{g,1}^2}{\lambda^2\mu^3\gamma}\tilde{L}_t^2\sum_{j=1}^t d_{t,j}\left(1-\frac{\mu\gamma}{2}\right)^{j-1}\tilde{V}_1 + \frac{4\eta^2 l_{g,1}^2}{\lambda^2\mu^3\gamma}\tilde{V}_1\left(1-\frac{\mu\gamma}{2}\right)\sum_{i=1}^{t-1}\left(1-\frac{\mu\gamma}{2}\right)^{t-1-i}i\alpha_i(1-\beta)^{i-1}\tilde{L}_i^2 \\
&= \frac{4\eta^2 l_{g,1}^2}{\lambda^2\mu^3\gamma}t\alpha_t(1-\beta)^{t-1}\tilde{L}_t^2\tilde{V}_1 + \frac{4\eta^2 l_{g,1}^2}{\lambda^2\mu^3\gamma}\tilde{V}_1\left(1-\frac{\mu\gamma}{2}\right)\sum_{i=1}^{t-1}\left(1-\frac{\mu\gamma}{2}\right)^{t-1-i}i\alpha_i(1-\beta)^{i-1}\tilde{L}_i^2 \\
&= \frac{4\eta^2 l_{g,1}^2}{\lambda^2\mu^3\gamma}\tilde{V}_1\sum_{i=1}^t\left(1-\frac{\mu\gamma}{2}\right)^{t-i}i\alpha_i(1-\beta)^{i-1}\tilde{L}_i^2.
\end{aligned}
$$

**Bounding $(A_5)$.** By Lemma B.3 and the choice of $\gamma = 2\beta/\mu$, we have

$$
\begin{aligned}
\frac{4\eta^2 l_{g,1}^2}{\lambda^2\mu^3\gamma}\tilde{L}_t^2\sum_{j=1}^t d_{t,j}\cdot 2\gamma^2\sigma_{g,1}^2\sum_{i=1}^{j-1}\left(1-\frac{\mu\gamma}{2}\right)^{i-1} &\le \frac{8\eta^2 l_{g,1}^2}{\lambda^2\mu^4\gamma^2}2\gamma^2\sigma_{g,1}^2\tilde{L}_t^2\sum_{j=1}^t d_{t,j} \\
&= \frac{16\eta^2 l_{g,1}^2}{\lambda^2\mu^4}\sigma_{g,1}^2\tilde{L}_t^2.
\end{aligned}
$$

Then we obtain

$$
\begin{aligned}
(A_5) &= \frac{4\eta^2 l_{g,1}^2}{\lambda^2\mu^3\gamma}\tilde{L}_t^2\sum_{j=1}^t d_{t,j}\cdot 2\gamma^2\sigma_{g,1}^2\sum_{i=1}^{j-1}\left(1-\frac{\mu\gamma}{2}\right)^{i-1} + \frac{16\eta^2 l_{g,1}^2}{\lambda^2\mu^4}\sigma_{g,1}^2\left(1-\frac{\mu\gamma}{2}\right)\sum_{i=1}^{t-1}\left(1-\frac{\mu\gamma}{2}\right)^{t-1-i}\tilde{L}_i^2 \\
&\le \frac{16\eta^2 l_{g,1}^2}{\lambda^2\mu^4}\sigma_{g,1}^2\tilde{L}_t^2 + \frac{16\eta^2 l_{g,1}^2}{\lambda^2\mu^4}\sigma_{g,1}^2\sum_{i=1}^{t-1}\left(1-\frac{\mu\gamma}{2}\right)^{t-i}\tilde{L}_i^2 \\
&= \frac{16\eta^2 l_{g,1}^2}{\lambda^2\mu^4}\sigma_{g,1}^2\sum_{i=1}^t\left(1-\frac{\mu\gamma}{2}\right)^{t-i}\tilde{L}_i^2.
\end{aligned}
$$

**Bounding $(A_6)$.** By Lemma B.3 and the choice of $\gamma = 2\beta/\mu$, we have

$$
\begin{aligned}
\frac{4\eta^2 l_{g,1}^2}{\lambda^2\mu^3\gamma}\tilde{L}_t^2\sum_{j=1}^t d_{t,j}\frac{4\eta^2 l_{g,1}^2}{\lambda^2\mu^3\gamma}\sum_{i=1}^{j-1}\left(1-\frac{\mu\gamma}{2}\right)^{j-1-i}\|\tilde{u}_i\|^2 &\le \frac{4\eta^2 l_{g,1}^2}{\lambda^2\mu^3\gamma}\frac{8\eta^2 l_{g,1}^2}{\lambda^2\mu^4\gamma^2}\tilde{L}_t^2\sum_{j=1}^t d_{t,j}\|\tilde{u}_j\|^2 \\
&\le \frac{4\eta^2 l_{g,1}^2}{\lambda^2\mu^3\gamma}\frac{8\eta^2 l_{g,1}^2}{\lambda^2\mu^4\gamma^2}\alpha_t\tilde{L}_t^2\sum_{j=1}^t(1-\beta)^{t-j}\|\tilde{u}_j\|^2 \\
&= \frac{32\eta^4 l_{g,1}^4}{\lambda^4\mu^7\gamma^3}\alpha_t\tilde{L}_t^2\sum_{j=1}^t\left(1-\frac{\mu\gamma}{2}\right)^{t-j}\|\tilde{u}_j\|^2.
\end{aligned}
$$

Then we obtain

$$
\begin{aligned}
(A_6) &= \frac{4\eta^2 l_{g,1}^2}{\lambda^2\mu^3\gamma}\tilde{L}_t^2\sum_{j=1}^t d_{t,j}\frac{4\eta^2 l_{g,1}^2}{\lambda^2\mu^3\gamma}\sum_{i=1}^{j-1}\left(1-\frac{\mu\gamma}{2}\right)^{j-1-i}\|\tilde{u}_i\|^2 \\
&\quad + \frac{32\eta^4 l_{g,1}^4}{\lambda^4\mu^7\gamma^3}\left(1-\frac{\mu\gamma}{2}\right)\sum_{i=1}^{t-1}\left(1-\frac{\mu\gamma}{2}\right)^{t-1-i}\alpha_i\tilde{L}_i^2\sum_{j=1}^i\left(1-\frac{\mu\gamma}{2}\right)^{i-j}\|\tilde{u}_j\|^2 \\
&\le \frac{32\eta^4 l_{g,1}^4}{\lambda^4\mu^7\gamma^3}\alpha_t\tilde{L}_t^2\sum_{j=1}^t\left(1-\frac{\mu\gamma}{2}\right)^{t-j}\|\tilde{u}_j\|^2 + \frac{32\eta^4 l_{g,1}^4}{\lambda^4\mu^7\gamma^3}\sum_{i=1}^{t-1}\left(1-\frac{\mu\gamma}{2}\right)^{t-i}\alpha_i\tilde{L}_i^2\sum_{j=1}^i\left(1-\frac{\mu\gamma}{2}\right)^{i-j}\|\tilde{u}_j\|^2 \\
&= \frac{32\eta^4 l_{g,1}^4}{\lambda^4\mu^7\gamma^3}\sum_{i=1}^t\left(1-\frac{\mu\gamma}{2}\right)^{t-i}\alpha_i\tilde{L}_i^2\sum_{j=1}^i\left(1-\frac{\mu\gamma}{2}\right)^{i-j}\|\tilde{u}_j\|^2.
\end{aligned}
$$

**Final Bound for the Induction Step.** Putting these terms together and rearranging yields

$$
\begin{aligned}
\mathbb{E}\left[\exp(\theta \tilde{V}_{t+1})\right] \leq \mathbb{E}\Bigg[ \exp\Bigg\{ &\theta \Bigg[ \left(1 - \frac{\mu\gamma}{2}\right)^t \tilde{V}_1 + 2\gamma^2 \sigma_{g,1}^2 \sum_{i=1}^t \left(1 - \frac{\mu\gamma}{2}\right)^{i-1} + \frac{4\eta^2 l_{g,1}^2}{\lambda^2 \mu^3 \gamma} \sum_{i=1}^t \left(1 - \frac{\mu\gamma}{2}\right)^{t-i} \|\tilde{u}_i\|^2 \\
&+ \frac{4\eta^2 l_{g,1}^2}{\lambda^2 \mu^3 \gamma} \tilde{V}_1 \sum_{i=1}^t \left(1 - \frac{\mu\gamma}{2}\right)^{t-i} i\alpha_i (1-\beta)^{i-1} \tilde{L}_i^2 + \frac{16\eta^2 l_{g,1}^2}{\lambda^2 \mu^4} \sigma_{g,1}^2 \sum_{i=1}^t \left(1 - \frac{\mu\gamma}{2}\right)^{t-i} \tilde{L}_i^2 \\
&+ \frac{32\eta^4 l_{g,1}^4}{\lambda^4 \mu^7 \gamma^3} \sum_{i=1}^t \left(1 - \frac{\mu\gamma}{2}\right)^{t-i} \alpha_i \tilde{L}_i^2 \sum_{j=1}^i \left(1 - \frac{\mu\gamma}{2}\right)^{i-j} \|\tilde{u}_j\|^2 \Bigg] \Bigg\} \Bigg],
\end{aligned}
$$

which aligns with (33) for $k = t + 1$. Thus, the induction step is complete, and (33) holds for any $t \geq 1$. $\quad\square$

# D Convergence Analysis of AdamBO (Algorithm 1)

In this section, we provide detailed convergence analysis of Algorithm 1 (or equivalently, Algorithm 3). Before presenting the lemmas and the main theorem, we will first define (or restate) a few key concepts and useful notations.

## D.1 Technical Definitions and Useful Notations

**Filtration.** Define $\mathcal{F}_{\text{init}}$ as the filtration for updating $y_1$ (i.e., the filtration of warm-start phase):

$$\mathcal{F}_t^{\text{init}} = \sigma(\pi_0, \dots, \pi_{T_0 - 1}).$$

For any $t \geq 2$, define $\mathcal{F}_t^x$ and $\mathcal{F}_t^y$ as the filtrations of the randomness used in updating $x_t$ and $y_t$, respectively, before the $t$-th iteration:

$$\mathcal{F}_t^x = \sigma(\bar{\xi}_1, \dots, \bar{\xi}_{t-1}), \quad \mathcal{F}_t^y = \sigma(\zeta_1, \dots, \zeta_{t-1}),$$

where $\sigma(\cdot)$ denotes the $\sigma$-algebra generated by the random variables within the argument. Additionally, let $\mathcal{F}_t$ denote the filtration of all randomness before the $t$-th iteration:

$$\mathcal{F}_t = \sigma(\mathcal{F}_{\text{init}} \cup \mathcal{F}_t^x \cup \mathcal{F}_t^y).$$

**Expectation.** We use $\mathbb{E}_t[\cdot]$ to denote the conditional expectation $\mathbb{E}[\cdot \mid \mathcal{F}_t]$.

**Auxiliary Sequence.** Note that $\hat{m}_t$ (line 7 of Algorithm 3) can be written as

$$\hat{m}_t = (1 - \alpha_t)\hat{m}_{t-1} + \alpha_t \hat{\nabla}\phi(x_t, y_t; \bar{\xi}_t) = \sum_{j=1}^t d_{t,j} \hat{\nabla}\phi(x_t, y_t; \bar{\xi}_t). \tag{34}$$

Similar to Appendix C.2, we introduce the following auxiliary sequence $\{\hat{u}_t\}$ for our analysis:

$$\hat{u}_t = (1 - \alpha_t)\hat{u}_{t-1} + \alpha_t \hat{\nabla}\phi(x_t, y_t^*; \bar{\xi}_t) = \sum_{j=1}^t d_{t,j} \hat{\nabla}\phi(x_t, y_t^*; \bar{\xi}_t). \tag{35}$$

**Other Definitions.** We define the deviation of the rescaled auxiliary momentum from the conditional expectation of the hypergradient estimator as

$$\epsilon_t := \hat{u}_t - \mathbb{E}_t[\hat{\nabla}\phi(x_t, y_t^*; \bar{\xi}_t)]. \tag{36}$$

Also, let $h_t$ be the learning rate vector and $H_t$ be the learning rate matrix:

$$h_t := \frac{\eta}{\sqrt{\hat{v}_t} + \lambda} \quad \text{and} \quad H_t := \text{diag}(h_t). \tag{37}$$

Then the update rule for upper-level variable $x_t$ (line 10 of Algorithm 1) can be written as

$$x_{t+1} = x_t - h_t \odot \hat{m}_t = x_t - H_t \hat{m}_t. \tag{38}$$

**Stopping Time.** Given a large enough constant $G$ as defined in Theorem D.12, denote $L$ and $\psi$ as

$$L = L_0 + L_1 G \quad \text{and} \quad \psi = \frac{C_L G^2}{2L}, \tag{39}$$

where constants $L_0, L_1$ and $C_L$ are defined in (13) and (42). Now we formally define the stopping time $\tau$ as

$$\tau := \min\{t \mid \Phi(x_t) - \Phi^* > \psi\} \wedge (T+1). \tag{40}$$

In other words, $\tau$ is the first time when the sub-optimality gap is strictly larger than $\psi$, truncated at $T+1$ to make sure it is bounded. Based on Lemma D.1, we know that if $t < \tau$, we have both $\Phi(x_t) - \Phi^* \leq \psi$ and $\|\nabla\Phi(x_t)\| \leq G$.

**Constants.** We define the following constants, which will be useful for analysis.

$$G_t = \max_{1 \leq k \leq t} \|\nabla\Phi(x_k)\|, \quad \hat{L}_t = L_0 + L_1 G_t, \quad L = L_0 + L_1 G, \quad \Delta_1 = \Phi(x_1) - \Phi^*, \tag{41}$$

$$C_L = \frac{L_{x,1}}{\sqrt{L_{x,1}^2 + L_{y,1}^2}}, \quad C_{u,0} = C_{\phi,0} + G, \quad C_{u,1} = C_{\phi,1} + L_1 G, \tag{42}$$

$$\sigma_\phi = \frac{\mu + 3l_{g,1} + \sigma_{g,2}}{\mu} + \frac{2l_{g,1} + \sigma_{g,2}}{\mu} l_{f,0} + \frac{2l_{g,1} + \sigma_{g,2}}{\mu}(L_{y,0} + L_{y,1}l_{f,0})r. \tag{43}$$

$$C_\beta \geq \max\left\{\frac{8e\sigma_\phi^4 G^2 \max\{1,\iota\}}{c_1^2\delta\lambda^2\epsilon^4}, \frac{8C_2 e\Delta_1 L\sigma_\phi G^3}{c_1 c_2\delta\lambda^2\epsilon^4}\left(1 + \frac{\sigma_\phi^2 G}{c_1\lambda\epsilon^2}\right)\max\{1,\sqrt{\iota},\iota\}, \right.$$
$$\left. \left(\frac{32e\sigma_\phi^4 G^2}{c_1^2\delta\lambda^2\epsilon^4}\right)^2, \left(\frac{48C_2 e\Delta_1 L\sigma_\phi G^3}{c_1 c_2\delta\lambda^2\epsilon^4}\left(1 + \frac{\sigma_\phi^2 G}{c_1\lambda\epsilon^2}\right)\max\{1,\sqrt{\iota},\iota\}\right)^2\right\}. \tag{44}$$

Besides, constants $L_0, L_1$ are defined in (13), $C_{\phi,0}, C_{\phi,1}$ are defined in (16), and $r$ is defined in (12), respectively.

## D.2 Auxiliary Lemmas

We first introduce the following useful lemma, which is crucial for the subsequent stopping time analysis and for establishing the contradiction argument.

**Lemma D.1.** *Under Assumption 3.2, we have*

$$\|\nabla\Phi(x)\|^2 \leq \frac{2}{C_L}(L_0 + L_1\|\nabla\Phi(x)\|)(\Phi(x) - \Phi^*),$$

*where constants $L_0, L_1$ and $C_L$ are defined in (13) and (42). Further, for any given constant $G > 0$, if we denote $\psi$ as in (39) and $\Phi(x) - \Phi^* \leq \psi$, then we have $\|\nabla\Phi(x)\| \leq G$.*

*Proof of Lemma D.1.* Let $x'$ be

$$x' = x - \frac{C_L\|\nabla\Phi(x)\|}{L_0 + L_1\|\nabla\Phi(x)\|},$$

then we have

$$\|x' - x\| = \frac{C_L\|\nabla\Phi(x)\|}{L_0 + L_1\|\nabla\Phi(x)\|} \leq \frac{C_L}{L_1} = \frac{1}{\sqrt{(1 + l_{g,1}^2/\mu^2)(L_{x,1}^2 + L_{y,1}^2)}} = r,$$

where the inequality can be verified by considering both cases of $\|\nabla\Phi(x)\| \leq L_0/L_1$ and $\|\nabla\Phi(x)\| \geq L_0/L_1$. By Lemma B.10, we have

$$\Phi^* - \Phi(x) \leq \Phi(x') - \Phi(x) \leq \langle\nabla\Phi(x), x' - x\rangle + \frac{L_0 + L_1\|\nabla\Phi(x)\|}{2}\|x' - x\|^2$$

$$= -\frac{C_L(2 - C_L)}{2(L_0 + L_1\|\nabla\Phi(x)\|)}\|\nabla\Phi(x)\|^2.$$

Rearranging the above inequality yields

$$\|\nabla\Phi(x)\|^2 \leq \frac{2(L_0 + L_1\|\nabla\Phi(x)\|)}{C_L(2 - C_L)}(\Phi(x) - \Phi^*) \leq \frac{2(L_0 + L_1\|\nabla\Phi(x)\|)}{C_L}(\Phi(x) - \Phi^*). \tag{45}$$

where the last inequality uses the definition of $C_L$ in (42) and $C_L \leq 1$.

Now define the function $\varphi : \mathbb{R}_0^+ \to \mathbb{R}$ as

$$\varphi(u) := \frac{C_L u^2}{2(L_0 + L_1 u)}.$$

It is easy to verify $\varphi$ is increasing and $\varphi(u) \in [0, \infty)$. Thus, $\varphi$ is invertible and $\varphi^{-1}$ is also increasing. Then for any constant $G \geq 0$, denote $L$ and $\psi$ as in (39),

$$L = L_0 + L_1 G, \qquad \psi = \frac{C_L G^2}{2L} = \varphi(G).$$

The property of function $\varphi^{-1}$ and (45) imply that if $\Phi(x) - \Phi^* \leq \psi$, we have

$$\|\nabla\Phi(x)\| \leq \varphi^{-1}(\Phi(x) - \Phi^*) \leq \varphi^{-1}(\psi) = G.$$

$\square$

Note that when $t < \tau$, some of the quantities in Algorithm 1 and Appendix D.1 are bounded almost surely. In particular, we have the following lemma.

**Lemma D.2.** *If $t < \tau$, we have*

$$\|\nabla\Phi(x_t)\| \leq G, \quad \hat{L}_t \leq L, \quad \|\hat{u}_t\| \leq C_{u,0}, \quad h_t \preceq \frac{\eta}{\lambda}, \quad \|H_t\| \preceq \frac{\eta}{\lambda}.$$

*where $h_t$ is defined in (37), constants $\hat{L}_t, L$ and $C_{u,0}$ are defined in (41) and (42), respectively.*

*Proof of Lemma D.2.* By Lemma D.1 and definition of $\tau$, we have $\|\nabla\Phi(x_t)\| \leq G$ if $t < \tau$. Also, recall the definition of $G_t$, $\hat{L}_t$ and $L$ as in (41), we have $G_t = \max_{k \leq t} \|\nabla\Phi(x_k)\| \leq G$ if $t < \tau$, and hence gives $\hat{L}_t = L_0 + L_1 G_t \leq L_0 + L_1 G = L$. Before bounding $\|\hat{u}_t\|$, we first show $\|\hat{\nabla}\phi(x_t, y_t^*; \bar{\xi}_t)\| \leq C_{u,0}$. Lemma B.14 directly implies that if $t < \tau$, then

$$\|\hat{\nabla}\phi(x_t, y_t^*; \bar{\xi}_t)\| \leq C_{\phi,0} + (C_{\phi,1} + L_1\|\nabla\Phi(x_t)\|)\|y_t^* - y_t^*\| + \|\nabla\Phi(x_t)\| \leq C_{\phi,0} + G = C_{u,0},$$

where the last equality is due to the definition of $C_{u,0}$ in (42). Now $\|\hat{u}_t\|$ can be bounded by a standard induction argument as follows. First, for the base case $k = 1$, note that $\|\hat{\nabla}\phi(x_1, y_1^*; \bar{\xi}_1)\| \leq C_{u,0}$. Suppose $\|\hat{u}_{k-1}\| \leq C_{u,0}$ for some $k < \tau$, then by update rule of $\hat{u}_k$ in (35) we have

$$\|\hat{u}_k\| \leq (1 - \alpha_k)\|\hat{u}_{k-1}\| + \alpha_k\|\hat{\nabla}\phi(x_k, y_k; \bar{\xi}_k)\| \leq C_{u,0}.$$

Therefore, the induction is complete. The last two results directly follow from the definitions of $h_t$ and $H_t$ in (37). $\square$

### D.3 Proof of Lemma 4.3

In the next lemma, we provide high probability bound for the warm-start phase.

**Lemma D.3** (Warm-Start, Restatement of Lemma 4.3). *Suppose that Assumptions 3.2 and 3.3 hold. Let $\{y_t^{init}\}$ be the iterates generated by Algorithm 2 with constant learning rate $\gamma \leq 1/2l_{g,1}$. Then for any given $\delta \in (0, 1)$, the following estimate holds with probability at least $1 - \delta/4$ over the randomness in $\mathcal{F}_{init}$ (we denote this event as $\mathcal{E}_0$):*

$$\|y_1 - y_1^*\|^2 \leq \left(1 - \frac{\mu\gamma}{2}\right)^{T_0}\|y_0 - y_0^*\|^2 + \frac{8\gamma\sigma_{g,1}^2}{\mu}\ln\frac{4e}{\delta}. \tag{46}$$

*Proof of Lemma D.3.* For any given $\delta \in (0,1)$ and any fixed $t \geq 0$, we invoke (Cutler et al., 2023, Theorem 30) to obtain that

$$\|y_t^{\text{init}} - y_0^*\|^2 \leq \left(1 - \frac{\mu\gamma}{2}\right)^t \|y_0 - y_0^*\|^2 + \frac{8\gamma\sigma_{g,1}^2}{\mu} \ln \frac{4e}{\delta} \tag{47}$$

holds with probability at least $1 - \delta$ over the randomness in $\mathcal{F}_{\text{init}}$. Set $t = T_0$ and then we have

$$\|y_1 - y_1^*\|^2 = \|y_{T_0}^{\text{init}} - y_0^*\|^2 \leq \left(1 - \frac{\mu\gamma}{2}\right)^{T_0} \|y_0 - y_0^*\|^2 + \frac{8\gamma\sigma_{g,1}^2}{\mu} \ln \frac{4e}{\delta},$$

where the first equality is due to $y_1 = y_{T_0}^{\text{init}}$ and $y_1^* = y_0^*$ (since $x_1 = x_0$) by line 2 of Algorithm 1. $\qquad \square$

## D.4 Proof of Lemma 4.4

The following Lemma D.4 (i.e., the full statement of Lemma 4.4) is a direct application of the randomness decoupling lemma (i.e., Lemma 4.2) to the actual sequences $\{x_t\}, \{y_t\}$ in Algorithm 1.

**Lemma D.4** (Restatement of Lemma 4.4)**.** *Suppose that Assumptions 3.2 to 3.4 hold. Let $\{y_t\}$ be the iterates generated by Algorithm 1. Under the parameter choices in Theorem D.12, let $\eta$ further satisfy*

$$\eta \leq c_2 \min\left\{ \frac{r\lambda}{G_T}, \frac{\lambda}{6L}, \frac{\sigma_\phi\lambda\beta}{\hat{L}_T G_T \max\{1, \sqrt{\iota}, \ln(1/\beta), \ln(C_\beta)\}}, \frac{\lambda^{3/2}\beta}{\hat{L}_T\sqrt{G_T}} \right\}, \tag{48}$$

*then for any given $\delta \in (0,1)$ and all $t \geq 1$, the following estimate holds with probability at least $1 - \delta/4$ over the randomness in $\mathcal{F}_{T+1}^y$ (we denote this event as $\mathcal{E}_y$):*

$$\begin{aligned}
\|y_t - y_t^*\|^2 &\leq \left( \left(1 - \frac{\mu\gamma}{2}\right)^{t-1} + \frac{4\eta^2 l_{g,1}^2}{\lambda^2\mu^3\gamma} \sum_{i=1}^{t-1} \left(1 - \frac{\mu\gamma}{2}\right)^{t-1-i} \hat{L}_i^2 \right) \|y_1 - y_1^*\|^2 \\
&\quad + \left( \frac{8\gamma}{\mu} \ln \frac{4eT}{\delta} + \frac{16\eta^2 l_{g,1}^2}{\lambda^2\mu^4} \sum_{i=1}^{t-1} \left(1 - \frac{\mu\gamma}{2}\right)^{t-1-i} \hat{L}_i^2 \right) \sigma_{g,1}^2 \\
&\quad + \frac{4\eta^2 l_{g,1}^2}{\lambda^2\mu^3\gamma} \sum_{i=1}^{t-1} \left(1 - \frac{\mu\gamma}{2}\right)^{t-1-i} \|\hat{u}_i\|^2 + \frac{64\eta^4 l_{g,1}^4}{\lambda^4\mu^8\gamma^4} \sum_{i=1}^{t-1} \left(1 - \frac{\mu\gamma}{2}\right)^{t-1-i} \alpha_i \hat{L}_i^2 \|\hat{u}_i\|^2,
\end{aligned} \tag{49}$$

*where constant $\hat{L}_i$ and sequence $\{\hat{u}_i\}$ are defined in (41) and (35), respectively.*

*Proof of Lemma D.4.* First, with the parameter choices in Theorem D.12 and the additional choice for $\eta$ as in (48), we can follow the same procedure as Lemma D.13 (see "Verification for $\varrho \leq \min\{r, 1/4L_1\}$") to show that $\|y_t - y_t^*\| \leq r$ for all $t \in [T]$. Thus, the condition for applying Lemma B.15 is satisfied. Recall the definitions of $\hat{m}_t$ and $\hat{u}_t$ in (34) and (35), we have

$$\begin{aligned}
\|\hat{m}_t - \hat{u}_t\|^2 &\leq \left\| \sum_{j=1}^t d_{t,j}(\hat{\nabla}\phi(x_j, y_j; \bar{\xi}_j) - \hat{\nabla}\phi(x_j, y_j^*; \bar{\xi}_j)) \right\|^2 \\
&\leq \sum_{j=1}^t d_{t,j} \|\hat{\nabla}\phi(x_j, y_j; \bar{\xi}_j) - \hat{\nabla}\phi(x_j, y_j^*; \bar{\xi}_j)\|^2 \\
&\leq \sum_{j=1}^t d_{t,j}(L_0 + L_1\|\nabla\Phi(x_j)\|)^2 \|y_j - y_j^*\|^2 \leq \hat{L}_t^2 \sum_{j=1}^t d_{t,j} \|y_j - y_j^*\|^2,
\end{aligned} \tag{50}$$

where the second inequality uses Jensen's inequality, the third inequality is due to Lemma B.15, and the last inequality uses the definition of $\hat{L}_t$ in (41). By the update rule in Algorithm 3, we have

$$\|x_{t+1} - x_t\|^2 \leq \|H_t\|^2 \|\hat{m}_t\|^2 \leq \frac{\eta^2}{\lambda^2} \|\hat{m}_t\|^2 \leq \frac{2\eta^2}{\lambda^2} \left( \|\hat{u}_t\|^2 + \|\hat{m}_t - \hat{u}_t\|^2 \right)$$

$$\leq \frac{2\eta^2}{\lambda^2} \left( \|\hat{u}_t\|^2 + \hat{L}_t^2 \sum_{j=1}^t d_{t,j} \|y_j - y_j^*\|^2 \right),$$

where the first inequality uses (37); the second inequality is due to Lemma D.2; the third inequality uses Young's inequality; and the last inequality is due to (50). This implies that the sequence $\{x_t\}$ and the randomness $\{\bar{\xi}_t\}$ generated by Algorithm 1 satisfy the condition (28) in Lemma C.3. Therefore, the result follows by applying Lemma C.3 with $\{\tilde{x}_t\} = \{x_t\}$ and $\{\hat{\bar{\xi}}_t\} = \{\bar{\xi}_t\}$. □

**Remark.** In the end, we will show $\tau = T + 1$ in the proof of Theorem D.12 (i.e., the Full statement of Theorem 4.1), thus we can apply Lemma D.2 to obtain $G_T \leq G$ and $\hat{L}_T \leq L$. This suggests that under event $\mathcal{E}_0 \cap \mathcal{E}_y$, the additional requirement (48) is actually included in the parameter choices of Theorem D.12. Therefore, there is no need to worry about this temporary iterate-dependent requirement for the choice of $\eta$.

### D.5 Proof of Lemma 4.5

Before proving Lemma 4.5, first note that when $t < \tau$ and $\mathcal{E}_0 \cap \mathcal{E}_y$ holds, some of the time-dependent quantities (such as $\hat{L}_t$ and $\|\hat{u}_t\|$) in Lemma D.4 can be well bounded by Lemma D.2. In particular, we have the following two high probability bounds for the lower-level approximation error $\|y_t - y_t^*\|$: the first one, (51), is useful for the convergence analysis; and the second one, (52), is crucial for proving Lemmas D.6 and D.8.

**Lemma D.5.** *Under event $\mathcal{E}_0 \cap \mathcal{E}_y$ and the parameter choices in Lemma D.4, if $t \leq \tau$, we have*

$$
\begin{aligned}
\|y_t - y_t^*\|^2 \leq{}& \left( \left(1 - \frac{\mu\gamma}{2}\right)^{t-1} + \frac{8\eta^2 l_{g,1}^2 L^2}{\lambda^2 \mu^4 \gamma^2} \right) \|y_1 - y_1^*\|^2 + \left( \frac{8\gamma}{\mu} \ln \frac{4eT}{\delta} + \frac{32\eta^2 l_{g,1}^2 L^2}{\lambda^2 \mu^5 \gamma} \right) \sigma_{g,1}^2 \\
&+ \frac{4\eta^2 l_{g,1}^2}{\lambda^2 \mu^3 \gamma} \sum_{i=1}^{t-1} \left(1 - \frac{\mu\gamma}{2}\right)^{t-1-i} \|\hat{u}_i\|^2 + \frac{64\eta^4 l_{g,1}^4 L^2}{\lambda^4 \mu^8 \gamma^4} \sum_{i=1}^{t-1} \left(1 - \frac{\mu\gamma}{2}\right)^{t-1-i} \alpha_i \|\hat{u}_i\|^2
\end{aligned}
\tag{51}
$$

*and*

$$
\begin{aligned}
\|y_t - y_t^*\|^2 \leq{}& \left( 1 + \frac{8\eta^2 l_{g,1}^2 L^2}{\lambda^2 \mu^4 \gamma^2} \right) \|y_1 - y_1^*\|^2 + \left( \frac{8\gamma}{\mu} \ln \frac{4eT}{\delta} + \frac{32\eta^2 l_{g,1}^2 L^2}{\lambda^2 \mu^5 \gamma} \right) \sigma_{g,1}^2 \\
&+ \frac{8\eta^2 l_{g,1}^2 C_{u,0}^2}{\lambda^2 \mu^4 \gamma^2} + \frac{1024\eta^4 l_{g,1}^4 L^2 C_{u,0}^2}{\lambda^4 \mu^8 \gamma^4} \left( 2 + \ln \frac{1}{\beta} \right) =: \varrho^2,
\end{aligned}
\tag{52}
$$

*where constants $L$ and sequence $\{\hat{u}_i\}$ are defined in (41) and (35), respectively.*

*Proof of Lemma D.5.* By Lemma D.2, we know that $\hat{L}_t \leq L$ and $\|\hat{u}_t\| \leq C_{u,0}$ if $t < \tau$. Then under event $\mathcal{E}_0 \cap \mathcal{E}_y$, (51) is obtained by replacing $\hat{L}_i$ with $L$, and (52) is obtained by substituting both $\hat{L}_i$ and $\|\hat{u}_i\|$ with $L$ and $C_{u,0}$, respectively. □

With Lemma D.5 in place, we now formally present the statement of Lemma 4.5 below.

**Lemma D.6** (Restatement of Lemma 4.5)**.** *Under event $\mathcal{E}_0 \cap \mathcal{E}_y$ and the parameter choices in Lemma D.4, if $t < \tau$, we have*

$$
\|\hat{m}_t\| \leq C_{u,0} + C_{u,1}\varrho, \quad \hat{v}_t \preceq (C_{u,0} + C_{u,1}\varrho)^2, \quad \frac{\eta}{C_{u,0} + C_{u,1}\varrho + \lambda} \preceq h_t \preceq \frac{\eta}{\lambda};
$$

*if $t \leq \tau$, we have*

$$
\|\hat{\nabla}\phi(x_t, y_t; \bar{\xi}_t) - \mathbb{E}_t[\hat{\nabla}\phi(x_t, y_t; \bar{\xi}_t)]\| \leq \sigma_\phi,
$$

$$
\|\mathbb{E}_t[\hat{\nabla}\phi(x_t, y_t^*; \bar{\xi}_t)] - \mathbb{E}_{t-1}[\hat{\nabla}\phi(x_{t-1}, y_{t-1}^*; \bar{\xi}_{t-1})]\| \leq L\|x_t - x_{t-1}\|;
$$

*where constants $C_{u,0}, C_{u,1}, \sigma_\phi, L$ and $\varrho$ are defined in (42), (41) and (52), respectively.*

*Proof of Lemma D.6.* By Lemma B.14, under event $\mathcal{E}_0 \cap \mathcal{E}_y$, if $t < \tau$, we have

$$
\begin{aligned}
\|\hat{\nabla}\phi(x_t, y_t; \bar{\xi}_t)\| &\leq C_{\phi,0} + (C_{\phi,1} + L_1\|\nabla\Phi(x_t)\|)\|y_t - y_t^*\| + \|\nabla\Phi(x_t)\| \\
&\leq C_{\phi,0} + G + (C_{\phi,1} + L_1 G)\varrho = C_{u,0} + C_{u,1}\varrho,
\end{aligned}
$$

where the second inequality is due to Lemma D.2 and (52) in Lemma D.5, and the last equality uses the definitions in (42). We can bound $\|\hat{m}_t\|$ by a standard induction argument as follows. First, for the base case $k = 1$, note that

$$\|\hat{m}_1\| = \|\hat{\nabla}\phi(x_1, y_1; \bar{\xi}_1)\| \leq C_{u,0} + C_{u,1}\varrho.$$

Suppose $\|\hat{m}_{k-1}\| \leq C_{u,0} + C_{u,1}\varrho$ for some $k < \tau$, then we have

$$\|\hat{m}_k\| \leq (1 - \alpha_k)\|\hat{m}_{k-1}\| + \alpha_k\|\hat{\nabla}\phi(x_k, y_k; \bar{\xi}_k)\| \leq C_{u,0} + C_{u,1}\varrho.$$

Then we can show $\hat{v}_t \preceq (C_{u,0} + C_{u,1}\varrho)^2$ in a similar way (by induction argument) by noting that

$$(\hat{\nabla}\phi(x_t, y_t; \bar{\xi}_t))^2 \preceq \|\hat{\nabla}\phi(x_t, y_t; \bar{\xi}_t)\|^2 \leq (C_{u,0} + C_{u,1}\varrho)^2.$$

Given the bound on $\hat{v}_t$, it is straight forward to bound the learning rate $h_t$. As for the second last bound, by Lemma B.13 and (52) of Lemma D.5, under event $\mathcal{E}_0 \cap \mathcal{E}_y$, if $t \leq \tau$, we have

$$\|\hat{\nabla}\phi(x_t, y_t; \bar{\xi}_t) - \mathbb{E}_t[\hat{\nabla}\phi(x_t, y_t; \bar{\xi}_t)]\|$$
$$\leq \frac{\mu + 3l_{g,1} + \sigma_{g,2}}{\mu} + \frac{2l_{g,1} + \sigma_{g,2}}{\mu}l_{f,0} + \frac{2l_{g,1} + \sigma_{g,2}}{\mu}(L_{y,0} + L_{y,1}l_{f,0})\|y_t - y_t^*\|$$
$$\leq \frac{\mu + 3l_{g,1} + \sigma_{g,2}}{\mu} + \frac{2l_{g,1} + \sigma_{g,2}}{\mu}l_{f,0} + \frac{2l_{g,1} + \sigma_{g,2}}{\mu}(L_{y,0} + L_{y,1}l_{f,0})\varrho$$
$$\leq \sigma_\phi,$$

where the last equality uses $\varrho \leq r$ by Lemma D.13 and the definition of $\sigma_\phi$ in (43). The last bound can be obtained by applying Lemmas B.15 and D.2:

$$\|\mathbb{E}_t[\hat{\nabla}\phi(x_t, y_t^*; \bar{\xi}_t)] - \mathbb{E}_{t-1}[\hat{\nabla}\phi(x_{t-1}, y_{t-1}^*; \bar{\xi}_{t-1})]\| \leq (L_0 + L_1\|\nabla\Phi(x_{t-1})\|)\|x_t - x_{t-1}\|$$
$$\leq (L_0 + L_1 G)\|x_t - x_{t-1}\|$$
$$= L\|x_t - x_{t-1}\|,$$

where the last inequality uses the definition of $L$ in (41). $\qquad\square$

### D.6 Proof of Lemma 4.6

The following lemma provides a bound for the difference between the actual momentum $\hat{m}_t$ versus the auxiliary momentum $\hat{u}_t$ under the good event $\mathcal{E}_0 \cap \mathcal{E}_y$, which is crucial for establishing the convergence guarantees for Algorithm 1.

**Lemma D.7** (Restatement of Lemma 4.6). *Under event $\mathcal{E}_0 \cap \mathcal{E}_y$ and the parameter choices in Lemma D.4, we have*

$$\sum_{t=1}^{\tau-1} \|\hat{m}_t - \hat{u}_t\|^2 \leq TL^2\left(\left(1 + \frac{8\eta^2 l_{g,1}^2 L^2}{\lambda^2\mu^4\gamma^2}\right)\|y_1 - y_1^*\|^2 + \left(\frac{8\gamma}{\mu}\ln\frac{4eT}{\delta} + \frac{32\eta^2 l_{g,1}^2 L^2}{\lambda^2\mu^5\gamma}\right)\sigma_{g,1}^2\right)$$
$$+ L^2\left(\frac{8\eta^2 l_{g,1}^2}{\lambda^2\mu^4\gamma^2} + \frac{2048\eta^4 l_{g,1}^4 L^2}{\lambda^4\mu^8\gamma^4}\left(2 + \ln\frac{1}{\beta}\right)\right)\sum_{t=1}^{\tau-1}\|\epsilon_t\|^2 + 2\|\mathbb{E}_t[\hat{\nabla}\phi(x_t, y_t^*; \bar{\xi}_t)] - \nabla\Phi(x_t)\|^2$$
$$+ 2L^2\left(\frac{8\eta^2 l_{g,1}^2}{\lambda^2\mu^4\gamma^2} + \frac{2048\eta^4 l_{g,1}^4 L^2}{\lambda^4\mu^8\gamma^4}\left(2 + \ln\frac{1}{\beta}\right)\right)\sum_{t=1}^{\tau-1}\|\nabla\Phi(x_t)\|^2.$$

*Proof of Lemma D.7.* Under event $\mathcal{E}_0 \cap \mathcal{E}_y$, if $t < \tau$, by Lemma D.2 and (50) in Lemma D.4 we have

$$\|\hat{m}_t - \hat{u}_t\|^2 \leq \hat{L}_t^2\sum_{j=1}^{t} d_{t,j}\|y_j - y_j^*\|^2 \leq L^2\sum_{j=1}^{t} d_{t,j}\|y_j - y_j^*\|^2.$$

Now we apply (51) of Lemma D.5 and take summation to obtain

$$\sum_{t=1}^{\tau-1}\sum_{j=1}^{t} d_{t,j}\|y_j - y_j^*\|^2$$

$$\leq \sum_{t=1}^{\tau-1}\sum_{j=1}^{t} d_{t,j}\left(\left(\left(1-\frac{\mu\gamma}{2}\right)^{j-1}+\frac{8\eta^2 l_{g,1}^2 L^2}{\lambda^2\mu^4\gamma^2}\right)\|y_1-y_1^*\|^2+\left(\frac{8\gamma}{\mu}\ln\frac{4eT}{\delta}+\frac{32\eta^2 l_{g,1}^2 L^2}{\lambda^2\mu^5\gamma}\right)\sigma_{g,1}^2\right) \quad (A_1)$$

$$+\sum_{t=1}^{\tau-1}\sum_{j=1}^{t} d_{t,j}\left(\frac{4\eta^2 l_{g,1}^2}{\lambda^2\mu^3\gamma}\sum_{i=1}^{j-1}\left(1-\frac{\mu\gamma}{2}\right)^{j-1-i}\|\hat{u}_i\|^2+\frac{64\eta^4 l_{g,1}^4 L^2}{\lambda^4\mu^8\gamma^4}\sum_{i=1}^{j-1}\left(1-\frac{\mu\gamma}{2}\right)^{j-1-i}\alpha_i\|\hat{u}_i\|^2\right) \quad (A_2)$$

We continue to bound each term individually.

**Bounding $(A_1)$.** By Lemmas B.3 and B.5 and choice of $\gamma = 2\beta/\mu$, we have

$$(A_1) = \sum_{t=1}^{\tau-1}\sum_{j=1}^{t} d_{t,j}\left(\left(\left(1-\frac{\mu\gamma}{2}\right)^{j-1}+\frac{8\eta^2 l_{g,1}^2 L^2}{\lambda^2\mu^4\gamma^2}\right)\|y_1-y_1^*\|^2+\left(\frac{8\gamma}{\mu}\ln\frac{4eT}{\delta}+\frac{32\eta^2 l_{g,1}^2 L^2}{\lambda^2\mu^5\gamma}\right)\sigma_{g,1}^2\right)$$

$$= \sum_{t=1}^{\tau-1}\sum_{j=1}^{t} d_{t,j}\left(1-\frac{\mu\gamma}{2}\right)^{j-1}\|y_1-y_1^*\|^2$$

$$+ \sum_{t=1}^{\tau-1}\sum_{j=1}^{t} d_{t,j}\left(\frac{8\eta^2 l_{g,1}^2 L^2}{\lambda^2\mu^4\gamma^2}\|y_1-y_1^*\|^2+\left(\frac{8\gamma}{\mu}\ln\frac{4eT}{\delta}+\frac{32\eta^2 l_{g,1}^2 L^2}{\lambda^2\mu^5\gamma}\right)\sigma_{g,1}^2\right) \quad (53)$$

$$= \sum_{t=1}^{\tau-1} t\alpha_t(1-\beta)^{t-1}\|y_1-y_1^*\|^2+\sum_{t=1}^{\tau-1}\left(\frac{8\eta^2 l_{g,1}^2 L^2}{\lambda^2\mu^4\gamma^2}\|y_1-y_1^*\|^2+\left(\frac{8\gamma}{\mu}\ln\frac{4eT}{\delta}+\frac{32\eta^2 l_{g,1}^2 L^2}{\lambda^2\mu^5\gamma}\right)\sigma_{g,1}^2\right)$$

$$\leq T\left(\left(1+\frac{8\eta^2 l_{g,1}^2 L^2}{\lambda^2\mu^4\gamma^2}\right)\|y_1-y_1^*\|^2+\left(\frac{8\gamma}{\mu}\ln\frac{4eT}{\delta}+\frac{32\eta^2 l_{g,1}^2 L^2}{\lambda^2\mu^5\gamma}\right)\sigma_{g,1}^2\right),$$

where the last inequality uses $\tau \leq T+1$ by definition of $\tau$.

**Bounding $(A_2)$.** By Lemmas B.3 and B.6 and choice of $\gamma = 2\beta/\mu$, we have

$$(A_2) = \sum_{t=1}^{\tau-1}\sum_{j=1}^{t} d_{t,j}\left(\frac{4\eta^2 l_{g,1}^2}{\lambda^2\mu^3\gamma}\sum_{i=1}^{j-1}\left(1-\frac{\mu\gamma}{2}\right)^{j-1-i}\|\hat{u}_i\|^2+\frac{64\eta^4 l_{g,1}^4 L^2}{\lambda^4\mu^8\gamma^4}\sum_{i=1}^{j-1}\left(1-\frac{\mu\gamma}{2}\right)^{j-1-i}\alpha_i\|\hat{u}_i\|^2\right)$$

$$\leq \frac{4\eta^2 l_{g,1}^2}{\lambda^2\mu^4\gamma^2}\sum_{t=1}^{\tau-1}\sum_{j=1}^{t} d_{t,j}\|\hat{u}_j\|^2+\frac{64\eta^4 l_{g,1}^4 L^2}{\lambda^4\mu^8\gamma^4}\left(32+16\ln\frac{1}{\beta}\right)\sum_{t=1}^{\tau-1}\sum_{j=1}^{t} d_{t,j}\|\hat{u}_j\|^2 \quad (54)$$

$$\leq \left(\frac{4\eta^2 l_{g,1}^2}{\lambda^2\mu^4\gamma^2}+\frac{1024\eta^4 l_{g,1}^4 L^2}{\lambda^4\mu^8\gamma^4}\left(2+\ln\frac{1}{\beta}\right)\right)\sum_{t=1}^{\tau-1}\|\hat{u}_t\|^2.$$

**Final Bound.** Combining (53) and (54) yields

$$\sum_{t=1}^{\tau-1}\sum_{j=1}^{t} d_{t,j}\|y_j-y_j^*\|^2 \leq T\left(\left(1+\frac{8\eta^2 l_{g,1}^2 L^2}{\lambda^2\mu^4\gamma^2}\right)\|y_1-y_1^*\|^2+\left(\frac{8\gamma}{\mu}\ln\frac{4eT}{\delta}+\frac{32\eta^2 l_{g,1}^2 L^2}{\lambda^2\mu^5\gamma}\right)\sigma_{g,1}^2\right)$$

$$+\left(\frac{4\eta^2 l_{g,1}^2}{\lambda^2\mu^4\gamma^2}+\frac{1024\eta^4 l_{g,1}^4 L^2}{\lambda^4\mu^8\gamma^4}\left(2+\ln\frac{1}{\beta}\right)\right)\sum_{t=1}^{\tau-1}\|\hat{u}_t\|^2.$$

In addition, recall the definition of $\hat{u}_t$ and $\epsilon_t$ in (35) and (36), by Young's inequality we have

$$\|\hat{u}_t\|^2 \leq 2\|\epsilon_t\|^2+4\|\mathbb{E}_t[\hat{\nabla}\phi(x_t, y_t^*; \bar{\xi}_t)]-\nabla\Phi(x_t)\|^2+4\|\nabla\Phi(x_t)\|^2.$$

Therefore, we conclude that

$$
\sum_{t=1}^{\tau-1} \|\hat{m}_t - \hat{u}_t\|^2 \le L^2 \sum_{t=1}^{\tau-1} \sum_{j=1}^{t} d_{t,j} \|y_j - y_j^*\|^2
$$

$$
\le TL^2 \left( \left( 1 + \frac{8\eta^2 l_{g,1}^2 L^2}{\lambda^2 \mu^4 \gamma^2} \right) \|y_1 - y_1^*\|^2 + \left( \frac{8\gamma}{\mu} \ln \frac{4eT}{\delta} + \frac{32\eta^2 l_{g,1}^2 L^2}{\lambda^2 \mu^5 \gamma} \right) \sigma_{g,1}^2 \right)
$$

$$
+ L^2 \left( \frac{8\eta^2 l_{g,1}^2}{\lambda^2 \mu^4 \gamma^2} + \frac{2048 \eta^4 l_{g,1}^4 L^2}{\lambda^4 \mu^8 \gamma^4} \left( 2 + \ln \frac{1}{\beta} \right) \right) \sum_{t=1}^{\tau-1} \|\epsilon_t\|^2 + 2 \|\mathbb{E}_t[\hat{\nabla}\phi(x_t, y_t^*; \bar{\xi}_t)] - \nabla\Phi(x_t)\|^2
$$

$$
+ 2L^2 \left( \frac{8\eta^2 l_{g,1}^2}{\lambda^2 \mu^4 \gamma^2} + \frac{2048 \eta^4 l_{g,1}^4 L^2}{\lambda^4 \mu^8 \gamma^4} \left( 2 + \ln \frac{1}{\beta} \right) \right) \sum_{t=1}^{\tau-1} \|\nabla\Phi(x_t)\|^2.
$$

$\square$

## D.7 Proof of Theorem 4.1

The following lemma ensures that $x_{t+1}$ and $x_t$ remain close for sufficiently small $\eta$, allowing us to apply Lemma B.11 in Lemma D.9.

**Lemma D.8.** *Under event $\mathcal{E}_0 \cap \mathcal{E}_y$ and the parameter choices in Lemma D.4, if $t < \tau$, then we have $\|x_{t+1} - x_t\| \le \eta D$ where $D := 2G/\lambda$.*

*Proof of Lemma D.8.* Under event $\mathcal{E}_0 \cap \mathcal{E}_y$, if $t < \tau$, then we have

$$
\|x_{t+1} - x_t\| \le \|H_t\| \|\hat{m}_t\| \le \frac{\eta}{\lambda} \|\hat{m}_t\| \le \frac{\eta(C_{u,0} + C_{u,1}\varrho)}{\lambda} \le \frac{2\eta G}{\lambda} = \eta D,
$$

where the first inequality uses (37), the second inequality is due to Lemma D.2, the third inequality uses Lemma D.6, the fourth inequality is due to Lemma D.13, and the last equality uses the definition of $D$. $\square$

Next, we provide a descent lemma for AdamBO.

**Lemma D.9.** *Under event $\mathcal{E}_0 \cap \mathcal{E}_y$ and the parameter choices in Lemma D.4, if $t < \tau$, we have*

$$
\Phi(x_{t+1}) - \Phi(x_t) \le -\frac{\eta}{4G} \|\nabla\Phi(x_t)\|^2 + \frac{2\eta}{\lambda} \|\hat{m}_t - \hat{u}_t\|^2
$$
$$
+ \frac{4\eta}{\lambda} \|\epsilon_t\|^2 + \frac{4\eta}{\lambda} \|\mathbb{E}_t[\hat{\nabla}\phi(x_t, y_t^*; \bar{\xi}_t)] - \nabla\Phi(x_t)\|^2. \tag{55}
$$

*Proof of Lemma D.9.* By Lemmas D.6 and D.13 and choice of $G$, if $t < \tau$, we have

$$
\frac{\eta I}{2G} \preceq \frac{\eta}{C_{u,0} + C_{u,1}\varrho + \lambda} \preceq H_t \preceq \frac{\eta I}{\lambda}. \tag{56}
$$

Since we choose $\eta \le r/D$, then by Lemma D.8 we have $\|x_{t+1} - x_t\| \le r$ if $t < \tau$. Define $\hat{\epsilon}_t$ and $\epsilon_t$ as

$$
\hat{\epsilon}_t = \hat{m}_t - \nabla\Phi(x_t) \qquad \text{and} \qquad \epsilon_t = \hat{u}_t - \mathbb{E}_t[\hat{\nabla}\phi(x_t, y_t^*; \bar{\xi}_t)]. \tag{57}
$$

For any $t < \tau$, we apply Lemma B.11 to obtain that

$$
\begin{aligned}
\Phi(x_{t+1}) - \Phi(x_t) &\leq \langle \nabla\Phi(x_t), x_{t+1} - x_t \rangle + \frac{L_0 + L_1 \|\nabla\Phi(x_t)\|}{2} \|x_{t+1} - x_t\|^2 \\
&\leq \langle \nabla\Phi(x_t), x_{t+1} - x_t \rangle + \frac{L}{2} \|x_{t+1} - x_t\|^2 \\
&= -\nabla\Phi(x_t)^\top H_t \hat{m}_t + \frac{L}{2} \hat{m}_t^\top H_t^2 \hat{m}_t \\
&\leq -\|\nabla\Phi(x_t)\|_{H_t}^2 - \nabla\Phi(x_t)^\top H_t \hat{\epsilon}_t + \frac{\eta L}{2\lambda} \|\hat{m}_t\|_{H_t}^2 \\
&\leq -\frac{2}{3} \|\nabla\Phi(x_t)\|_{H_t}^2 + \frac{3}{4} \|\hat{\epsilon}_t\|_{H_t}^2 + \frac{\eta L}{\lambda} \left( \|\nabla\Phi(x_t)\|_{H_t}^2 + \|\hat{\epsilon}_t\|_{H_t}^2 \right) \\
&\leq -\frac{1}{2} \|\nabla\Phi(x_t)\|_{H_t}^2 + \|\hat{\epsilon}_t\|_{H_t}^2 \\
&\leq -\frac{\eta}{4G} \|\nabla\Phi(x_t)\|^2 + \frac{\eta}{\lambda} \|\hat{\epsilon}_t\|^2 \\
&\leq -\frac{\eta}{4G} \|\nabla\Phi(x_t)\|^2 + \frac{2\eta}{\lambda} \|\hat{m}_t - \hat{u}_t\|^2 + \frac{4\eta}{\lambda} \|\epsilon_t\|^2 + \frac{4\eta}{\lambda} \|\mathbb{E}_t[\hat{\nabla}\phi(x_t, y_t^*; \bar{\xi}_t)] - \nabla\Phi(x_t)\|^2,
\end{aligned}
$$

where the second inequality is due to Lemma D.2 and definition of $L$ in (41); the third inequality uses (57) and (56); the fourth inequality is due to Young's inequality $a^\top A b \leq \frac{1}{3}\|a\|_A^2 + \frac{3}{4}\|b\|_A^2$ and $\|a+b\|^2 \leq 2\|a\|_A^2 + 2\|b\|_A^2$ for any PSD matrix $A$; the fifth inequality uses the choice of $\eta \leq \lambda/6L$; the second last inequality is due to (56); and the last inequality uses (57) and Young's inequality. $\qquad\square$

The following lemma is essential for bounding the sum of the error terms $\|\epsilon_t\|^2$ before time $\tau$. Since we introduce $\mathbb{E}_t[\hat{\nabla}\phi(x_t, y_t^*; \bar{\xi}_t)]$ as part of the definition of $\epsilon_t$ (see (57)), we can directly invoke (Li et al., 2023a, Lemma C.10) to obtain the high probability bound.

**Lemma D.10** ((Li et al., 2023a, Lemma C.10))**.** *Denote $w_t$ as*

$$
w_{t-1} = (1 - \alpha_t)(\epsilon_{t-1} + \mathbb{E}_{t-1}[\hat{\nabla}\phi(x_{t-1}, y_{t-1}^*; \bar{\xi}_{t-1})] - \mathbb{E}_t[\hat{\nabla}\phi(x_t, y_t^*; \bar{\xi}_t)]).
$$

*Under the parameter choices in Theorem D.12, for any given $\delta \in (0, 1)$, the following holds with probability at least $1 - \delta/4$ over the randomness in $\mathcal{F}_{T+1}$ (we denote this event as $\mathcal{E}_x$):*

$$
\sum_{t=2}^{\tau} \alpha_t \langle w_{t-1}, \hat{\nabla}\phi(x_t, y_t^*; \bar{\xi}_t) - \mathbb{E}_t[\hat{\nabla}\phi(x_t, y_t^*; \bar{\xi}_t)] \rangle \leq 5\sigma_\phi^2 \sqrt{(1 + \beta^2 T)\ln(4/\delta)}.
$$

The next lemma bounds the sum of the error terms $\|\epsilon_t\|^2$ before time $\tau$.

**Lemma D.11.** *Under event $\mathcal{E}_0 \cap \mathcal{E}_y \cap \mathcal{E}_x$ and the parameter choices in Lemma D.4, we have*

$$
\sum_{t=1}^{\tau-1} \|\epsilon_t\|^2 - \frac{\lambda}{128G} \|\nabla\Phi(x_t)\|^2 \leq 8\sigma_\phi^2(1/\beta + \beta T) + 20\sigma_\phi^2 \sqrt{(1/\beta^2 + T)\ln(4/\delta)}
$$

$$
+ \frac{\lambda}{128G} \sum_{t=1}^{\tau-1} \|\hat{m}_t - \hat{u}_t\|^2 + \|\mathbb{E}_t[\hat{\nabla}\phi(x_t, y_t^*; \bar{\xi}_t)] - \nabla\Phi(x_t)\|^2.
$$

(58)

*Proof of Lemma D.11.* We first denote $w_t$ as

$$
w_{t-1} = (1 - \alpha_t)(\epsilon_{t-1} + \mathbb{E}_{t-1}[\hat{\nabla}\phi(x_{t-1}, y_{t-1}^*; \bar{\xi}_{t-1})] - \mathbb{E}_t[\hat{\nabla}\phi(x_t, y_t^*; \bar{\xi}_t)]).
$$

By definition of $\epsilon_t$ and the update rule (7), we have

$$
\begin{aligned}
\epsilon_t &= (1 - \alpha_t)(\epsilon_{t-1} + \mathbb{E}_{t-1}[\hat{\nabla}\phi(x_{t-1}, y_{t-1}^*; \bar{\xi}_{t-1})] - \mathbb{E}_t[\hat{\nabla}\phi(x_t, y_t^*; \bar{\xi}_t)]) \\
&\quad + \alpha_t(\hat{\nabla}\phi(x_t, y_t^*; \bar{\xi}_t) - \mathbb{E}_t[\hat{\nabla}\phi(x_t, y_t^*; \bar{\xi}_t)]) \\
&= w_{t-1} + \alpha_t(\hat{\nabla}\phi(x_t, y_t^*; \bar{\xi}_t) - \mathbb{E}_t[\hat{\nabla}\phi(x_t, y_t^*; \bar{\xi}_t)]).
\end{aligned}
$$

(59)

By choice of $\eta$ we have

$$\eta \le \frac{c_2 r \lambda}{G} \le \frac{2c_2 r}{D} \le \frac{r}{D},$$

where in the last inequality we choose small enough $c_2$. By Lemma D.8 we have $\|x_t - x_{t-1}\| \le r$ if $t \le \tau$. Then for $2 \le t \le \tau$, we apply Lemma D.6 to obtain

$$\|\mathbb{E}_{t-1}[\hat{\nabla}\phi(x_{t-1}, y_{t-1}^*; \bar{\xi}_{t-1})] - \mathbb{E}_t[\hat{\nabla}\phi(x_t, y_t^*; \bar{\xi}_t)]\|$$

$$\le L\|x_{t-1} - x_t\| \le \frac{\eta L}{\lambda}\|\hat{m}_{t-1}\| \le \frac{\eta L}{\lambda}(\|\nabla\Phi(x_{t-1})\| + \|\hat{\epsilon}_{t-1}\|) \tag{60}$$

$$\le \frac{\eta L}{\lambda}\left(\|\nabla\Phi(x_{t-1})\| + \|\hat{m}_{t-1} - \hat{u}_{t-1}\| + \|\epsilon_{t-1}\| + \|\mathbb{E}_{t-1}[\hat{\nabla}\phi(x_{t-1}, y_{t-1}^*; \bar{\xi}_{t-1})] - \nabla\Phi(x_{t-1})\|\right),$$

where the third inequality uses (57). Hence we have

$$\|w_{t-1}\|^2 = \|(1-\alpha_t)(\epsilon_{t-1} + \mathbb{E}_{t-1}[\hat{\nabla}\phi(x_{t-1}, y_{t-1}^*; \bar{\xi}_{t-1})] - \mathbb{E}_t[\hat{\nabla}\phi(x_t, y_t^*; \bar{\xi}_t)])\|^2$$

$$\le (1-\alpha_t)^2(1+\alpha_t)\|\epsilon_{t-1}\|^2$$

$$+ (1-\alpha_t)^2\left(1 + \frac{1}{\alpha_t}\right)\|\mathbb{E}_{t-1}[\hat{\nabla}\phi(x_{t-1}, y_{t-1}^*; \bar{\xi}_{t-1})] - \mathbb{E}_t[\hat{\nabla}\phi(x_t, y_t^*; \bar{\xi}_t)])\|^2$$

$$\le (1-\alpha_t)\|\epsilon_{t-1}\|^2 + \frac{1}{\alpha_t}\|\mathbb{E}_{t-1}[\hat{\nabla}\phi(x_{t-1}, y_{t-1}^*; \bar{\xi}_{t-1})] - \mathbb{E}_t[\hat{\nabla}\phi(x_t, y_t^*; \bar{\xi}_t)])\|^2$$

$$\le (1-\alpha_t)\|\epsilon_{t-1}\|^2 + \frac{4\eta^2 L^2}{\lambda^2 \beta}\left(\|\nabla\Phi(x_{t-1})\|^2 + \|\epsilon_{t-1}\|^2\right) \tag{61}$$

$$+ \frac{4\eta^2 L^2}{\lambda^2 \beta}\left(\|\hat{m}_{t-1} - \hat{u}_{t-1}\|^2 + \|\mathbb{E}_{t-1}[\hat{\nabla}\phi(x_{t-1}, y_{t-1}^*; \bar{\xi}_{t-1})] - \nabla\Phi(x_{t-1})\|^2\right)$$

$$\le \left(1 - \frac{\alpha_t}{2}\right)\|\epsilon_{t-1}\|^2 + \frac{\lambda\beta}{256G}\|\nabla\Phi(x_{t-1})\|^2$$

$$+ \frac{\lambda\beta}{256G}\left(\|\hat{m}_{t-1} - \hat{u}_{t-1}\|^2 + \|\mathbb{E}_{t-1}[\hat{\nabla}\phi(x_{t-1}, y_{t-1}^*; \bar{\xi}_{t-1})] - \nabla\Phi(x_{t-1})\|^2\right),$$

where the first inequality uses Young's inequality $\|a+b\|^2 \le (1+c)\|a\|^2 + (1+1/c)\|b\|^2$ for any $c > 0$; the second inequality is due to

$$(1-\alpha_t)^2(1+\alpha_t) \le (1-\alpha_t)(1-\alpha_t^2) \le 1-\alpha_t,$$

$$(1-\alpha_t)^2\left(1 + \frac{1}{\alpha_t}\right) = \frac{1}{\alpha_t}(1-\alpha_t)^2(1+\alpha_t) \le \frac{1}{\alpha}(1-\alpha_t) \le \frac{1}{\alpha_t};$$

the third inequality uses (60) and Young's inequality; and the last inequality is due to the choice of $G$ and $\eta$ with small enough $c_2$:

$$\eta \le \frac{c_2 \lambda^{3/2}\beta}{L\sqrt{G}} \le \frac{\lambda^{3/2}\beta}{32L\sqrt{G}} \quad \Longrightarrow \quad \frac{4\eta^2 L^2}{\lambda^2 \beta} \le \frac{\lambda\beta}{256G} \le \frac{\beta}{256} \le \frac{\beta}{2} \le \frac{\alpha_t}{2}.$$

Plugging (61) back into (59) gives

$$\|\epsilon_t\|^2 = \|w_{t-1}\|^2 + 2\alpha_t\langle w_{t-1}, \hat{\nabla}\phi(x_t, y_t^*; \bar{\xi}_t) - \mathbb{E}_t[\hat{\nabla}\phi(x_t, y_t^*; \bar{\xi}_t)]\rangle$$

$$+ \alpha_t^2\|\hat{\nabla}\phi(x_t, y_t^*; \bar{\xi}_t) - \mathbb{E}_t[\hat{\nabla}\phi(x_t, y_t^*; \bar{\xi}_t)]\|^2$$

$$\le \left(1 - \frac{\alpha_t}{2}\right)\|\epsilon_{t-1}\|^2 + \frac{\lambda\beta}{256G}\|\nabla\Phi(x_{t-1})\|^2 + \alpha_t^2\sigma_\phi^2$$

$$+ 2\alpha_t\langle \nu_{t-1}, \hat{\nabla}\phi(x_t, y_t^*; \bar{\xi}_t) - \mathbb{E}_t[\hat{\nabla}\phi(x_t, y_t^*; \bar{\xi}_t)]\rangle$$

$$+ \frac{\lambda\beta}{256G}\left(\|\hat{m}_{t-1} - \hat{u}_{t-1}\|^2 + \|\mathbb{E}_{t-1}[\hat{\nabla}\phi(x_{t-1}, y_{t-1}^*; \bar{\xi}_{t-1})] - \nabla\Phi(x_{t-1})\|^2\right).$$

Rearranging the above inequality, for any $2 \le t \le \tau$, we have

$$\frac{\beta}{2}\|\epsilon_{t-1}\|^2 \le \frac{\alpha_t}{2}\|\epsilon_{t-1}\|^2 \le \|\epsilon_{t-1}\|^2 - \|\epsilon_t\|^2 + \frac{\lambda\beta}{256G}\|\nabla\Phi(x_{t-1})\|^2$$
$$+ \alpha_t^2\sigma_\phi^2 + 2\alpha_t\langle\nu_{t-1}, \hat{\nabla}\phi(x_t, y_t^*; \bar{\xi}_t) - \mathbb{E}_t[\hat{\nabla}\phi(x_t, y_t^*; \bar{\xi}_t)]\rangle$$
$$+ \frac{\lambda\beta}{256G}\left(\|\hat{m}_{t-1} - \hat{u}_{t-1}\|^2 + \|\mathbb{E}_{t-1}[\hat{\nabla}\phi(x_{t-1}, y_{t-1}^*; \bar{\xi}_{t-1})] - \nabla\Phi(x_{t-1})\|^2\right).$$

Then taking summation over $t$ from 2 to $\tau$ we obtain that

$$\sum_{t=2}^{\tau}\frac{\beta}{2}\|\epsilon_{t-1}\|^2 - \frac{\lambda\beta}{256G}\|\nabla\Phi(x_{t-1})\|^2$$

$$\le \|\epsilon_1\|^2 - \|\epsilon_\tau\|^2 + \sigma_\phi^2\sum_{t=2}^{\tau}\alpha_t^2 + 2\sum_{t=2}^{\tau}\alpha_t\langle w_{t-1}, \hat{\nabla}\phi(x_t, y_t^*; \bar{\xi}_t) - \mathbb{E}_t[\hat{\nabla}\phi(x_t, y_t^*; \bar{\xi}_t)]\rangle$$

$$+ \frac{\lambda\beta}{256G}\sum_{t=2}^{\tau}\|\hat{m}_{t-1} - \hat{u}_{t-1}\|^2 + \|\mathbb{E}_{t-1}[\hat{\nabla}\phi(x_{t-1}, y_{t-1}^*; \bar{\xi}_{t-1})] - \nabla\Phi(x_{t-1})\|^2$$

$$\le 4\sigma_\phi^2(1 + \beta^2 T) + 10\sigma_\phi^2\sqrt{(1 + \beta^2 T)\ln(4/\delta)}$$

$$+ \frac{\lambda\beta}{256G}\sum_{t=2}^{\tau}\|\hat{m}_{t-1} - \hat{u}_{t-1}\|^2 + \|\mathbb{E}_{t-1}[\hat{\nabla}\phi(x_{t-1}, y_{t-1}^*; \bar{\xi}_{t-1})] - \nabla\Phi(x_{t-1})\|^2,$$

where the last inequality uses Lemmas B.7 and D.10 and the fact that $\|\epsilon_1\|^2 \le \sigma_\phi^2$. Then we complete the proof by multiplying both sides by $2/\beta$. $\qquad\square$

With Lemmas D.9 and D.11, we are ready to prove Theorem 4.1. Below is the full statement of Theorem 4.1 with detailed parameter choices, where we use $c_1, c_2, c_3$ to denote small enough constants and $C_1, C_2$ to denote large enough ones. The definitions of problem-dependent constants $\sigma_\phi, C_{\phi,0}, C_{\phi,1}, \Delta_1, L_0, L_1, L, C_\beta$ are provided in Appendix D.1.

**Theorem D.12** (Restatement of Theorem 4.1). *Suppose that Assumptions 3.2 to 3.4 hold. Let $G$ be a constant satisfying*

$$G \ge \max\left\{4\lambda, 2\sigma_\phi, 4C_{\phi,0}, \frac{C_{\phi,1}}{L_1}, \sqrt{\frac{C_1\Delta_1 L_0}{C_L}}, \frac{C_1\Delta_1 L_1}{C_L}\right\}, \tag{62}$$

*Given any $\epsilon > 0$ and $\delta \in (0, 1)$, denote $\iota := \ln(4/\delta)$, and choose*

$$0 \le \beta_{\mathrm{sq}} \le 1, \quad \beta \le \min\left\{1, \frac{c_1\lambda\epsilon^2}{\sigma_\phi^2 G\max\{1, \sqrt{\iota}, \ln(C_\beta)\}}\right\}, \quad \gamma = \frac{2\beta}{\mu}, \tag{63}$$

$$\eta \le c_2\min\left\{\frac{r\lambda}{G}, \frac{\lambda}{6L}, \frac{\sigma_\phi\lambda\beta}{LG\max\{1, \sqrt{\iota}, \ln(1/\beta), \ln(C_\beta)\}}, \frac{\lambda^{3/2}\beta}{L\sqrt{G}}\right\}, \tag{64}$$

$$Q \ge \frac{1}{2}\max\left\{\ln\beta\Big/\ln\left(1 - \frac{\mu}{l_{g,1}}\right), \ln\left(\frac{c_3\lambda\mu^2\epsilon^2}{Gl_{g,1}^2 l_{f,0}^2}\right)\Big/\ln\left(1 - \frac{\mu}{l_{g,1}}\right)\right\}, \tag{65}$$

$$T_0 = \ln\left(\frac{\sigma_{g,1}^2\beta}{\mu^2\|y_0 - y_0^*\|^2}\right)\Big/\ln(1 - \beta), \quad T = \max\left\{\frac{1}{\beta^2}, \frac{C_2\Delta_1 G}{\eta\epsilon^2}\right\}, \tag{66}$$

*where constant $C_\beta$ is defined as*

$$C_\beta \ge \max\left\{\frac{8e\sigma_\phi^4 G^2\max\{1, \iota\}}{c_1^2\delta\lambda^2\epsilon^4}, \frac{8C_2 e\Delta_1 L\sigma_\phi G^3}{c_1 c_2\delta\lambda^2\epsilon^4}\left(1 + \frac{\sigma_\phi^2 G}{c_1\lambda\epsilon^2}\right)\max\{1, \sqrt{\iota}, \iota\},\right.$$
$$\left.\left(\frac{32e\sigma_\phi^4 G^2}{c_1^2\delta\lambda^2\epsilon^4}\right)^2, \left(\frac{48C_2 e\Delta_1 L\sigma_\phi G^3}{c_1 c_2\delta\lambda^2\epsilon^4}\left(1 + \frac{\sigma_\phi^2 G}{c_1\lambda\epsilon^2}\right)\max\{1, \sqrt{\iota}, \iota\}\right)^2\right\}.$$

*Run Algorithm 1 for $T$ iterations. Then with probability at least $1-\delta$ over the randomness in $\mathcal{F}_{T+1}$, we have $\|\nabla\Phi(x_t)\| \leq G$ for all $t \in [T]$, and $\frac{1}{T}\sum_{t=1}^{T}\|\nabla\Phi(x_t)\| \leq \epsilon^2$.*

*Proof of Theorem D.12.* By Lemmas D.3, D.4 and D.10, we have $\Pr(\mathcal{E}_0 \cap \mathcal{E}_y \cap \mathcal{E}_x) \geq 1 - 3\delta/4 \geq 1 - \delta$. The following analysis is conditioned on the event $\mathcal{E}_0 \cap \mathcal{E}_y \cap \mathcal{E}_x$.

Rearranging (55) of Lemma D.9 and telescoping over $t$ from 1 to $\tau-1$, we have

$$
\begin{aligned}
\sum_{t=1}^{\tau-1} 4\|\nabla\Phi(x_t)\|^2 - \frac{64G}{\lambda}\|\epsilon_t\|^2 \leq {} & \frac{16G}{\eta}[(\Phi(x_1) - \Phi^*) - (\Phi(x_\tau) - \Phi^*)] \\
& + \frac{32G}{\lambda}\sum_{t=1}^{\tau-1}\|\hat{m}_t - \hat{u}_t\|^2 + 2\|\mathbb{E}_t[\hat{\nabla}\phi(x_t, y_t^*; \bar{\xi}_t)] - \nabla\Phi(x_t)\|^2.
\end{aligned}
\tag{67}
$$

Also, (58) of Lemma D.11 can be written as

$$
\begin{aligned}
\sum_{t=1}^{\tau-1} \frac{128G}{\lambda}\|\epsilon_t\|^2 - \|\nabla\Phi(x_t)\|^2 \leq {} & \frac{128G}{\lambda}\left(8\sigma_\phi^2(1/\beta + \beta T) + 20\sigma_\phi^2\sqrt{(1/\beta^2 + T)\ln(4/\delta)}\right) \\
& + \sum_{t=1}^{\tau-1}\|\hat{m}_t - \hat{u}_t\|^2 + \|\mathbb{E}_t[\hat{\nabla}\phi(x_t, y_t^*; \bar{\xi}_t)] - \nabla\Phi(x_t)\|^2.
\end{aligned}
\tag{68}
$$

Summing (67) and (68) and rearranging gives

$$
\begin{aligned}
& \frac{16G}{\eta}(\Phi(x_\tau) - \Phi^*) + 3\sum_{t=1}^{\tau-1}\|\nabla\Phi(x_t)\|^2 + \frac{64G}{\lambda}\sum_{t=1}^{\tau-1}\|\epsilon_t\|^2 \\
& \leq \frac{16G}{\eta\lambda}\left(\lambda\Delta_1 + 64\sigma_\phi^2\left(\frac{\eta}{\beta} + \eta\beta T\right) + 160\eta\sigma_\phi^2\sqrt{(1/\beta^2 + T)\ln(4/\delta)}\right) \\
& \quad + \left(1 + \frac{32G}{\lambda}\right)\sum_{t=1}^{\tau-1}\|\hat{m}_t - \hat{u}_t\|^2 + \left(1 + \frac{64G}{\lambda}\right)\sum_{t=1}^{\tau-1}\|\mathbb{E}_t[\hat{\nabla}\phi(x_t, y_t^*; \bar{\xi}_t)] - \nabla\Phi(x_t)\|^2 \\
& \leq \frac{16G}{\eta\lambda}\left(\lambda\Delta_1 + 64\sigma_\phi^2\left(\frac{\eta}{\beta} + \eta\beta T\right) + 160\eta\sigma_\phi^2\sqrt{(1/\beta^2 + T)\ln(4/\delta)}\right) \\
& \quad + \frac{33G}{\lambda}\sum_{t=1}^{\tau-1}\|\hat{m}_t - \hat{u}_t\|^2 + \frac{65G}{\lambda}\sum_{t=1}^{\tau-1}\|\mathbb{E}_t[\hat{\nabla}\phi(x_t, y_t^*; \bar{\xi}_t)] - \nabla\Phi(x_t)\|^2,
\end{aligned}
$$

where the last inequality uses $G \geq \lambda$. By Lemma D.7, we further have

$$
\begin{aligned}
& \frac{16G}{\eta}(\Phi(x_\tau) - \Phi^*) + 3\sum_{t=1}^{\tau-1}\|\nabla\Phi(x_t)\|^2 + \frac{64G}{\lambda}\sum_{t=1}^{\tau-1}\|\epsilon_t\|^2 \\
& \leq \frac{16G}{\eta\lambda}\left(\lambda\Delta_1 + 64\sigma_\phi^2\left(\frac{\eta}{\beta} + \eta\beta T\right) + 160\eta\sigma_\phi^2\sqrt{(1/\beta^2 + T)\ln(4/\delta)}\right) \\
& \quad + \frac{33L^2 GT}{\lambda}\left(\left(1 + \frac{8\eta^2 l_{g,1}^2 L^2}{\lambda^2\mu^4\gamma^2}\right)\|y_1 - y_1^*\|^2 + \left(\frac{8\gamma}{\mu}\ln\frac{4eT}{\delta} + \frac{32\eta^2 l_{g,1}^2 L^2}{\lambda^2\mu^5\gamma}\right)\sigma_{g,1}^2\right) \\
& \quad + \frac{33L^2 G}{\lambda}\left(\frac{8\eta^2 l_{g,1}^2}{\lambda^2\mu^4\gamma^2} + \frac{2048\eta^4 l_{g,1}^4 L^2}{\lambda^4\mu^8\gamma^4}\left(2 + \ln\frac{1}{\beta}\right)\right)\sum_{t=1}^{\tau-1}\|\epsilon_t\|^2 + 2\|\mathbb{E}_t[\hat{\nabla}\phi(x_t, y_t^*; \bar{\xi}_t)] - \nabla\Phi(x_t)\|^2 \\
& \quad + \frac{66L^2 G}{\lambda}\left(\frac{8\eta^2 l_{g,1}^2}{\lambda^2\mu^4\gamma^2} + \frac{2048\eta^4 l_{g,1}^4 L^2}{\lambda^4\mu^8\gamma^4}\left(2 + \ln\frac{1}{\beta}\right)\right)\sum_{t=1}^{\tau-1}\|\nabla\Phi(x_t)\|^2 \\
& \quad + \frac{65G}{\lambda}\sum_{t=1}^{\tau-1}\|\mathbb{E}_t[\hat{\nabla}\phi(x_t, y_t^*; \bar{\xi}_t)] - \nabla\Phi(x_t)\|^2.
\end{aligned}
\tag{69}
$$

By Lemma D.13, we know that

$$\frac{66L^2G}{\lambda}\left(\frac{8\eta^2 l_{g,1}^2}{\lambda^2\mu^4\gamma^2}+\frac{2048\eta^4 l_{g,1}^4 L^2}{\lambda^4\mu^8\gamma^4}\left(2+\ln\frac{1}{\beta}\right)\right)\le 2,$$

Then with $G\ge\lambda$, (69) can be simplified as

$$
\begin{aligned}
\frac{16G}{\eta}&(\Phi(x_\tau)-\Phi^*)+\sum_{t=1}^{\tau-1}\|\nabla\Phi(x_t)\|^2\\
&\le\frac{16G}{\eta\lambda}\left(\lambda\Delta_1+64\sigma_\phi^2\left(\frac{\eta}{\beta}+\eta\beta T\right)+160\eta\sigma_\phi^2\sqrt{(1/\beta^2+T)\ln(4/\delta)}\right)\\
&\quad+\frac{33L^2GT}{\lambda}\left(\left(1+\frac{8\eta^2 l_{g,1}^2 L^2}{\lambda^2\mu^4\gamma^2}\right)\|y_1-y_1^*\|^2+\left(\frac{8\gamma}{\mu}\ln\frac{4eT}{\delta}+\frac{32\eta^2 l_{g,1}^2 L^2}{\lambda^2\mu^5\gamma}\right)\sigma_{g,1}^2\right)\\
&\quad+\frac{67GTl_{g,1}^2 l_{f,0}^2}{\lambda\mu^2}\left(1-\frac{\mu}{l_{g,1}}\right)^{2Q}=:I_1.
\end{aligned}
\tag{70}
$$

By definition of $\tau$ in (40), we have

$$\frac{16G}{\eta}(\Phi(x_\tau)-\Phi^*)>\frac{16G\psi}{\eta}=\frac{8C_L G^3}{\eta L}=:I_2.$$

By Lemma D.14, we have $I_1\le I_2$, which leads to a contradiction. Thus, we must have $\tau=T+1$ conditioned on $\mathcal{E}_0\cap\mathcal{E}_y\cap\mathcal{E}_x$. Therefore, combining (70) and Lemma D.14 finally yields that under event $\mathcal{E}_0\cap\mathcal{E}_y\cap\mathcal{E}_x$,

$$\frac{1}{T}\sum_{t=1}^{T}\|\nabla\Phi(x_t)\|^2\le\epsilon^2.$$

Moreover, since $\tau=T+1$, then by Lemma D.2 we can replace $\hat{L}_T$ and $G_T$ with $L$ and $G$ respectively, in the additional requirement (48) for $\eta$. Therefore, (48) is now included in the parameter choices of Theorem D.12, which indicates that the current parameter choices are sufficient. $\qquad\square$

## D.8 Parameter Choices for AdamBO (Theorem D.12)

The following two lemmas, Lemmas D.13 and D.14, hide complicate calculations and will be useful in the contradiction argument and upper-level convergence analysis.

**Lemma D.13.** *Under the parameter choices in Theorem D.12, we have the following facts:*

$$\ln\frac{4eT}{\delta}\le\ln(C_\beta),\quad \|y_1-y_1^*\|^2\le\frac{17\beta\sigma_{g,1}^2}{\mu^2}\ln(C_\beta),\tag{71}$$

$$\varrho\le\min\left\{r,\frac{1}{4L_1}\right\},\quad C_{u,0}+C_{u,1}\varrho+\lambda\le 2G,\tag{72}$$

$$\frac{66L^2G}{\lambda}\left(\frac{8\eta^2 l_{g,1}^2}{\lambda^2\mu^4\gamma^2}+\frac{2048\eta^4 l_{g,1}^4 L^2}{\lambda^4\mu^8\gamma^4}\left(2+\ln\frac{1}{\beta}\right)\right)\le 2\tag{73}$$

*Proof of Lemma D.13.* We first list all the relevant parameter choices below for convenience:

$$G\ge\max\left\{4\lambda,2\sigma_\phi,4C_{\phi,0},\frac{C_{\phi,1}}{L_1},\sqrt{\frac{C_1\Delta_1 L_0}{C_L}},\frac{C_1\Delta_1 L_1}{C_L}\right\},$$

$$\beta\le\min\left\{1,\frac{c_1\lambda\epsilon^2}{\sigma_\phi^2 G\max\{1,\sqrt{\iota},\ln(C_\beta)\}}\right\},\quad\gamma=\frac{2\beta}{\mu},$$

$$\eta \leq c_2 \min \left\{ \frac{r\lambda}{G}, \frac{\sigma_\phi \lambda \beta}{LG \max\{1, \sqrt{\iota}, \ln(1/\beta), \ln(C_\beta)\}}, \frac{\lambda^{3/2}\beta}{L\sqrt{G}} \right\},$$

$$Q \geq \frac{1}{2} \max \left\{ \ln \beta \Big/ \ln \left( 1 - \frac{\mu}{l_{g,1}} \right), \ln \left( \frac{c_3 \lambda \mu^2 \epsilon^2}{G l_{g,1}^2 l_{f,0}^2} \right) \Big/ \ln \left( 1 - \frac{\mu}{l_{g,1}} \right) \right\},$$

$$T_0 = \ln \left( \frac{\sigma_{g,1}^2 \beta}{\mu^2 \|y_0 - y_0^*\|^2} \right) \Big/ \ln(1 - \beta), \quad T = \max \left\{ \frac{1}{\beta^2}, \frac{C_2 \Delta_1 G}{\eta \epsilon^2} \right\},$$

where $C_\beta$ is defined in (44). Now we verify the above listed facts one by one.

**Verification for** (71)**:** $\ln(4eT/\delta) \leq \ln(C_\beta)$. We focus on the dominant terms for each parameter choice when $\epsilon$ is sufficiently small. For the remaining cases, the result can be easily obtained by following the same procedure. Specifically, we consider the case where $\beta, \eta$ and $T$ are chosen as

$$\beta = \frac{c_1 \lambda \epsilon^2}{\sigma_\phi^2 G \max\{1, \sqrt{\iota}, \ln(C_\beta)\}}, \quad \eta = \frac{c_2 \sigma_\phi \lambda \beta}{LG \max\{1, \sqrt{\iota}, \ln(1/\beta), \ln(C_\beta)\}}, \quad T = \max \left\{ \frac{1}{\beta^2}, \frac{C_2 \Delta_1 G}{\eta \epsilon^2} \right\}.$$

(Case 1) If $T = 1/\beta^2$, then we have

$$
\begin{aligned}
\ln \frac{4eT}{\delta} &= \ln \frac{4e}{\delta \beta^2} \\
&= \ln \left( \frac{4e\sigma_\phi^4 G^2 \max\{1, \iota, \ln^2(C_\beta)\}}{c_1^2 \delta \lambda^2 \epsilon^4} \right) \leq \ln \left( \frac{4e\sigma_\phi^4 G^2 (\max\{1, \iota\} + \ln^2(C_\beta))}{c_1^2 \delta \lambda^2 \epsilon^4} \right) \\
&\leq \ln \left( \frac{4e\sigma_\phi^4 G^2 (\max\{1, \iota\} + 4C_\beta^{1/2})}{c_1^2 \delta \lambda^2 \epsilon^4} \right) \leq \ln(C_\beta),
\end{aligned}
$$

where the second inequality uses $\ln x \leq 2x^{1/4}$ for $x > 0$, and the last inequality is due to

$$\frac{4e\sigma_\phi^4 G^2 \max\{1, \iota\}}{c_1^2 \delta \lambda^2 \epsilon^4} \leq \frac{C_\beta}{2} \qquad \text{and} \qquad \frac{16e\sigma_\phi^4 G^2}{c_1^2 \delta \lambda^2 \epsilon^4} C_\beta^{1/2} \leq \frac{C_\beta}{2}$$

since

$$C_\beta \geq \max \left\{ \frac{8e\sigma_\phi^4 G^2 \max\{1, \iota\}}{c_1^2 \delta \lambda^2 \epsilon^4}, \left( \frac{32e\sigma_\phi^4 G^2}{c_1^2 \delta \lambda^2 \epsilon^4} \right)^2 \right\}.$$

(Case 2) If $T = \frac{C_2 \Delta_1 G}{\eta \epsilon^2}$, then we have

$$
\begin{aligned}
\ln \frac{4eT}{\delta} &= \ln \left( \frac{4C_2 e \Delta_1 L \sigma_\phi G^3 \max\{1, \sqrt{\iota}, \ln(1/\beta), \ln(C_\beta)\} \max\{1, \sqrt{\iota}, \ln(C_\beta)\}}{c_1 c_2 \delta \lambda^2 \epsilon^4} \right) \\
&= \ln \left( \frac{4C_2 e \Delta_1 L \sigma_\phi G^3 (\max\{1, \sqrt{\iota}\} + \ln(1/\beta) + \ln(C_\beta))(\max\{1, \sqrt{\iota}\} + \ln(C_\beta))}{c_1 c_2 \delta \lambda^2 \epsilon^4} \right).
\end{aligned}
$$

(74)

Also note that

$$
\begin{aligned}
\ln \frac{1}{\beta} &= \ln \left( \frac{\sigma_\phi^2 G \max\{1, \sqrt{\iota}, \ln(C_\beta)\}}{c_1 \lambda \epsilon^2} \right) \leq \ln \left( \frac{\sigma_\phi^2 G (\max\{1, \sqrt{\iota}\} + \ln(C_\beta))}{c_1 \lambda \epsilon^2} \right) \\
&\leq \frac{\sigma_\phi^2 G (\max\{1, \sqrt{\iota}\} + \ln(C_\beta))}{c_1 \lambda \epsilon^2}.
\end{aligned}
$$

Then we obtain

$$
\begin{aligned}
(\max\{1, & \sqrt{\iota}\} + \ln(1/\beta) + \ln(C_\beta))(\max\{1, \sqrt{\iota}\} + \ln(C_\beta)) \\
&\leq \left( \max\{1, \sqrt{\iota}\} + \frac{\sigma_\phi^2 G(\max\{1, \sqrt{\iota}\} + \ln(C_\beta))}{c_1 \lambda \epsilon^2} + \ln(C_\beta) \right) (\max\{1, \sqrt{\iota}\} + \ln(C_\beta)) \\
&= \left( 1 + \frac{\sigma_\phi^2 G}{c_1 \lambda \epsilon^2} \right) (\max\{1, \iota\} + 2 \max\{1, \sqrt{\iota}\} \ln(C_\beta) + \ln^2(C_\beta)) \\
&\leq \left( 1 + \frac{\sigma_\phi^2 G}{c_1 \lambda \epsilon^2} \right) (\max\{1, \iota\} + 2 \max\{1, \sqrt{\iota}\} C_\beta^{1/2} + 4 C_\beta^{1/2}) \\
&\leq \left( 1 + \frac{\sigma_\phi^2 G}{c_1 \lambda \epsilon^2} \right) (\max\{1, \iota\} + 6 \max\{1, \sqrt{\iota}, \iota\} C_\beta^{1/2}) \\
&\leq \left( 1 + \frac{\sigma_\phi^2 G}{c_1 \lambda \epsilon^2} \right) \max\{1, \sqrt{\iota}, \iota\} \left( 1 + 6 C_\beta^{1/2} \right)
\end{aligned}
\tag{75}
$$

where the second inequality uses $\ln x \leq x^{1/2}$ and $\ln x \leq 2x^{1/4}$ for $x > 0$. Thus, plugging (75) back into (74) and we have

$$
\begin{aligned}
\ln \frac{4eT}{\delta} &= \ln \left( \frac{4 C_2 e \Delta_1 L \sigma_\phi G^3 (\max\{1, \sqrt{\iota}\} + \ln(1/\beta))(\max\{1, \sqrt{\iota}\} + \ln(C_\beta))}{c_1 c_2 \delta \lambda^2 \epsilon^4} \right) \\
&\leq \ln \left( \frac{4 C_2 e \Delta_1 L \sigma_\phi G^3}{c_1 c_2 \delta \lambda^2 \epsilon^4} \left( 1 + \frac{\sigma_\phi^2 G}{c_1 \lambda \epsilon^2} \right) \max\{1, \sqrt{\iota}, \iota\} \left( 1 + 6 C_\beta^{1/2} \right) \right) \\
&\leq \ln(C_\beta),
\end{aligned}
$$

where the last inequality is due to

$$
\frac{4 C_2 e \Delta_1 L \sigma_\phi G^3}{c_1 c_2 \delta \lambda^2 \epsilon^4} \left( 1 + \frac{\sigma_\phi^2 G}{c_1 \lambda \epsilon^2} \right) \max\{1, \sqrt{\iota}, \iota\} \leq \frac{C_\beta}{2}
$$

and

$$
\frac{24 C_2 e \Delta_1 L \sigma_\phi G^3}{c_1 c_2 \delta \lambda^2 \epsilon^4} \left( 1 + \frac{\sigma_\phi^2 G}{c_1 \lambda \epsilon^2} \right) \max\{1, \sqrt{\iota}, \iota\} C_\beta^{1/2} \leq \frac{C_\beta}{2}
$$

since

$$
C_\beta \geq \max \left\{ \frac{8 C_2 e \Delta_1 L \sigma_\phi G^3}{c_1 c_2 \delta \lambda^2 \epsilon^4} \left( 1 + \frac{\sigma_\phi^2 G}{c_1 \lambda \epsilon^2} \right) \max\{1, \sqrt{\iota}, \iota\}, \right.
$$
$$
\left. \left( \frac{48 C_2 e \Delta_1 L \sigma_\phi G^3}{c_1 c_2 \delta \lambda^2 \epsilon^4} \left( 1 + \frac{\sigma_\phi^2 G}{c_1 \lambda \epsilon^2} \right) \max\{1, \sqrt{\iota}, \iota\} \right)^2 \right\}.
$$

**Verification for** (71): $\|y_1 - y_1^*\|^2 \leq 17 \beta \sigma_{g,1}^2 \ln(C_\beta)/\mu^2$. By choice of $T_0$ and $\gamma$, we have

$$
\begin{aligned}
\|y_1 - y_1^*\|^2 &\leq \left( 1 - \frac{\mu \gamma}{2} \right)^{T_0} \|y_0 - y_0^*\|^2 + \frac{8 \gamma \sigma_{g,1}^2}{\mu} \ln \frac{4e}{\delta} \\
&\leq \frac{\beta \sigma_{g,1}^2}{\mu^2} + \frac{16 \beta \sigma_{g,1}^2}{\mu^2} \ln \frac{4e}{\delta} \\
&\leq \frac{17 \beta \sigma_{g,1}^2}{\mu^2} \ln(C_\beta),
\end{aligned}
$$

where the last inequality uses $T \geq 1/\beta^2 \geq 1$ and $\ln(4eT/\delta) \leq \ln(C_\beta)$.

**Verification for** (72): $\varrho \leq \min\{r, 1/4L_1\}$. By Lemma D.5 and choices of $\eta, \gamma$ and $\beta$, we have

$$
\begin{aligned}
\varrho^2 &= \left(1 + \frac{8\eta^2 l_{g,1}^2 L^2}{\lambda^2 \mu^4 \gamma^2}\right) \|y_1 - y_1^*\|^2 + \left(\frac{8\gamma}{\mu} \ln \frac{4eT}{\delta} + \frac{32\eta^2 l_{g,1}^2 L^2}{\lambda^2 \mu^5 \gamma}\right) \sigma_{g,1}^2 \\
&\quad + \frac{8\eta^2 l_{g,1}^2 C_{u,0}^2}{\lambda^2 \mu^4 \gamma^2} + \frac{1024 \eta^4 l_{g,1}^4 L^2 C_{u,0}^2}{\lambda^4 \mu^8 \gamma^4} \left(2 + \ln \frac{1}{\beta}\right) \\
&\leq \left(1 + \frac{2\eta^2 l_{g,1}^2 L^2}{\lambda^2 \mu^2 \beta^2}\right) \frac{17\beta \sigma_{g,1}^2}{\mu^2} \ln(C_\beta) + \left(\frac{16\beta}{\mu^2} \ln(C_\beta) + \frac{16\eta^2 l_{g,1}^2 L^2}{\lambda^2 \mu^4 \beta}\right) \sigma_{g,1}^2 \\
&\quad + \frac{2\eta^2 l_{g,1}^2 C_{u,0}^2}{\lambda^2 \mu^2 \beta^2} + \frac{64 \eta^4 l_{g,1}^4 L^2}{\lambda^4 \mu^4 \beta^4} \left(2 + \ln \frac{1}{\beta}\right) \\
&\leq \left(1 + \frac{2c_2^2 \sigma_\phi^2 l_{g,1}^2}{\mu^2 G^2}\right) \frac{17 c_1 \lambda \sigma_{g,1}^2 \epsilon^2}{\mu^2 \sigma_\phi^2 G} + \left(\frac{16 c_1 \lambda \epsilon^2}{\mu^2 \sigma_\phi^2 G} + \frac{16 c_1 c_2^2 \lambda l_{g,1}^2 \epsilon^2}{\mu^4 G^3}\right) \sigma_{g,1}^2 \\
&\quad + \frac{2c_2^2 \sigma_\phi^2 l_{g,1}^2 C_{u,0}^2}{\mu^2 L^2 G^2} + \frac{192 c_2^4 \sigma_\phi^4 l_{g,1}^4}{\mu^4 L^2 G^4} \leq \min\left\{r, \frac{1}{4L_1}\right\},
\end{aligned}
$$

where in the last inequality we choose small enough $c_1$ and $c_2$.

**Verification of** (72): $C_{u,0} + C_{u,1} \varrho + \lambda \leq 2G$. By definitions of $C_{u,0}, C_{u,1}$ in (42) and choice of $G$, we have

$$
\begin{aligned}
C_{u,0} + C_{u,1} \varrho + \lambda &= C_{\phi,0} + G + (C_{\phi,1} + L_1 G)\varrho + \lambda \\
&\leq \frac{G}{4} + G + \frac{G}{2} + \frac{G}{4} = G.
\end{aligned}
$$

**Verification for** (73). By choices of $\eta, \gamma$ and $\beta$, we have

$$
\begin{aligned}
&\frac{66 L^2 G}{\lambda} \left(\frac{8\eta^2 l_{g,1}^2}{\lambda^2 \mu^4 \gamma^2} + \frac{2048 \eta^4 l_{g,1}^4 L^2}{\lambda^4 \mu^8 \gamma^4} \left(2 + \ln \frac{1}{\beta}\right)\right) \\
&\leq \frac{66 L^2 G}{\lambda} \left(\frac{2\eta^2 l_{g,1}^2}{\lambda^2 \mu^2 \beta^2} + \frac{128 \eta^4 l_{g,1}^4 L^2}{\lambda^4 \mu^4 \beta^4} \left(2 + \ln \frac{1}{\beta}\right)\right) \leq \frac{66 L^2 G}{\lambda} \left(\frac{2c_2^2 \sigma_\phi^2 l_{g,1}^2}{\mu^2 L^2 G^2} + \frac{384 c_2^4 \sigma_\phi^4 l_{g,1}^4}{\mu^4 L^2 G^4}\right) \leq 2,
\end{aligned}
$$

where in the last inequality we choose small enough $c_2$. $\qquad\square$

**Lemma D.14.** *Denote $I_1$ and $I_2$ as*

$$
\begin{aligned}
I_1 &:= \frac{16G}{\eta\lambda} \left(\lambda \Delta_1 + 64 \sigma_\phi^2 \left(\frac{\eta}{\beta} + \eta\beta T\right) + 160 \eta \sigma_\phi^2 \sqrt{(1/\beta^2 + T) \ln(4/\delta)}\right) \\
&\quad + \frac{33 L^2 GT}{\lambda} \left(\left(1 + \frac{8\eta^2 l_{g,1}^2 L^2}{\lambda^2 \mu^4 \gamma^2}\right) \|y_1 - y_1^*\|^2 + \left(\frac{8\gamma}{\mu} \ln \frac{4eT}{\delta} + \frac{32\eta^2 l_{g,1}^2 L^2}{\lambda^2 \mu^5 \gamma}\right) \sigma_{g,1}^2\right) \\
&\quad + \frac{67 GT l_{g,1}^2 l_{f,0}^2}{\lambda \mu^2} \left(1 - \frac{\mu}{l_{g,1}}\right)^{2Q}, \\
I_2 &:= \frac{8 C_L G^3}{\eta L}.
\end{aligned}
$$
(76)

*For any given $\epsilon > 0$, under the parameter choice in Theorem D.12, we have $I_1 \leq I_2$ and $I_1/T \leq \epsilon^2$.*

*Proof of Lemma D.14.* We first verify $I_1 \leq I_2$ and then verify $I_1/T \leq \epsilon^2$.

**Proof of $I_1 \leq I_2$.** We start to show $I_1/I_2 \leq 1$. We have

$$
\frac{I_1}{I_2} \leq \frac{2L}{\lambda C_L G^2} \left( \lambda \Delta_1 + 64\sigma_\phi^2 \left( \frac{\eta}{\beta} + \eta\beta T \right) + 160\eta\sigma_\phi^2 \sqrt{(1/\beta^2 + T)\ln(4/\delta)} \right)
$$

$$
+ \frac{5L^3 \eta T}{\lambda C_L G^2} \left( \left( 1 + \frac{8\eta^2 l_{g,1}^2 L^2}{\lambda^2 \mu^4 \gamma^2} \right) \|y_1 - y_1^*\|^2 + \left( \frac{8\gamma}{\mu} \ln \frac{4eT}{\delta} + \frac{32\eta^2 l_{g,1}^2 L^2}{\lambda^2 \mu^5 \gamma} \right) \sigma_{g,1}^2 \right)
$$

$$
+ \frac{9 l_{g,1}^2 l_{f,0}^2 L \eta T}{\lambda \mu^2 C_L G^2} \left( 1 - \frac{\mu}{l_{g,1}} \right)^{2Q}
$$

$$
\leq \frac{\lambda \Delta_1}{8\lambda \Delta_1} + \frac{2L}{\lambda C_L G^2} \left( 64\sigma_\phi^2 \left( \frac{2\eta}{\beta} + \frac{C_2 \Delta_1 G\beta}{\epsilon^2} \right) + 160\sigma_\phi^2 \sqrt{\iota} \sqrt{\frac{5\eta^2}{2\beta^2} + \frac{1}{2} \left( \frac{C_2 \Delta_1 G\beta}{\epsilon^2} \right)^2} \right)
$$

$$
+ \frac{5L^3}{\lambda C_L G^2} \left( \frac{\eta}{\beta^2} + \frac{C_2 \Delta_1 G}{\epsilon^2} \right) \left( 1 + \frac{2\eta^2 l_{g,1}^2 L^2}{\lambda^2 \mu^2 \beta^2} \right) \frac{17\beta \sigma_{g,1}^2}{\mu^2} \ln(C_\beta)
$$

$$
+ \frac{5L^3}{\lambda C_L G^2} \left( \frac{\eta}{\beta^2} + \frac{C_2 \Delta_1 G}{\epsilon^2} \right) \left( \frac{16\beta}{\mu^2} \ln(C_\beta) + \frac{16\eta^2 l_{g,1}^2 L^2}{\lambda^2 \mu^4 \beta} \right) \sigma_{g,1}^2
$$

$$
+ \frac{9 l_{g,1}^2 l_{f,0}^2 L}{\lambda \mu^2 C_L G^2} \left( \frac{\eta}{\beta^2} + \frac{C_2 \Delta_1 G}{\epsilon^2} \right) \beta
$$

$$
\leq \frac{1}{8} + \frac{16L}{\lambda C_L G^2} \left( \frac{48 c_2 \lambda \sigma_\phi^3}{LG} + 24 c_1 C_2 \lambda \Delta_1 \right)
$$

$$
+ \frac{5L^3}{\lambda C_L G^2} \left( \frac{c_2 \sigma_\phi \lambda}{LG} + \frac{c_1 C_2 \lambda \Delta_1}{\sigma_\phi^2} \right) \left( 1 + \frac{2 c_2^2 \sigma_\phi^2 l_{g,1}^2}{\mu^2 G^2} \right) \frac{17 \sigma_{g,1}^2}{\mu^2}
$$

$$
+ \frac{5L^3}{\lambda C_L G^2} \left( \frac{c_2 \sigma_\phi \lambda}{LG} + \frac{c_1 C_2 \lambda \Delta_1}{\sigma_\phi^2} \right) \left( \frac{16}{\mu^2} + \frac{2 c_2^2 \sigma_\phi^2 l_{g,1}^2}{\mu^4 G^2} \right) \sigma_{g,1}^2
$$

$$
+ \frac{9 l_{g,1}^2 l_{f,0}^2 L}{\lambda \mu^2 C_L G^2} \left( \frac{c_2 \sigma_\phi \lambda}{LG} + \frac{c_1 C_2 \lambda \Delta_1}{\sigma_\phi^2} \right)
$$

$$
= \frac{1}{8} + \frac{384L}{\lambda C_L G^2} \left( \frac{2 c_2 \lambda \sigma_\phi^3}{LG} + c_1 C_2 \lambda \Delta_1 \right)
$$

$$
+ \left( \frac{c_2 \sigma_\phi \lambda}{LG} + \frac{c_1 C_2 \lambda \Delta_1}{\sigma_\phi^2} \right) \left( \frac{5L^3}{\lambda C_L G^2} \left( \frac{17}{\mu^2} + \frac{36 c_2^2 \sigma_\phi^2 l_{g,1}^2}{\mu^4 G^2} \right) \sigma_{g,1}^2 + \frac{9 l_{g,1}^2 l_{f,0}^2 L}{\lambda \mu^2 C_L G^2} \right)
$$

$$
\leq 1,
$$

where the first inequality is due to (76); the second inequality uses large enough $C_1$ and (Li et al., 2023a, Lemma C.5), the fact that $\ln(4eT/\delta) \leq \ln(C_\beta)$ and $\ln(4eT/\delta) \leq \ln(C_\beta)$, and the choice of $\gamma, Q, T$ that

$$
\eta T \leq \frac{\eta}{\beta^2} + \frac{C_2 \Delta_1 G}{\epsilon^2}, \quad \gamma = \frac{2\beta}{\mu}, \quad Q \geq \frac{1}{2} \ln \beta \Big/ \ln \left( 1 - \frac{\mu}{l_{g,1}} \right);
$$

the third inequality uses (Li et al., 2023a, Lemma C.5), the choice of $\eta, \beta$ that

$$
\frac{\eta}{\beta} \leq \frac{c_2 \sigma_\phi \lambda}{LG \max\{1, \sqrt{\iota}, \ln(C_\beta)\}}, \quad \frac{\beta}{\epsilon^2} \leq \frac{c_1 \lambda}{\sigma_\phi^2 G \max\{1, \sqrt{\iota}, \ln(C_\beta)\}};
$$

and in the last inequality we choose small enough $c_1$ and $c_2$.

**Proof of $I_1/T \leq \epsilon^2$.** Last, we show $I_1/T \leq \epsilon^2$. We have

$$
\begin{aligned}
\frac{I_1}{T} =\ & \frac{16G\Delta_1}{\eta T} + \frac{1024\sigma_\phi^2 G}{\lambda\beta T} + \frac{1024\sigma_\phi^2 G}{\lambda} + \frac{2560\sigma_\phi^2 G\sqrt{\iota}}{\lambda}\sqrt{\frac{1}{\beta^2 T^2} + \frac{1}{T}} \\
& + \frac{33L^2 G}{\lambda}\left(\left(1 + \frac{8\eta^2 l_{g,1}^2 L^2}{\lambda^2\mu^4\gamma^2}\right)\|y_1 - y_1^*\|^2 + \left(\frac{8\gamma}{\mu}\ln\frac{4eT}{\delta} + \frac{32\eta^2 l_{g,1}^2 L^2}{\lambda^2\mu^5\gamma}\right)\sigma_{g,1}^2\right) \\
& + \frac{67G l_{g,1}^2 l_{f,0}^2}{\lambda\mu^2}\left(1 - \frac{\mu}{l_{g,1}}\right)^{2Q} \\
\leq\ & \frac{16\epsilon^2}{C_2} + \frac{3584\sigma_\phi^2 G\max\{1,\sqrt{\iota}\}}{\lambda\beta T} + \frac{1024\sigma_\phi^2 G}{\lambda} + \frac{2560\sigma_\phi^2 G\sqrt{\iota}}{\lambda\sqrt{T}} + 67c_3\epsilon^2 \\
& + \frac{33L^2 G}{\lambda}\left(\left(1 + \frac{2\eta^2 l_{g,1}^2 L^2}{\lambda^2\mu^2\beta^2}\right)\frac{17\beta\sigma_{g,1}^2}{\mu^2}\ln(C_\beta) + \left(\frac{16\beta}{\mu^2}\ln(C_\beta) + \frac{16\eta^2 l_{g,1}^2 L^2}{\lambda^2\mu^4\beta}\right)\sigma_{g,1}^2\right) \\
\leq\ & \frac{16\epsilon^2}{C_2} + \frac{7168\sigma_\phi^2 G\max\{1,\sqrt{\iota}\}\beta}{\lambda} + 67c_3\epsilon^2 \\
& + \frac{33L^2 G}{\lambda}\left(\left(1 + \frac{2c_2^2\sigma_\phi^2 l_{g,1}^2}{\mu^2 G^2}\right)\frac{17c_1\lambda\sigma_{g,1}^2\epsilon^2}{\mu^2\sigma_\phi^2 G} + \left(\frac{16c_1\lambda\epsilon^2}{\mu^2\sigma_\phi^2 G} + \frac{16c_1 c_2^2\lambda l_{g,1}^2\epsilon^2}{\mu^4 G^3}\right)\sigma_{g,1}^2\right) \\
=\ & \left(\frac{16}{C_2} + 7168c_1 + 67c_3 + \frac{33c_1 L^2 G}{\lambda}\left(\left(1 + \frac{2c_2^2\sigma_\phi^2 l_{g,1}^2}{\mu^2 G^2}\right)\frac{17\lambda\sigma_{g,1}^2}{\mu^2\sigma_\phi^2 G} + \left(\frac{16\lambda}{\mu^2\sigma_\phi^2 G} + \frac{16c_2^2\lambda l_{g,1}^2}{\mu^4 G^3}\right)\sigma_{g,1}^2\right)\right)\epsilon^2 \\
\leq\ & \epsilon^2,
\end{aligned}
$$

where the first inequality uses $\sqrt{a+b} \leq \sqrt{a} + \sqrt{b}$ for $a,b \geq 0$, the fact that $\ln(4eT/\delta) \leq \ln(C_\beta)$ and $\|y_1 - y_1^*\|^2 \leq 17\beta\sigma_{g,1}^2\ln(C_\beta)/\mu^2$, and the choice of $T, Q, \gamma$ that

$$
T \geq \frac{C_2\Delta_1 G}{\eta\epsilon^2}, \quad Q \geq \frac{1}{2}\ln\left(\frac{c_3\lambda\mu^2\epsilon^2}{G l_{g,1}^2 l_{f,0}^2}\right) \bigg/ \ln\left(1 - \frac{\mu}{l_{g,1}}\right), \quad \gamma = \frac{2\beta}{\mu};
$$

the second inequality uses the choice of $T, \eta, \beta$ that

$$
T \geq \frac{1}{\beta^2}, \quad \eta \leq \frac{c_2\sigma_\phi\lambda\beta}{LG}, \quad \beta \leq \frac{c_1\lambda\epsilon^2}{\sigma_\phi^2 G\max\{1,\sqrt{\iota},\ln(C_\beta)\}};
$$

and in the last inequality we choose small enough $c_1, c_2, c_3$ and large enough $C_2$. $\qquad\square$

## E   Comparison Tables

**Assumption E.1.** Consider the following smoothness assumptions:

(A) The objective function is $L$-smooth.

(B) The objective function is $(L_0, L_1)$-smooth (Zhang et al., 2020a, Definition 1.1, Remark 2.3).

(C) The objective function is $(\rho, L_0, L_\rho)$-smooth with $0 \leq \rho < 2$ (Li et al., 2023a, Definition 3.2).

The above assumptions satisfy: Assumption E.1(A) $\implies$ Assumption E.1(B) $\implies$ Assumption E.1(C). In other words, Assumption E.1(A) is the strongest, and Assumption E.1(C) is the weakest.

**Assumption E.2.** The (stochastic) gradient norm of the objective function is (almost surely) bounded.

**Assumption E.3.** Suppose the following stochastic estimators are unbiased and satisfy:

$$
\mathbb{E}_{\xi\sim\mathcal{D}_f}[\|\nabla_x F(x,y;\xi) - \nabla_x f(x,y)\|^2] \leq \sigma_{f,1}^2, \quad \mathbb{E}_{\xi\sim\mathcal{D}_f}[\|\nabla_y F(x,y;\xi) - \nabla_y f(x,y)\|^2] \leq \sigma_{f,1}^2,
$$

$$
\Pr\{\|\nabla_y G(x,y;\xi) - \nabla_y g(x,y)\| \geq \lambda\} \leq 2\exp(-2\lambda^2/\sigma_{g,1}^2) \quad \forall\lambda > 0,
$$

$$
\mathbb{E}_{\zeta\sim\mathcal{D}_g}[\|\nabla_{xy}^2 G(x,y;\zeta) - \nabla_{xy}^2 g(x,y)\|^2] \leq \sigma_{g,2}^2, \quad \mathbb{E}_{\zeta\sim\mathcal{D}_g}[\|\nabla_{yy}^2 G(x,y;\zeta) - \nabla_{yy}^2 g(x,y)\|^2] \leq \sigma_{g,2}^2.
$$

# F  More Experimental Details

## F.1  Hyperparameter Settings for Hyper-representation

**Hyper-representation on BERT.** The upper-level learning rate $\eta$ and the lower-level learning rate $\gamma$ are tuned in a range of $[1.0 \times 10^{-4}, 0.1]$ for all the baselines. The optimal learning rate pairs $(\eta, \gamma)$ are, $(0.01, 0.001)$ for MAML, $(0.01, 0.02)$ for ANIL, $(0.01, 0.002)$ for StocBio, $(0.01, 0.001)$ for TTSA, $(0.01, 0.01)$ for SABA, $(0.01, 0.01)$ for MA-SOBA, and $(0.1, 0.05)$ for both BO-REP and SLIP, $(1.0 \times 10^{-4}, 5.0 \times 10^{-3})$ for AdamBO.

**Hyper-representation on RNN.** The upper-level learning rate $\eta$ and the lower-level learning rate $\gamma$ are tuned in a range of $[1.0 \times 10^{-4}, 0.1]$ for all the baselines. The optimal learning rate pairs are listed as follows, $(0.01, 0.01)$ for MAML, $(0.01, 0.05)$ for ANIL, $(0.01, 0.01)$ for StocBio, $(0.02, 0.05)$ for TTSA, $(0.01, 0.05)$ for SABA, $(0.05, 0.05)$ for MA-SOBA, and $(0.1, 0.05)$ for both BO-REP and SLIP, $(1.0 \times 10^{-4}, 1.0 \times 10^{-3})$ for AdamBO.

Other hyper-parameter settings are summarized as follows. The steps for neumann series estimation in StocBiO, AdamBO is set to 3, while it is uniformly sampled from $\{1, 2, 3\}$ in TTSA. The momentum parameter $\beta = 0.1$ is fixed in SLIP, MA-SOBA, BO-REP, AdamBO, and $\beta_{sq} = 0.001$ in AdamBO. The warm start steps for the lower level variable in BO-REP, SLIP, AdamBO are set to 3. The number of inner loops for StocBio is set to 3. BO-REP uses the periodic update for the low-level variable, and sets the iterations $N = 3$ and the update interval $I = 2$. The hyperparameter $\lambda$ in the Adam update is fixed as $1.0 \times 10^{-8}$ in AdamBO.

## F.2  Hyperparameter Settings for Deep AUC Maximization

We tune the best hyperparameters for each algorithm, including upper-/lower-level step size, the number of inner loops, momentum parameters, etc. The upper-level learning rate $\eta$ and the lower-level learning rate $\gamma$ are tuned in a wide range of $[1.0 \times 10^{-6}, 0.1]$ for all the baselines on experiments of AUC maximization.

**AUC maximization on Transformer**. The best learning rates $(\eta, \gamma)$ are summarized as follows: Stocbio: $(0.005, 0.0001)$, TTSA: $(0.0005, 0.001)$, SABA: $(0.001, 0.005)$, MA-SOBA: $(0.0005, 0.005)$, SUSTAIN: $(0.005, 0.001)$, VRBO: $(0.005, 0.0005)$, BO-REP: $(0.0001, 0.0001)$, SLIP: $(0.0001, 0.001)$, AccBO: $(0.0005, 0.0001)$, AdamBO: $(5.0 \times 10^{-6}, 0.005)$. Note that SUSTAIN decays its upper-/lower-level step size with epoch $(t)$ by $\eta = \eta/(t+2)^{1/3}, \eta_{low} = \gamma/(t+2)^{1/3}$. Other algorithms use a constant learning rate.

**AUC maximization on RNN**. The best learning rates $(\eta, \gamma)$ are summarized as follows: StocBio: $(0.01, 0.001)$, TTSA: $(0.005, 0.01)$, SABA: $(0.01, 0.005)$, MA-SOBA: $(0.01, 0.005)$, SUSTAIN: $(0.03, 0.01)$, VRBO: $(0.05, 0.01)$, BO-REP: $(0.001, 0.001)$, SLIP: $(0.001, 0.001)$, AccBO: $(0.005, 0.005)$, AdamBO: $(1.0 \times 10^{-5}, 0.001)$.

Other hyper-parameter settings are summarized as follows. The steps for neumann series estimation in StocBiO, VRBO, and AdamBO is set to 3, while it is uniformly sampled from $\{1, 2, 3\}$ in TTSA, SUSTAIN, and AccBO. AccBO uses the Nesterov accelerated gradient descent for the lower-level update, the momentum parameter $\alpha = 0.5$ for AccBO, the averaging parameter $\nu = 0.5$ for AccBO. The batch size is set to 32 for all algorithms except VRBO, which uses a larger batch size of 64 (tuned in the range of $\{32, 64, 128, 256, 512\}$) at the checkpoint (snapshot) step and 32 otherwise. The momentum parameter $\beta = 0.1$ is fixed in SLIP, AccBO, MA-SOBA, BO-REP, and AdamBO, and $\beta_{sq} = 0.001$ in AdamBO. The warm start steps for the lower level variable in BO-REP, SLIP, AccBO, and AdamBO are set to 3. The number of inner loops for StocBio is set to 3. BO-REP uses the periodic updates for low-level variable, and sets the iterations $N = 3$ and the update interval $I = 2$. The hyperparameter $\lambda$ in the Adam update is fixed as $1.0 \times 10^{-8}$ for AdamBO.

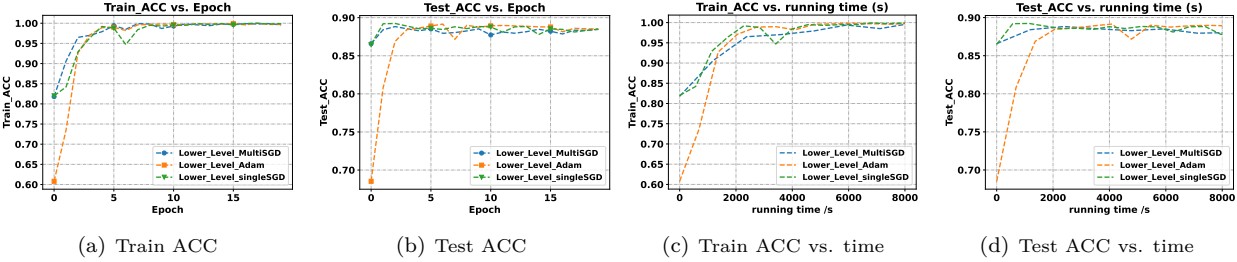

(a) Train ACC      (b) Test ACC      (c) Train ACC vs. time      (d) Test ACC vs. time

Figure 7: Ablation for the lower level update, where Lower_Level_singleSGD is the current update strategy used by AdamBO, Lower_Level_MultiSGD uses multiple-step SGD to update the lower level function, and Lower_Level_Adam uses Adam with a single step to update the lower level function.

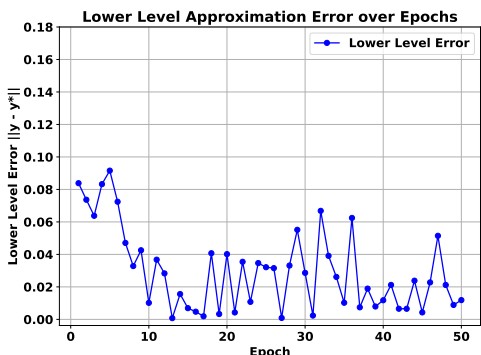

Figure 8: Lower level approximation error over epochs in deep AUC maximization.

# G  Ablation for Lower Level Update Strategies

We empirically compare three lower-level update strategies on BERT for hyper-representation: one-step SGD (our default choice), multiple-step SGD, and one-step Adam. Figure 7 (a, b) show that multiple-step SGD performs almost the same as one-step SGD, with no significant improvement in either train or test accuracy. One-step Adam gives the best final performance. Here, one epoch means that the upper-level variable passes through all training samples once. The subfigures (c, d) show running time results where Lower_level_MultiSGD runs 3-step SGD in every upper level update, so its running time results are worse than other baselines. Therefore, from an empirical perspective, our results support the claim that the lower-level problem is relatively easy to optimize, and one-step SGD is already a strong and efficient choice. We keep one-step SGD in the main method because it maintains the simplicity of the single-loop framework and matches our convergence analysis, which currently does not extend to Adam-type lower-level updates.

# H  Approximation Error of Lower Level Variable

We further empirically verify the lower-level approximation error $\|y_t - y_t^*\|$ in the deep AUC experiment. Specifically, at the end of each epoch, we compute two lower-level solutions: $y_t^*$, obtained by running the lower-level optimization over all training samples, and $y_t$, obtained by following our algorithm with only a single SGD step for the lower-level update. We then measure their distance $\|y_t - y_t^*\|$. As shown in Figure 8, this distance remains consistently small throughout training, and quickly decreases to a low level after the initial epochs. Although there are some fluctuations, the approximation error is generally close to zero and stays well controlled. This provides empirical evidence that the one-step lower-level update already approximates the exact lower-level solution closely in practice.

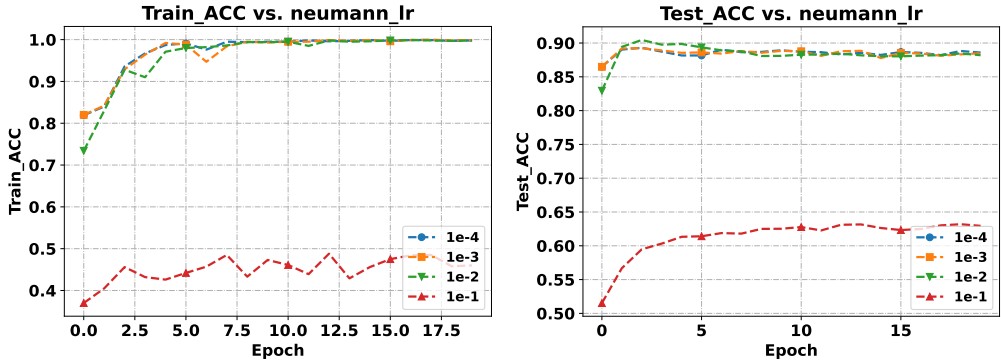

Figure 9: Training and Test results with varying Neumann series learning rates.

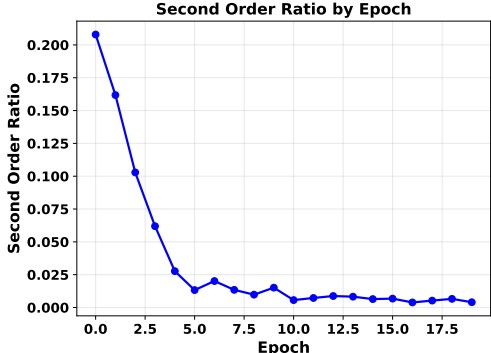

Figure 10: Second order ratio in hypergradient evolves with training epochs.

## I  Sensitivity Analysis of `neumann_lr`

We conducted a sensitivity study over `neumann_lr` $\in \{1.0 \times 10^{-4}, 1.0 \times 10^{-3}, 1.0 \times 10^{-2}, 1.0 \times 10^{-1}\}$ in hyper-representation. As shown in Figure 9, our method is fairly stable over a moderate range of values: $1.0 \times 10^{-4}, 1.0 \times 10^{-3}, 1.0 \times 10^{-2}$ , both training and test accuracy are very similar, with test accuracy staying around 0.88-0.90. In contrast, performance drops substantially when `neumann_lr` is set too large $(1.0 \times 10^{-1})$, indicating that overly aggressive `neumann_lr` can destabilize the HVP approximation. These results suggest that while `neumann_lr` is indeed an implementation hyper-parameter, the method is not overly sensitive to it within a reasonable range.

## J  Second-Order Ratio in the Hypergradient Evolves with Training Epochs

For the task of hyper-representation, we track the ratio of the second-order part to the total hypergradient: $r_t = \|g_t^{(2)}\|/\|\hat{\phi}_t\|$, where $g_t^{(2)}$ denotes the second-order component and $\hat{\phi}_t$ is the total hypergradient. In practice, at the end of each epoch, we sample one mini-batch and compute this ratio to monitor how the relative contribution of the second-order term evolves during training.

Our result in Figure 10 show a clear stage-dependent pattern: the ratio is relatively large in the early phase of training, indicating that the second-order term makes a substantial contribution when the model is still far from convergence. As training proceeds, the ratio decreases steadily and becomes much smaller in later epochs, which is consistent with the intuition that curvature-related corrections become less pronounced once optimization enters a more stable regime.

Therefore, this result suggests that the second-order term is not merely of theoretical interest. It has a practically meaningful impact, especially in the early stage of training, while its relative importance naturally diminishes as the model converges.

