# OpenReview forum: "On the Convergence of Adam-Type Algorithm for Bilevel Optimization under Unbounded Smoothness"
_TMLR — Accepted by TMLR_

### Review · Reviewer_K1r2 · 2026-02-18

**Summary Of Contributions:**

This paper introduces AdamBO, an Adam-type algorithm for stochastic bilevel optimization under generalized (unbounded) smoothness conditions. The proposed method applies SGD to the strongly convex lower-level problem while updating the upper-level variable using an adaptive Adam-style rule. The central technical contribution is a novel randomness decoupling lemma, which enables high-probability control of the lower-level tracking error despite stochastic gradients and adaptive upper-level updates. The authors evaluate their proposed method on meta-learning with transformer-based models and deep AUC maximization with RNNs, where AdamBO demonstrates faster convergence and competitive or improved performance relative to existing bilevel baselines.
The primary strength of the paper lies in its theoretical contribution. While Adam has strong empirical success, its convergence theory in stochastic bilevel settings is limited. This work meaningfully extends adaptive optimization theory to a more challenging regime, including unbounded smoothness and stochastic updates, and introduces analytical tools that may be useful beyond this specific algorithm.
That said, the algorithmic novelty itself is somewhat limited, as AdamBO largely combines existing components (SGD for the inner problem and Adam for the outer problem) rather than introducing a fundamentally new optimization scheme. Additionally, although the experiments cover multiple applications, they are conducted on moderately sized transformer and RNN models in classification settings. It would strengthen the empirical validation to evaluate AdamBO on larger-scale models or more diverse bilevel tasks to better assess its robustness and scalability.

**Audience:**

Yes

**Audience Explanation:**

Yes, I believe this work would be of interest to some individuals in TMLR’s audience. This paper provides a theoretical extension of Adam to the stochastic bilevel optimization setting under generalized smoothness, where convergence guarantees remain limited. The introduction of the randomness decoupling technique may also be useful for future theoretical research in this area.

**Claims And Evidence:**

Yes

**Claims Explanation:**

The theoretical claims appear internally consistent and are derived under clearly stated assumptions, including strong convexity of the lower-level problem and generalized smoothness of the upper-level objective. While the analysis is technically involved and I did not independently verify every step, the arguments seem to follow established proof techniques in stochastic bilevel optimization. The empirical results provide additional support for the practical effectiveness of the method, though further validation on larger-scale models would strengthen the claims.

**Requested Changes:**

Please refer to my comments above to address my concerns. In addition, there is only a couple of adjustments I would like to see made to the submission which are not critical to my overall recommendation but would strengthen the submission. First is the evaluation of the proposed approach on larger models which would further demonstrate the robustness and scalability of the approach and strengthen the practical relevance of the work. Second, in several plots, portions of the curves are partially obscured by the legend placement, which makes detailed comparison slightly difficult. Adjusting the legend positioning by placing it outside the plotting area could improve readability.

---

> ### Author Response · Authors · 2026-03-16
>
> **A1.**
> Regarding larger models, our current experiments already include evaluations beyond small architectures: hyper-representation on an 8-layer BERT model (Section 5.1) and deep AUC maximization on Transformer model (Section 5.2). The hyper-representation model contains 68.5M trainable parameters and already achieves over 99\% accuracy in the experiment of Section 5.1. These results suggest that our method is not limited to small models and can scale to moderately large neural architectures.
>
> Regarding readability, we agree that some legends can obscure parts of curves. In the revised version, we have adjusted figure layouts and placed legends outside the main plotting area to improve visual clarity and detailed comparison. Please refer to the revised paper for details.

---

### Review · Reviewer_EwGD · 2026-02-28

**Summary Of Contributions:**

The paper studies an Adam-type algorithm for bilevel optimization where the lower-level problem is strongly convex and the upper-level objective satisfies $(L_0, L_1)$-smoothness. Specifically, the authors propose a single-loop method, termed \textbf{AdamBO}, which performs stochastic gradient descent updates on the lower-level variable $y$ while applying Adam, based on stochastic hypergradient estimates, to the upper-level variable. Under the stated assumptions, the algorithm is shown to achieve a sample complexity of $\mathcal{O}(\epsilon^{-4})$ for obtaining an $\epsilon$-stationary point.

**Audience:**

Yes

**Audience Explanation:**

While Adam has achieved significant success in single-level optimization, it is natural to consider its extension to the bilevel setting. Providing a rigorous theoretical analysis for such an approach is therefore of clear interest to the TMLR community.

**Claims And Evidence:**

Yes

**Claims Explanation:**

The claims are supported by both theoretical and empirical evidence. The paper provides formal complexity proofs (with a proof sketch in the main text), and the experimental results further support the theoretical findings.

**Requested Changes:**

I have the following questions and concerns:

1. Does Assumption 3.4 imply parts of Assumption 3.2? For instance, if the $(L_0, L_1)$-smoothness condition holds for each realization $\xi$, does this imply the corresponding smoothness property for the expected objective in Assumption 3.2(i)?

2. In the experiments, how large is $Q$ when estimating the hypergradient?

3. I suggest explicitly stating the dependence on $\delta$ in Theorem 4.1. What are the orders of the constants $C_1$ and $C_2$?

4. In the proof sketch, it would be helpful to briefly explain how the stopping-time technique is applied, even if it closely follows (Li et al, 2023a).

5. The experimental setup in Section 5.1, where only the adapter $y$ is trained in the lower level, appears different from Finn et al.~(2017).

6.  Is using a single SGD step for the lower-level variable empirically better than performing multiple steps, or applying Adam to the lower level as well?

7.  The experiments report performance versus epochs; however, different methods may incur different per-epoch computational costs.

8.  In Section 5.1, how is strong convexity of the lower-level problem ensured? Is it induced by setting $\mu$ sufficiently large?

9.  There appears to be a typo in the paragraph below Algorithm 1: "lower level" should likely be  "upper level".

---

> ### Author Response · Authors · 2026-03-16
>
> **A1.**
> Yes, Assumption 3.4 imply parts of Assumption 3.2. For example, $\\|\nabla_{y} f(z)-\nabla_{y} f(z')\\| = \\|\mathbb{E}\_{\xi}[\nabla_{y} F(z;\xi)-\nabla_{y} F(z';\xi)]\\| \leq \mathbb{E}\_{\xi}\\|\nabla_{y} F(z;\xi)-\nabla_{y} F(z';\xi)\\| \leq (L_{y,0}+L_{y,1}\\|\nabla_{y} f(z)\|)\\|z-z'\\|$.
>
> **A2.**
> We fix $Q=3$ in the neumann series when estimating the hypergradient.
>
> **A3.**
> We explicitly stated in Section 4.2 that the small enough constants $c_1, c_2, c_3$ and the large enough constants $C_1, C_2$ are independent of $\epsilon$ and the failure probability $\delta$. They depend only on problem-specific constants such as $\mu, l_{g,1}$; see the proofs of Lemma D.11 (p. 42), Lemma D.13 (p. 48), and Lemma D.14 (pp. 49-50) for details. Because these dependencies are quite involved, we follow [1, Section 4] and refer to them simply as small enough or large enough.
>
> **A4.**
> We have added the following paragraph to explain how the stopping-time technique is applied in Section 4.3.3:
>
> Let $\tau$ be the first time the sub-optimality gap becomes strictly larger than $\psi$, truncated at $T+1$; formally,
> $
>     \tau \coloneqq \min\\{t \mid \Phi(x_t)-\Phi^* > \psi\\} \wedge (T+1).
> $
> Then, for any $t<\tau$, we have $\Phi(x_t)-\Phi^*\leq \psi$ and therefore $\\|\nabla\Phi(x_t)\\|\leq G$. Based on our discussion above, we can analyze the updates before time $\tau$ and construct a Lyapunov function to derive an upper bound on $\Phi(x_\tau)-\Phi^*$. On the other hand, if $\tau\leq T$, the definition of $\tau$ immediately gives a lower bound, namely $\Phi(x_\tau)-\Phi^*>\psi$. If this lower bound exceeds the upper bound, we obtain a contradiction, which implies $\tau=T+1$. That is, before the algorithm terminates, the sub-optimality gap and gradient norm are always bounded by $F$ and $G$, respectively.
>
> **A5.**
> Our setup in Section 5.1 follows the same meta-learning protocol as [2]: each task is split into task-train and task-validation sets ($\mathcal{D}_i^{tr},\mathcal{D}_i^{val}$), the inner level performs task-specific adaptation on $\mathcal{D}_i^{tr}$, and the outer level updates shared meta-parameters using $\mathcal{D}_i^{val}$. In our notation, this corresponds to the decomposition into shared parameters $x$ and task-specific parameters $y_i$. In experiments, we instantiate $y_i$ as the adapter (last linear layer) and $x$ as representation layers; this is a modeling choice for our bilevel formulation (and to satisfy the strongly-convex lower-level condition).
>
> **A6.**
> We empirically compare three lower-level update strategies on BERT for hyper-representation: one-step SGD (our default choice), multiple-step SGD, and one-step Adam. Figure 7 (a, b) in revised Appendix G show that multiple-step SGD performs almost the same as one-step SGD, with no significant improvement in either train or test accuracy. One-step Adam gives the best final performance. Here, one epoch means that the upper-level variable passes through all training samples once. The  Figure 7 (c, d) show running time results where Lower\_level\_MultiSGD runs 3-step SGD in every upper level update, so its running time results are worse than other baselines. Therefore, from an empirical perspective, our results support the claim that the lower-level problem is relatively easy to optimize, and one-step SGD is already a strong and efficient choice. We keep one-step SGD in the main method because it maintains the simplicity of the single-loop framework and matches our convergence analysis, which currently does not extend to Adam-type lower-level updates.
>
> **A7.**
> We agree that epoch-based curves alone may not fully reflect computational cost differences across methods. For this reason, in the experiments we report both performance-vs-epoch and performance-vs-running-time curves (for both hyper-representation and AUC tasks) in Figures 1-4. These time-based plots are intended to provide a compute-aware comparison when per-epoch cost differs across algorithms. We also run all methods on the same hardware platform (see Section 5).
>
>
> [1] Haochuan Li, Alexander Rakhlin, and Ali Jadbabaie. "Convergence of adam under relaxed assumptions." Advances in Neural Information Processing Systems 36 (2023): 52166-52196.
>
> [2] Raghu, Aniruddh, Maithra Raghu, Samy Bengio, and Oriol Vinyals. "Rapid Learning or Feature Reuse? Towards Understanding the Effectiveness of MAML." In Eighth International Conference on Learning Representations.

---

> > ### Author Response · Authors · 2026-03-16
> >
> > **A8.**
> > We first explain why the cross-entropy loss is convex with respect to a linear adapter $y_i$, then show how the $\ell_2$ regularization induces $\mu$-strong convexity per task, and finally why summing across tasks preserves $\mu$-strong convexity.
> >
> > - **Convexity of Cross-Entropy w.r.t. a Linear Layer.** Once the meta-learner's parameters $x$ are fixed, the feature representation for any given sample is a constant vector $h$. The adapter $y_i$ is a linear map (matrix or vector) that transforms $h$ into logits $z = y_i h$. In multi-class classification, the cross-entropy for a sample labeled $c$ is $-\log\left(\frac{\exp(z_c)}{\sum_{k=1}^{C}\exp(z_k)}\right)$, where $z = y_i h$. It is a standard result from logistic or multinomial-logistic regression that this cross-entropy is convex in the parameters of the linear layer $y_i$. Concretely, $z$ depends affinely (or linearly) on $y_i$, and since the function $\log(\sum_k \exp(\cdot))$ is well-known to be convex, it follows that $-z_c + \log(\sum_k \exp(z_k))$ is also convex.
> >
> > - **Per-Task Objective is $\mu$-Strongly Convex.** For a single task $i$, since $\mathcal{L}\_{D_i^{\mathrm{tr}}}(x,y_i;\zeta)$ is convex in $y_i$, adding $\frac{\mu}{2}\\|y_i\\|^2$ makes the per-task objective $\mu$-strongly convex. More precisely, the Hessian of $h_i(x,y_i) := \mathcal{L}\_{\mathcal{D}\_i^{\mathrm{tr}}}(x,y_i;\zeta) + \frac{\mu}{2}\\|y_i\\|^2$ with respect to $y_i$ is $\nabla^2_{yy} h_i(x,y_i) \succeq \mu I_d$. Thus each per-task objective is $\mu$-strongly convex.
> >
> > - **Summation over $K$ Tasks Preserves $\mu$-Strong Convexity.** The lower-level function sums these per-task functions across $K$ tasks: $g(x,y)=\sum_{i=1}^{K} h_i(x,y_i)$. Because each $h_i$ is $\mu$-strongly convex in its own block $y_i$ and does not depend on $y_j$ ($j\neq i$), the resulting function $g(x,\cdot)$ is block-separable and inherits $\mu$-strong convexity in the entire vector $y$. Concretely, $\nabla^2_{yy} g(x,y) = \text{diag}\left(\nabla^2_{yy} h_1(x,y_1),\dots,\nabla^2_{yy} h_K(x,y_K)\right) \succeq \mu I_{Kd}$. Hence, $g(x,y)$ is $\mu$-strongly convex in $y$.
> >
> > **A9.**
> > This is not actually a typo. That sentence is meant to emphasize that, in the analyses of [3, 4, 5], it is crucial to ensure that the lower-level error $\\|y_t-y_t^*\\|$ is small and the hypergradient bias is negligible.
> >
> > [3] Jie Hao, Xiaochuan Gong, and Mingrui Liu. "Bilevel Optimization under Unbounded Smoothness: A New Algorithm and Convergence Analysis." In The Twelfth International Conference on Learning Representations.
> >
> > [4] Xiaochuan Gong, Jie Hao, and Mingrui Liu. "A Nearly Optimal Single Loop Algorithm for Stochastic Bilevel Optimization under Unbounded Smoothness." In International Conference on Machine Learning, pp. 15854-15892. PMLR, 2024.
> >
> > [5] Xiaochuan Gong, Jie Hao, and Mingrui Liu. "An accelerated algorithm for stochastic bilevel optimization under unbounded smoothness." Advances in Neural Information Processing Systems 37 (2024): 78201-78243.

---

### Review · Reviewer_x4sK · 2026-03-06

**Summary Of Contributions:**

This paper proposes a bilevel optimization method, AdamBO, in which the upper-level is switched to an Adam-style update rule using the Neumann-series approximation of the hypergradient (while the lower-level is SGD). The main part of the paper is to provide a proof of convergence (in a style similar to the convergence proof of Adam under relaxed smoothness) of AdamBO, and empirically evaluate AdamBO on two tasks (namely, a meta-learning task that jointly fine-tunes a deep neural network on multiple classification tasks with separate classification head layers adapted to each task, and a binary classification task that maximizes AUC) that can be formalized as bilevel optimization. The authors have compared with several bilevel optimization methods and demonstrated fast convergence of AdamBO.

To serve as a reference for later discussion, I would like to briefly summarize the background problem settings here:

Using notations in the paper, the objective function $\Phi$ in bilevel optimization is given by:

$$
\min_x \Phi(x):=f(x,y^\ast(x)), \quad\mbox{where}\quad y^\ast(x)=\arg\min_y g(x,y).
$$

In which, $x$ is the "upper-level" trainable tensor, $f$ is usually a loss function on a deep neural network (i.e., non-convex), and $g$ is convex on $y$.

The approach adopted in this paper replaces $y^\ast(x)$ by a "lower-level" trainable tensor $y$, and applies stochastic optimization to minimizing $f(x,y)$, while keeping $y$ close to the minimum of $g(x,y)$ (with the current $x$ fixed).

One difference from the ordinary minimization of $f(x,y)$, is the use of "hypergradient" for $x$; this is derived by taking the Taylor expansion of $g(x,y)$ to the 2nd order on $y$:

$$
g(x,y)\approx\frac{1}{2}\nabla^2_{yy}g(x,y_t)(y-y^\ast_t)^2+C
$$

So, differentiating $y$ on both sides, we get

$$
\nabla_y g(x,y_t)\approx\nabla^2_{yy}g(x,y_t)(y_t-y^\ast_t),
$$

and solving $y^\ast_t$ we get

$$
y^\ast_t\approx y_t-\nabla_y g(x,y_t)[\nabla^2_{yy}g(x,y_t)]^{-1},
$$

 which is viewed as a function of $x$ and can be differentiated on $x$:

$$
\nabla_x y^\ast_t(x_t)\approx -\nabla^2_{yx} g(x_t,y_t)[\nabla^2_{yy}g(x_t,y_t)]^{-1},
$$

Note that, at this step, we have assumed that $[\nabla^2_{yy}g(x,y_t)]^{-1}\equiv[\nabla^2_{yy}g(x_t,y_t)]^{-1}$ is unchanged around $x_t$.

Then, substituting the above into the calculation of gradient $\nabla f(x,y^\ast(x))$, we obtain the "hypergradient":

$$
\nabla\Phi(x)=\nabla f(x,y^\ast(x))\approx \nabla_x f(x,y^\ast(x)) - \nabla^2_{yx} g(x_t,y_t)[\nabla^2_{yy}g(x_t,y_t)]^{-1}\nabla_y f(x,y^\ast(x)).
$$

It is characteristic in using the 2nd order derivatives of $g(x,y)$.

Then, the "Neumann series" is to approximate $[\nabla^2_{yy}g(x_t,y_t)]^{-1}$ by applying the formula

$$
(I-X)^{-1}=\sum_{q=0}^\infty X^q
$$

to $X=I-\nabla^2_{yy}g(x_t,y_t)$, assuming that $\nabla^2_{yy}g(x_t,y_t)$ is close enough to $I$.

**Additional Comments:**

Some minor issues:

a. The end of Page 6, and the beginning of Page 28: "line 5 of Algorithm 1" should be line 4?

b. In Equation (14), $G(x,y)$ should be $g(x,y)$?

c. Table 1 and Table 2 are put at the very end of the Appendix, but they are discussed in the main paper. Please put the tables in the main paper.

d. On Page 8, "To evaluate our approach on meta-learning, we construct K = 500 meta tasks": How would you construct 500 tasks from 6 coarse-grained categories? Does this mean that most of the 500 meta tasks are essentially the same task?

e. Please add more explanations to Section 4.3.3. Currently it's just a list of Lemmas without any description of how the lemmas are wired together to complete the remaining proof of the main theorem.

**Audience:**

Yes

**Audience Explanation:**

Bilevel optimization is an interesting research direction, and I find the application of bilevel optimization to the two tasks in the paper (meta-learning and AUC maximization) quite reasonable and might be empirically useful.

The theoretical convergence proof provided in the paper, although being very technical, follows a common stream of previous works and might inspire other works as well.

**Broader Impact Concerns:**

None.

**Claims And Evidence:**

Yes

**Claims Explanation:**

Yes and no. The technical content of this paper is quite dense, and currently it definitely needs revision to provide better and more detailed explanations regarding several aspects of the work (see Requested Changes below). Despite that, I can roughly follow the theoretical proof and it seems solid (and quite outstanding!) to me. I would like to learn more about the proof during the discussion with the authors.

**Requested Changes:**

Here are my questions:

1 The definition of $y^\ast(x)$, is the unique minimum of the convex function $g(x,y)$. However, as I demonstrated in the beginning, the derivation of the hypergradient seems to rely on the Taylor expansion:

$$
g(x,y)\approx\frac{1}{2}\nabla^2_{yy}g(x,y_t)(y-y^\ast_t)^2+C
$$

One should note that the $y^\ast_t$ in this Taylor expansion is generally not equal to $y^\ast(x)$, even if $g$ is convex -- they only coincide when $g$ is a quadratic function. Throughout the paper, it seems that $y^\ast(x)$ is actually treated as this $y^\ast_t$, which is an approximation of $y^\ast(x)$ by a one-step Newton's method -- So my question: Is this approximation error properly handled in the proof?

2. Related to the above, in the discussion of Main Challenges on Page 4, it says "[the first challenge...] is because the hypergradient estimator in bilevel optimization may have a non-negligible bias due to inaccurate estimation of the lower-level variable". What exactly does "non-negligible bias" mean here?

3. Related to the above, can we empirically measure $\lVert y_t - y^\ast(x_t) \rVert$ during the training process? Is the performance sensitive to controlling this error (e.g., by changing the learning-rate or number of steps in the inner loop)?

4. Related to the above, how does the bilevel optimization method compare to simply minimizing the relaxed problem $\min_{x,y} f(x,y)$ (i.e., without the constraint of $y$ being minimum of $g$) in practice? Because it seems to me that the two evaluation tasks in this paper (meta-learning and AUC minimization) do not actually require the constraint being satisfied throughout the training process?

5. Related to the above, can we theoretically understand the bilevel optimization proposed in this paper as a method of solving the relaxed problem $\min_{x,y} f(x,y)$ by Adam, but with some modifications to the gradient (namely, utilizing 2nd order information of $g$ to augment the gradient for $x$, while applying SGD to $y$)? From this perspective, it seems to me that the Adam convergence proof in previous works can still cover the case mostly? What would be the main technical challenge in doing so?

6. I believe $l_{g,1}$ is not properly defined in the formula of Neumann series approach, on Page 3. This seems to be some Lipschitz constant in the theory, and being treated as a hyperparameter in practice (I checked the code in Supplementary Material, there is a `neumann_lr` which seems to correspond to this constant. Can you explain more on this?

7. Related to the above, how large is the 2nd order part of the hypergradient in practice, compared to the ordinary gradient $\nabla_x f(x,y)$? Does the 2nd order part have any empirical impact on the performance, i.e., how does the performance compare to, say, completely omitting the 2nd order part during optimization?

8. Related to the above, the estimation of the 2nd order part of the hypergradient requires additional data points -- how do you actually do the data feeding in practice (which is not explained in the paper, and I could not fully understand by looking at the code)?

9. Related to the above, in the evaluation experiments (Figure 1, 2, 3, 4, 5, 6), how do you measure epochs? Is this counted by the number of iterations, or the number of data points consumed? It seems tricky to ensure a fair comparison here because different methods may require different numbers of data points per iteration.

10. As I demonstrated in the beginning, the hypergradient also ignores the drift of $[\nabla^2_{yy}g(x,y_t)]^{-1}$ (i.e., the difference $[\nabla^2_{yy}g(x_{t+1},y_t)]^{-1} - [\nabla^2_{yy}g(x_t,y_t)]^{-1}$). Is this drift properly handled in the proof?

11. Maybe related to the above, in the discussion of Main Challenges on Page 4, it says " Second, the existing algorithms... require the
lower-level error to be small, which may not hold for AdamBO". What exactly does not hold for AdamBO here? Is there empirical evidence for this?

12. Related to the above, the paper continues, "In particular, the existing analysis crucially relies on a fixed update length for the upper-level variable at every iteration (due to normalization)..."; I can see that Hao et al. (2024) indeed has an explicit normalization (Line 6 in Algorithm 1 in their paper) which makes the update of upper-level variable a unit vector -- but Adam also has similar normalization by the square-root of running average of gradient squares, which, although technically being more complicated, seems to achieve a similar effect? My question is, do you mean that "the Adam-style normalization is technically more complicated to analyze" here, or do you mean that "there is essentially different behavior brought into the training process by the Adam-style update of the upper-level variable"?

13. Maybe related to the above, regarding Remark 3: It is indeed a concern that the proof relies on the assumption that $\lambda$ is large -- although the authors try to make an excuse that the performance of AdamBO is not sensitive to the choice of $\lambda$ and it can empirically be set to a larger number -- what we actually observe empirically, at least for Adam, is that smaller $\lambda$ always perform better, especially when the model becomes larger (For example, see discussion about the "epsilon hyperparameter" in https://arxiv.org/abs/2407.05872); the hyperparameter $\lambda$ should be considered infinitesimal. Therefore, I would argue that any theory that relies on a finite value of $\lambda$ cannot properly capture the essence of Adam-style optimizers. I would like to see more discussion from the authors, for example, why the explicit normalization in Hao et al. (2024) does not seem to have a similar problem?

14. Related to the above, it seems to me that the issue is partially because all assumptions in the paper are about **worst-case upper bound** of gradient norms -- while for Adam-style optimization we also need some kind of **lower bound** for the **average-case** gradient norms. This is why I don't like the error-bound-based worst-case analysis techniques very much -- I think they are far from providing guidance to the practice. My opinion is that average-case analysis, although currently limited to convex Random Feature models, can provide much more powerful theoretical tools (see, for example, https://arxiv.org/abs/2405.15074). That said, I would not push too hard on this topic since it's not directly related.

---

> ### Author Response · Authors · 2026-03-16
>
> **A1.**
> In this work, the target objective of bilevel optimization is $\Phi(x)=f(x,y^\*(x))$, and the hypergradient is defined as (see [1, Lemma 2.1])
> $
> \nabla\Phi(x)=\nabla_x f(x,y^\*(x)) - \nabla_{xy} g(x,y^\*(x))[\nabla_{yy}^{2} g(x,y^\*(x))]^{-1}\nabla_y f(x,y^\*(x)).
> $
> So the hypergradient is the gradient of the upper-level objective after differentiating through the lower-level solution mapping $y^*(x)$.
>
> In our analysis, $y_t^\*$ is exactly $y^\*(x_t)$, not a one-step Newton approximation or a local Taylor surrogate minimizer. We then estimate the hypergradient by replacing $y_t^\*$ with the iterate $y_t$ (plus Neumann approximation for the inverse Hessian), and explicitly control the resulting error.
>
> The approximation error we do handle explicitly is the tracking error between the lower-level iterate and the exact minimizer, i.e., $\\|y_t-y_t^\*\\|$. This is controlled by the warm-start + randomness-decoupling results (high-probability lower-level error bound), and then propagated to the upper-level analysis through the hypergradient-bias bounds: (i) the deviation caused by using $y_t$ instead of $y_t^*$ is bounded by a term proportional to $\\|y_t-y_t^\*\\|$, and (ii) the Neumann truncation bias at $y_t^\*$ is bounded separately. These ingredients are then combined in the bias-control lemma for $\\|\hat m_t-\hat u_t\\|$ and the final convergence theorem.
>
> **A2.**
> Here, "non-negligible bias" means the (conditional) gap between the expected hypergradient estimator used by the algorithm and the true hypergradient, i.e.,
> $
> \\|\mathbb{E}_t[\hat\phi(x_t,y_t;\bar\xi_t)]-\nabla\Phi(x_t)\\|.
> $
> In our analysis, this quantity has a component caused by lower-level inaccuracy:
> $
> \\|\mathbb{E}_t[\hat\phi(x_t,y_t;\bar\xi_t)]-\mathbb{E}_t[\hat\phi(x_t,y_t^\*;\bar\xi_t)]\\|
> \leq
> (L_0+L_1\\|\nabla\Phi(x_t)\\|)\\|y_t-y_t^\*\\|,
> $
> plus the Neumann-approximation bias at $y_t^\*$ (controlled separately by the $Q$-dependent term). So "non-negligible" means that when $\\|y_t-y_t^\*\\|$ is not sufficiently small, the first term above is not automatically a higher-order/small remainder and must be explicitly controlled in the proof.
>
> **A3.**
> In principle, this quantity is measurable, but the practicality differs by task. For deep AUC maximization (Section 5.2), the lower-level variable is one-dimensional ($\alpha$), and the lower-level objective is strongly-convex quadratic; hence $\alpha^*(\cdot)$ can be computed exactly, so $|\alpha_t-\alpha_t^\*|$ can be tracked during training. For hyper-representation (Section 5.1), $y^\*(x)$ has no closed form; computing $\\|y_t-y_t^\*\\|$ exactly would require solving many task-level lower problems to high precision at each iteration, which is expensive, so we do not report this curve in the current paper.
>
> We further empirically verify the lower-level approximation error $\\|y_t-y_t^\*\\|$ in the deep AUC experiment. Specifically, at the end of each epoch, we compute two lower-level solutions: $y_t^\*$, obtained by running the lower-level optimization over all training samples, and $y_t$, obtained by following our algorithm with only a single SGD step for the lower-level update. We then measure their distance $\\|y_t-y_t^\*\\|$. As shown in Figure 8 in revised Appendix H, this distance remains consistently small throughout training, and quickly decreases to a low level after the initial epochs. Although there are some fluctuations, the approximation error is generally close to zero and stays well controlled. This provides empirical evidence that the one-step lower-level update already approximates the exact lower-level solution closely in practice.
>
> [1] Saeed Ghadimi, and Mengdi Wang. "Approximation methods for bilevel programming." arXiv preprint arXiv:1802.02246 (2018).

---

> > ### Author Response · Authors · 2026-03-16
> >
> > **A4.**
> > In the current paper, we do not include an explicit baseline that directly optimizes $\min_{x,y} f(x,y)$; however, for both tasks in Section 5, that relaxed objective is not equivalent to the target problem.
> >
> > For hyper-representation (Section 5.1), the bilevel formulation explicitly models task adaptation [2]: $y_i^\*(x)$ is obtained from task-train data $\mathcal{D}\_i^{tr}$, while $x$ is optimized using task-validation data $\mathcal{D}\_i^{val}$. Replacing this with $\min_{x,y} f(x,y)$ removes the adaptation mapping $y_i^*(x)$ and changes the objective to a different training criterion.
> >
> > For deep AUC maximization (Section 5.2), the problem is originally a min-max objective [3, 4] and is reformulated as bilevel with $g=-f$; the lower-level variable corresponds to the inner maximization (equivalently, minimizing $-f$). Optimizing $\min_{x,y} f(x,y)$ instead would reverse that inner objective, so it is again a different problem.
> >
> > We agree with your point that exact feasibility $y_t=y^\*(x_t)$ is not required at every iteration in practice. Our method allows inexact lower-level updates and controls this approximation error through $\\|y_t-y_t^*\\|$ in both the algorithmic design and the analysis.
> >
> > **A5.**
> > Algorithmically, there is indeed a similarity. However, the theoretical target in our paper is stationarity of the bilevel objective $\Phi(x)=f(x,y^\*(x))$, not stationarity of the relaxed objective $\min_{x,y} f(x,y)$. The key issue is that the $x$-update uses a hypergradient estimator, and when $y_t\neq y_t^\*$ its conditional expectation is biased relative to $\nabla\Phi(x_t)$; this bias contains a term proportional to $\\|y_t-y_t^\*\\|$ (plus the Neumann truncation term).
> >
> > This is the main reason why existing single-level Adam analyses (e.g., for generalized smooth objectives) do not directly apply: they do not need to control a moving lower-level target $y_t^*=y^\*(x_t)$ and its induced hypergradient bias. The core technical challenge is therefore the coupled control of (i) lower-level tracking error and (ii) upper-level Adam dynamics. In particular, Adam's coordinate-wise, history-dependent step sizes mean the upper-level drift is not fixed-length, so the standard bilevel arguments that rely on small lower-level error under fixed drift cannot be used directly. Our randomness-decoupling lemma and auxiliary-sequence analysis are introduced to handle exactly this coupling.
> >
> > **A6.**
> > In our paper, $l_{g,1}$ is defined in Assumption 3.2(iv): it is the joint smoothness constant of the lower-level function $g(x,y)$. In the Neumann-series hypergradient estimator, this constant is used as the scaling factor in
> > $
> > \frac{1}{l_{g,1}}\sum_{q=0}^{Q-1}\prod_{j=1}^{q}\left(I-\frac{\nabla_{yy}^2 G}{l_{g,1}}\right),
> > $
> > which approximates $[\nabla_{yy}^2 g]^{-1}$. The key role of $l_{g,1}$ is to make the Neumann iteration contractive (yielding the $(1-\mu/l_{g,1})^Q$ approximation term in our bounds).
> >
> > **A7.**
> > In our setting, the second-order part is a core component of the bilevel hypergradient estimator:
> > $
> > \hat\phi(x,y;\bar\xi)=\nabla_x F(x,y;\xi)-\nabla_{xy}G(x,y;\zeta^{(0)})P_Q\nabla_y F(x,y;\xi),
> > $
> > where $P_Q$ is the Neumann-series approximation of $[\nabla_{yy}^{2}g(x,y)]^{-1}$. So this second term or the hypergradient (estimator) is fundamentally different from the ordinary first-order gradient $\nabla_x f(x,y)$.
> >
> > If we omit this second-order term and update $x$ using only $\nabla_x F(x_t,y_t)$, the method becomes a first-order surrogate and no longer estimates the true bilevel gradient $\nabla\Phi(x_t)$. In other words, it generally optimizes a different direction/objective from the target bilevel problem.
> >
> >
> > [2] Raghu, Aniruddh, Maithra Raghu, Samy Bengio, and Oriol Vinyals. "Rapid Learning or Feature Reuse? Towards Understanding the Effectiveness of MAML." In The Eighth International Conference on Learning Representations.
> >
> > [3] Yiming Ying, Longyin Wen, and Siwei Lyu. "Stochastic online AUC maximization." Advances in neural information processing systems 29 (2016).
> >
> > [4] Mingrui Liu, Zhuoning Yuan, Yiming Ying, and Tianbao Yang. "Stochastic AUC Maximization with Deep Neural Networks." In The Eighth International Conference on Learning Representations.

---

> > > ### Author Response · Authors · 2026-03-16
> > >
> > > **A8.**
> > > In practice, we follow exactly the data roles implied by Algorithm 1 and the estimator definition. At outer iteration $t$, we use two groups of mini-batches:
> > > - one lower-level mini-batch $\zeta_t$ for the SGD update of $y$ (line 5);
> > > - an additional batch tuple $\bar\xi_t=\{\xi_t,\zeta_t^{(0)},\bar\zeta_t^{(0)},\ldots,\bar\zeta_t^{(Q-1)}\}$ for the hypergradient estimator.
> > > Within this tuple, $\xi_t$ is used for $\nabla_x F$ and $\nabla_y F$, $\zeta_t^{(0)}$ is used for $\nabla_{xy}G$, and $\zeta_t^{(q,j)}$ are used for the Hessian-vector products in the Neumann part.
> > >
> > > For hyper-representation, this means: $\xi_t$ comes from task-validation data $\mathcal D^{val}$, while all $\zeta$-type batches come from task-training data $\mathcal D^{tr}$. For AUC (where $g=-f$), they come from the same training distribution but are sampled as separate mini-batches for the different roles above.
> > >
> > > In our experiments we use a short Neumann depth ($Q=3$), so the additional second-order data/compute overhead is controlled.
> > >
> > > **A9.**
> > > In our experiments, an epoch is counted as one full pass over the corresponding outer-level training loader (meta-task loader for hyper-representation, sample loader for AUC), i.e., a fixed number of outer iterations determined by that loader.
> > >
> > > So the epoch axis is not the same as counting total stochastic oracle calls/data points consumed inside each method: algorithms with extra inner-loop updates or additional Neumann/HVP computations can indeed consume more samples and compute per epoch.
> > >
> > > For this reason, we report both performance-vs-epoch and performance-vs-running-time curves for all experiments (hyper-representation and AUC), and all methods are run on the same hardware platform. The running time results indicate that our algorithms converge much faster than other baselines.
> > >
> > > **A10.**
> > > Yes, this drift is handled in our proof, but not as a standalone term written exactly as
> > > $
> > > [\nabla_{yy}^{2} g(x_{t+1},y_t)]^{-1} - [\nabla_{yy}^{2} g(x_t,y_t)]^{-1}.
> > > $
> > > Instead, it is absorbed into the bias-control decomposition of the hypergradient estimator (see Lemmas 14 and 15), through terms such as
> > > $
> > > [\nabla_{yy}^{2} g(x_t,y_t)]^{-1} - [\nabla_{yy}^{2} g(x_t,y_t^\*)]^{-1}
> > > $
> > > and
> > > $
> > > [\nabla_{yy}^{2} g(x_{t+1},y_{t+1}^\*)]^{-1} - [\nabla_{yy}^{2} g(x_t,y_t^\*)]^{-1}.
> > > $
> > >
> > > Concretely, under Assumption 3.2(v) (joint Lipschitzness of $\nabla_{xy} g$ and $\nabla_{yy} g$), the appendix hypergradient-bias lemmas control how $\mathbb{E}\_{\bar\xi}[\hat\phi(x,y^\*(x);\bar\xi)]$ changes as $x$ moves (see Lemmas 14 and 15). In particular, in the proof of our upper-level lemma, we use
> > > $
> > > \\|\mathbb{E}\_t[\hat\phi(x_t,y_t^\*;\bar\xi_t)]-\mathbb{E}\_{t-1}[\hat\phi(x_{t-1},y_{t-1}^\*;\bar\xi_{t-1})]\\|
> > > \le (L_0+L_1\\|\nabla\Phi(x_{t-1})\\|)\\|x_t-x_{t-1}\\|.
> > > $
> > > The Hessian-inverse/Neumann contribution is included in the corresponding decomposition terms (the $A_3/A_4$-type terms in the hypergradient-bias proof), which are bounded using the $l_{g,2}$-Lipschitz property. In parallel, the induced lower-level minimizer drift $D_t=\\|y_t^\*-y_{t+1}^\*\\|$ is explicitly controlled in the recursion for $\\|y_t-y_t^\*\\|$ (see the randomness-decoupling analysis in Lemma 4.2). Therefore, both drift sources are accounted for in the final convergence argument.
> > >
> > > **A11.**
> > > The key point is that $\\|y_t-y_t^\*\\|$ may not remain uniformly small at every iteration under Adam-style upper-level updates. Thus, our statement is primarily theoretical: existing analyses [5, 6, 7] typically rely on this small-error condition, whereas for AdamBO it may not hold (see more details in A12 and A13) and we should establish a new lower-level error guarantee (i.e., Lemma 4.2).
> > >
> > > [5] Jie Hao, Xiaochuan Gong, and Mingrui Liu. "Bilevel Optimization under Unbounded Smoothness: A New Algorithm and Convergence Analysis." In The Twelfth International Conference on Learning Representations.
> > >
> > > [6] Xiaochuan Gong, Jie Hao, and Mingrui Liu. "A Nearly Optimal Single Loop Algorithm for Stochastic Bilevel Optimization under Unbounded Smoothness." In International Conference on Machine Learning, pp. 15854-15892. PMLR, 2024.
> > >
> > > [7] Xiaochuan Gong, Jie Hao, and Mingrui Liu. "An accelerated algorithm for stochastic bilevel optimization under unbounded smoothness." Advances in Neural Information Processing Systems 37 (2024): 78201-78243.

---

> > > > ### Author Response · Authors · 2026-03-16
> > > >
> > > > **A12.**
> > > > We mean both, but the more fundamental point is the second one: Adam-style updates induce a genuinely different drift structure from explicit unit-norm normalization. In [5, 6, 7], the upper-level update has (effectively) fixed length each iteration, so one can model a fixed distributional drift for the lower-level objective and then show the lower-level error remains small.
> > > >
> > > > For AdamBO, the upper-level update is
> > > > $
> > > > x_{t+1}=x_t-\frac{\eta}{\sqrt{\hat v_t}+\lambda}\odot \hat m_t,
> > > > $
> > > > so the step is coordinate-wise adaptive, history-dependent, and random (through both momentum and second-moment estimators based on stochastic hypergradients). Therefore, its length is not fixed and is coupled with the lower-level tracking error. This coupling is why prior fixed-drift proofs do not directly apply. Our randomness-decoupling lemma addresses exactly this issue by proving lower-level error control for any upper-level sequence satisfying the Adam-type bound on $\\|x_{t+1}-x_t\\|$, and then verifying that AdamBO satisfies this bound via the auxiliary-sequence analysis.
> > > >
> > > > **A13.**
> > > > We agree this is a limitation of our current theory and we have explicitly discussed it in Remark 3 in Section 4.2. In our analysis, $\lambda$ must be strictly positive because the Adam update uses
> > > > $
> > > > x_{t+1}=x_t-\frac{\eta}{\sqrt{\hat v_t}+\lambda}\odot \hat m_t,
> > > > $
> > > > and several key bounds control the update scale through factors like $\\|H_t\\|\leq \eta/\lambda$. This is the reason the current complexity has an $O(\lambda^{-2})$ dependence (Remark 3). Notably, the convergence analysis of vanilla Adam in [8] also exhibits an $O(\lambda^{-2})$ dependence in the complexity bound. So we are not claiming that "larger $\lambda$ is better" in practice; rather, finite $\lambda$ is what our present proof technique needs.
> > > >
> > > > Empirically, we still run with very small values (default $10^{-8}$), and we include a sensitivity study showing only minor performance changes when increasing $\lambda$ (up to $10^{-4}$ for AUC and $10^{-3}$ for BERT hyper-representation). This suggests practical robustness, but we agree it does not yet resolve the theoretical gap for the near-zero-$\lambda$ regime.
> > > >
> > > > Regarding why explicit normalization methods (e.g., [5]) do not show the same issue: their upper-level update is explicitly normalized to (effectively) fixed length, so they do not involve an Adam-style coordinate-wise denominator $\sqrt{\hat v_t}+\lambda$ and therefore avoid this particular $\lambda$-dependent control term. Adam’s per-coordinate adaptive scaling is a different dynamics; when some coordinates of $\hat v_t$ are small, controlling drift and lower-level tracking without a positive offset is substantially harder. We agree that improving/removing this dependence is an important future direction.
> > > >
> > > > **A14.**
> > > > We agree that average-case analyses can provide complementary and often sharper insights for practice. Our current paper deliberately uses a high-probability worst-case framework because we target stochastic nonconvex bilevel optimization with unbounded smooth upper-level objectives and Adam-type adaptive updates. In this setting, the upper-bound style assumptions (as in our Assumptions 3.2-3.4, following prior unbounded-smooth bilevel work) are what currently allow a global stationarity guarantee.
> > > > These assumptions can be conservative and are not meant to be a full model of practical training dynamics. Our goal here is to establish a first convergence result for AdamBO in this challenging setting, rather than to claim that worst-case analysis alone fully explains practice. Note that our assumptions are orthogonal to specific settings such as convex random feature models, because we focus on unbounded smooth, nonconvex objectives, while convex random feature models fall outside this regime. Extending AdamBO theory toward average-case regimes (and potentially special models such as convex random feature model) is an important future direction.
> > > >
> > > >
> > > > [8] Haochuan Li, Alexander Rakhlin, and Ali Jadbabaie. "Convergence of adam under relaxed assumptions." Advances in Neural Information Processing Systems 36 (2023): 52166-52196.

---

> > > > > ### Author Response · Authors · 2026-03-16
> > > > >
> > > > > **Minor issues:**
> > > > >
> > > > > **(a).**
> > > > > We have fixed the typos in the revised paper.
> > > > >
> > > > > **(b).**
> > > > > We have fixed the typos in the revised paper.
> > > > >
> > > > > **(c).**
> > > > > We have moved both comparison tables (Tables 1 and 2) from the Appendix F into the main paper, placing them in Section 4.2 close to where they are discussed (around Remarks 2 and 3).
> > > > >
> > > > > **(d).**
> > > > > Thank you for pointing this out. Here, the “500 meta tasks” are episodic tasks rather than 500 semantically distinct tasks. Each task is constructed by randomly selecting 2 classes from the 6 coarse-grained categories and then randomly sampling training/validation instances from those two classes. Therefore, there are at most 15 unique class-pair types, and the 500 tasks are different episodes sampled from these class pairs. We will clarify this wording in the revision to avoid the misunderstanding that all 500 tasks correspond to 500 distinct category-level tasks.
> > > > >
> > > > >
> > > > > **(e).**
> > > > > Thank you for the suggestion. We have revised Section 4.3.3 to add a roadmap paragraph explaining how the lemmas are connected in proving the main theorem.

---

> ### Comment · Reviewer_x4sK · 2026-04-01
> **Thanks for the reply**
>
> Regarding A1 and A10: I think now I understand that the hypergradient is exactly the gradient when $y$ is exactly at the minimum of $g$. In practice, $y_t$ is not exactly $y^\ast_t$, but this distance is bounded by the Randomness Decoupling Lemma, and then the difference between $\nabla_{yy}^{2} g(x_t,y_t)]^{-1}$ and $\nabla_{yy}^{2} g(x_t,y_t^*)]^{-1}$ can be bounded by the Lipschitz condition. Overall, the "true process" of which the proof is tracking the error, is as if $y_t$ stays at the minimum $y^\ast_t$ throughout the training procedure. Thanks for the explanation. The additional paragraphs in Section 4.3.3 also help understand the whole proof. I think there's no obvious logical gap in the theory.
>
> Regarding A3: Thanks for adding the experiment which shows $\|y_t-y_t^*\|$ in Appendix H. This is great.
>
> Regarding A6: My point is, if $l_{g,1}$ is a theoretically originated constant (i.e., the Lipschitz smoothness Assumption 3.2), but in practice it is treated as a hyper-parameter, there is a gap here because whether the Lipschitz condition actually holds for the constant that is chosen as the hyper-parameter in practice, is not verified. Would it be possible to show the results with different `neumann_lr`? How sensitive the results are to the choice of this hyper-parameter?
>
> Regarding A7: I mean, I understand that the theoretical interest is centered around the 2nd-order part -- it is the core component of the theory. But it would be less rigorous if we don't know how much impact the 2nd-order term actually has in practice. Would it be possible, to at least show e.g. the the norm ratio of the 2nd-order part vs the total hypergradient throughout the training process, for the tasks in this paper? I definitely want to see the graph.

---

> ### Author Response · Authors · 2026-04-06
>
> **A1.**
> In theory, $l_{g,1}$ (Assumption 3.2(iv)) is a smoothness constant used to scale the Neumann approximation in our hypergradient estimator, and the approximation error depends on the contraction factor $(1-\mu/l_{g,1})^Q$. In practice, this constant is unknown, so we use a tunable parameter $\texttt{neumann\\_lr}$ that plays the role of the scaling factor (as discussed in A6), while keeping the Neumann depth fixed at $Q=3$ in our experiments.
>
> To address this concern, we conducted a sensitivity study over $\texttt{neumann\\_lr} \in \\{1.0\times 10^{-4}, 1.0\times 10^{-3}, 1.0\times 10^{-2}, 1.0\times 10^{-1}\\}$ in hyper-representation. As shown in Figure 9 of revised Appendix I, our method is fairly stable over a moderate range of values: $1.0\times 10^{-4}, 1.0\times 10^{-3}, 1.0\times 10^{-2}$
> , both training and test accuracy are very similar, with test accuracy staying around 0.88-0.90. In contrast, performance drops substantially when $\texttt{neumann\\_lr}$ is set too large ($1.0\times 10^{-1}$), indicating that overly aggressive $\texttt{neumann\\_lr}$ can destabilize the HVP approximation. These results suggest that while $\texttt{neumann\\_lr}$ is indeed an implementation hyper-parameter, the method is not overly sensitive to it within a reasonable range.
>
> We have added Appendix I to make this hyperparameter choice clear and include the sensitivity results to clarify the empirical robustness of the method.
>
>
> **A2.**
> We first reemphasize our point in A7: the second-order term is not an optional component that we invent, but a required part of the bilevel hypergradient. Under implicit differentiation, the target gradient is
> $$
> \nabla\Phi(x) = \nabla_x f(x,y^\*(x)) - \nabla_{xy} g(x,y^\*(x))[\nabla_{yy}^2 g(x,y^*(x))]^{-1}\nabla_y f(x,y^\*(x)).
> $$
> Our estimator follows this structure via the Neumann approximation,
> $$
> \hat\phi_t = g_t^{(1)} - g_t^{(2)} = \nabla_x F(x_t,y_t;\xi_t)-\nabla_{xy}G(x_t,y_t;\zeta_t\^{(0)})P_Q\nabla_y F(x_t,y_t;\xi_t).
> $$
> So the hypergradient estimator is fundamentally different from an ordinary first-order gradient update using only $\nabla_x F$; removing $g_t^{(2)}$ gives a first-order surrogate direction and no longer estimates the bilevel gradient.
>
> Specifically, for the task of hyper-representation, we track the ratio of the second-order part to the total hypergradient:
> $
> r_t = \frac{\\|g_t^{(2)}\\|}{\\|\hat{\phi}_t\\|},
> $
> where $g_t^{(2)}$ denotes the second-order component and $\hat{\phi}_t$ is the total hypergradient. In practice, at the end of each epoch, we sample one mini-batch and compute this ratio to monitor how the relative contribution of the second-order term evolves during training.
>
> Our result in Figure 10 of revised Appendix J show a clear stage-dependent pattern: the ratio is relatively large in the early phase of training, indicating that the second-order term makes a substantial contribution when the model is still far from convergence. As training proceeds, the ratio decreases steadily and becomes much smaller in later epochs, which is consistent with the intuition that curvature-related corrections become less pronounced once optimization enters a more stable regime.
>
> Therefore, this result suggests that the second-order term is not merely of theoretical interest. It has a practically meaningful impact, especially in the early stage of training, while its relative importance naturally diminishes as the model converges. We have included this result and discussion in Appendix J.

---

> > ### Comment · Reviewer_x4sK · 2026-04-06
> > **Thanks**
> >
> > The additional experiments clarifies a lot how the theory would take effect in practice.

---

> > > ### Author Response · Authors · 2026-04-06
> > >
> > > Thank you for your helpful comments. We appreciate your constructive feedback in improving our work.

---

### Decision · Action_Editor_M1RG · 2026-06-02

**Recommendation:** Accept as is

**Additional Comments:**

This paper introduces AdamBO, a single-loop Adam-type algorithm for stochastic bilevel optimization under unbounded smoothness conditions. Its primary strength lies in its solid theoretical results, which provide the first rigorous sample complexity analysis ($\tilde{O}(\epsilon^{-4})$) for an Adam-style optimizer for bilevel optimization. Backed by a novel "randomness decoupling lemma" that maintains precise control over the tracking error of lower-level variables, the paper convincingly demonstrates faster convergence and empirical performance on modern deep architectures, including RNNs and Transformers.

**Audience:**

Yes

**Audience Explanation:**

Bi-level optimization finds a good number of applications in machine learning, and provable methods for solving bilevel optimization problems are of interest for both theory and practice

**Claims And Evidence:**

Yes

**Claims Explanation:**

The primary theoretical claims are supported by complete proofs, and experiments on two tasks are included to demonstrate the practical relevance of the algorithm